# Damage-induced IL-18 stimulates thymic NK cells limiting endogenous tissue regeneration

**David Granadier** [1,2] ✉, **Kirsten Cooper** [1], **Dante Acenas II** [1,3], **Anastasia Kousa** [1,4], **Makya Warren** [1], **Vanessa Hernandez** [1], **Lorenzo Iovino** [1], **Paul deRoos** [1], **Emma E. Lederer** [1,3], **Steve Shannon-Sevillano** [1], **Sinéad Kinsella** [1], **Cindy Evandy** [1], **Marcel R. M. van den Brink** [4], **Andri Lemarquis** [4] & **Jarrod A. Dudakov** [1,3] ✉

Interleukin-18 (IL-18) is an acute-phase proinflammatory molecule crucial for mediating viral clearance by activating T helper 1 CD4[+] T cells, cytotoxic CD8[+] T cells and natural killer (NK) cells. Here, we show that mature IL-18 is generated in the thymus following numerous distinct forms of tissue damage, all of which cause caspase-1-mediated immunogenic cell death. We report that IL-18-stimulated cytotoxic NK cells limit endogenous thymic regeneration, a critical process that ensures the restoration of immune competence after acute insults such as stress, infection, chemotherapy and radiation. NK cells suppress thymus recovery by aberrantly targeting thymic epithelial cells, which act as the master regulators of organ function and regeneration. Together, our data reveal a new pathway regulating tissue regeneration in the thymus and suggest IL-18 as a potential therapeutic target to boost thymic function. Moreover, given the enthusiasm for IL-18 as a cancer immunotherapy due to its capacity to elicit a type 1 immune response, these findings also offer insight into potential off-target effects.

Despite its importance for the production of a diverse and tolerant T cell repertoire, the thymus is exquisitely sensitive to acute insults such as infection and stress-induced increases in corticosteroid levels, as well as to more profound injuries including those caused by chemotherapy and myeloablative conditioning before hematopoietic cell transplantation (HCT)[1–3]. The thymus harbors an endogenous capacity for regeneration; however, this prolonged process leaves patients who receive thymus-damaging treatments vulnerable to extended periods of lymphopenia[4]. This is especially pertinent in HCT recipients, who are particularly vulnerable to opportunistic infections and malignant relapse[5–7]. Therefore, understanding the mechanisms underlying thymus recovery could offer therapeutic targets for improving T cell

reconstitution[1]. Several molecules mediating endogenous thymic regeneration have been identified, including interleukin-22 (IL-22), bone morphogenetic protein 4 (BMP4), keratinocyte growth factor and amphiregulin, all of which stimulate thymic epithelial cells (TECs), which are key components of the thymic microenvironment that support T cell development[8–12]. Yet, there remains no clinically approved strategy for treating T cell lymphopenia.

We have previously reported that HCT conditioning leads to an acute increase in not only apoptosis but also pyroptosis—a form of immunogenic cell death and a key trigger for the thymic regenerative response[13–17]. Pyroptosis also causes the release of inflammatory cytokines such as IL-1β and IL-18 (ref. 13), a potent stimulator of type II

[1]Translational Science and Therapeutic Division and Immunotherapy Integrated Research Center, Fred Hutchinson Cancer Center, Seattle, WA, USA. [2]Medical Scientist Training Program, University of Washington School of Medicine, Seattle, WA, USA. [3]Department of Immunology, University of Washington School of Medicine, Seattle, WA, USA. [4]Beckman Research Institute, City of Hope, Duarte, CA, USA. ✉e-mail: dgranadi@fredhutch.org; jdudakov@fredhutch.org

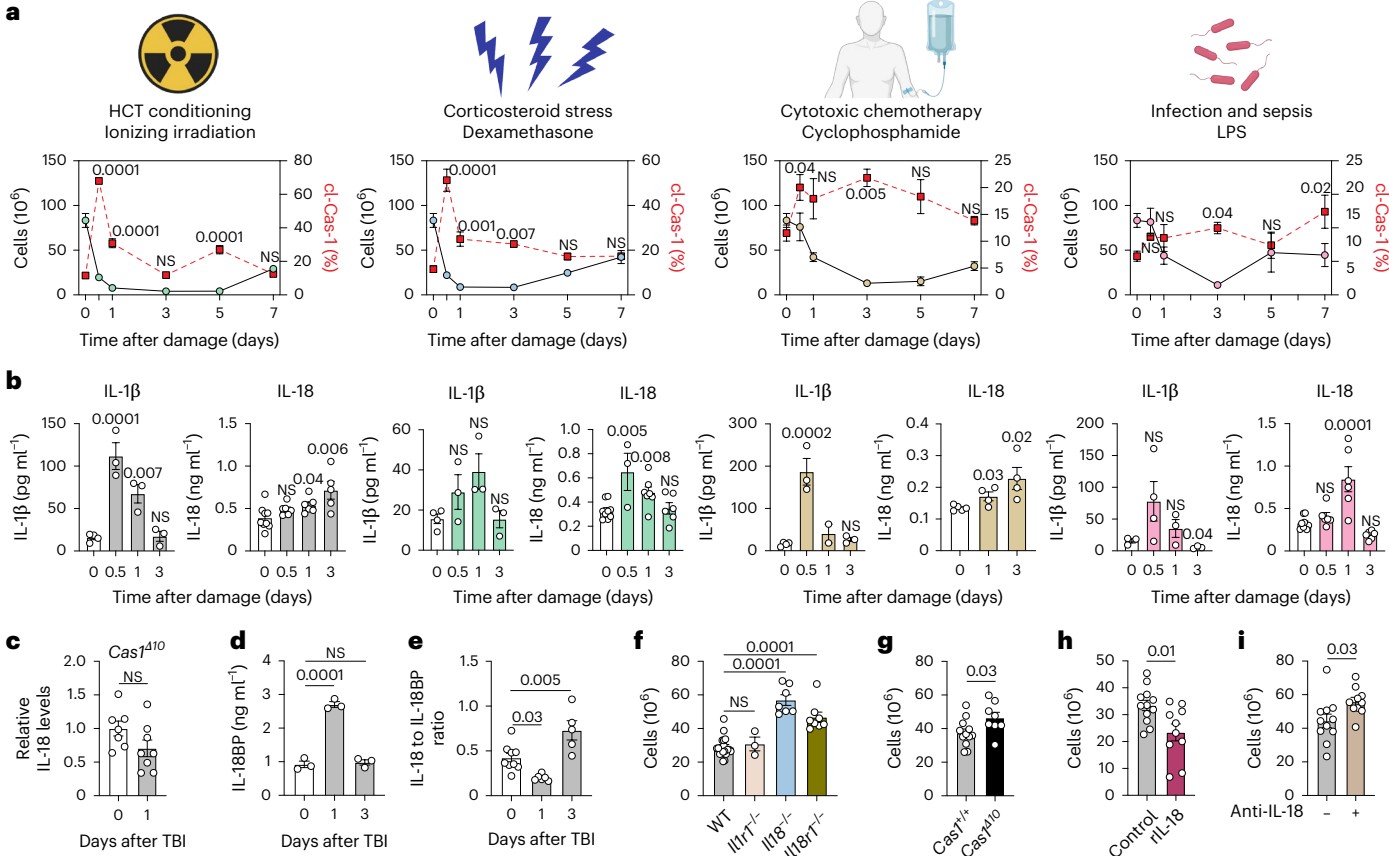

**Fig. 1 | Acute thymic damage triggers the cleavage of Cas-1 and the activation of IL-18, which suppresses thymus regeneration. a,b,** Female 1- to 2-month-old C57/BL6 mice were administered SL-TBI (550 cGy), dexamethasone (intraperitoneal (i.p.) injection, 20 mg kg⁻¹), cyclophosphamide (i.p., 200 mg kg⁻¹) or LPS (i.p., 1.5 mg kg⁻¹). **a,** Thymus cellularity (black) and cl-Cas-1 expression (red) were measured using fluorescently conjugated FAM-YVAD-FMK (a fluorescent probe that irreversibly binds and labels cl-Cas-1) in mice killed at baseline ($n = 15$), day 0.5 ($n = 7$), day 1 ($n = 8$), day 3 ($n = 8$), day 5 ($n = 4$) and day 7 ($n = 4$) after treatment; all statistics are compared to day 0. **b,** Amount of active IL-1β and active IL-18 in the thymus, measured by ELISA at the indicated time points after SL-TBI (IL-18: day 0, $n = 9$; day 0.5, $n = 6$; day 1, $n = 6$; day 3, $n = 5$; IL-1β: day 0, $n = 4$; day 0.5, $n = 3$; day 1, $n = 3$; day 3, $n = 3$), dexamethasone (i.p., 20 mg kg⁻¹) (IL-18: day 0, $n = 9$; day 0.5, $n = 3$; day 1, $n = 7$; day 3, $n = 6$; IL-1β: day 0, $n = 4$; day 0.5, $n = 3$; day 1, $n = 3$; day 3, $n = 3$), cyclophosphamide (i.p., 200 mg kg⁻¹) (IL-18: day 0, $n = 5$; day 1, $n = 4$; day 3, $n = 4$; IL-1β: day 0, $n = 4$; day 0.5, $n = 3$; day 1, $n = 2$; day 3, $n = 3$) or LPS (i.p., 1.5 mg kg⁻¹) (IL-18: day 0, $n = 9$; day 0.5, $n = 6$; day 1, $n = 6$; day 3, $n = 6$; IL-1β: day 0, $n = 3$; day 0.5, $n = 4$; day 1, $n = 3$; day 3, $n = 3$); all statistics are compared to day 0. **c,** Amount of active IL-18 measured by ELISA in thymuses of female 1- to 2-month-old $Cas1^{\Delta10}$ mice on day 0 ($n = 7$) and day 1 ($n = 8$) after SL-TBI.

**d,** Amount of IL-18BP in thymuses of 1- to 2-month-old C57/BL6 WT mice on days 0, 1 and 3 after SL-TBI ($n = 3$ per group). **e,** Ratio of active IL-18 to IL-18BP averaged on day 0 ($n = 9$), day 1 ($n = 6$) and day 3 ($n = 5$) after SL-TBI, representing the amount of free active IL-18. **f,** Female 1- to 2-month-old C57/BL6 WT ($n = 18$), $Il1r1^{-/-}$ ($n = 3$), $Il18^{-/-}$ ($n = 7$) and $Il18r1^{-/-}$ ($n = 8$) mice were exposed to SL-TBI, and thymus cellularity was measured 7 days later. **g,** Female 1- to 2-month-old C57/BL6 WT ($n = 7$) or $Cas1^{\Delta10}$ ($n = 8$) mice were exposed to SL-TBI, and thymus cellularity was measured on day 7. **h,** Female 1- to 2-month-old C57/BL6 WT mice were exposed to SL-TBI and then administered PBS vehicle ($n = 12$) or rIL-18 ($n = 10$) on day 3 (subcutaneous (s.c.) injection, 2.5 mg kg⁻¹); thymuses were isolated on day 7. **i,** Female 1- to 2-month-old C57BL/6 mice were lethally irradiated and transplanted (intravenous (i.v.) injection) with $5 \times 10^6$ CD45.1⁺ WT bone marrow hematopoietic cells. Recipient mice were treated with 200 μg of anti-IL-18 mAb ($n = 10$) or equal-volume control (PBS) ($n = 11$), and thymus cellularity was measured on day 50 following transplant. Graphs represent mean ± s.e.m.; each dot represents an individual biological replicate; NS, not significant. Statistics were generated for **a**, **b** and **d**–**f** using one-way analysis of variance (ANOVA) with Dunnet's correction for multiple comparisons and for **c** and **g**–**i** using unpaired two-tailed $t$ tests. Panel **a** icons created with BioRender.com.

interferon (IFNγ) and cytotoxicity in natural killer (NK) cells[18,19]. Here, we examined the impact of this proinflammatory cascade and identified that acute thymus damage induces the release of IL-18, which in turn suppresses the endogenous mechanisms of organ recovery by stimulating resident cytotoxic NK cells that aberrantly target TECs.

## Results

### Acute thymus injury leads to caspase-1 cleavage and release of active IL-1β and IL-18

As part of normal T cell development, the vast majority of CD4⁺CD8⁺ double-positive thymocytes and CD4⁺ or CD8⁺ single-positive thymocytes undergo apoptosis during positive and negative selection[20,21]. Importantly, apoptosis is an immunologically silent process, and there is minimal inflammation within the homeostatic thymus[14]. Following

acute damage, such as that caused by pre-HCT cytoreductive conditioning (modeled by sublethal total body irradiation (SL-TBI), 550 cGy), thymus cellularity precipitously declines[8,22] (Fig. 1a). This ionizing radiation damage leads to cell death by both apoptosis and pyroptosis within the thymus[16]. In contrast to the immunologically silent apoptosis, pyroptosis is a form of immunogenic cell death mediated by cleaved caspase-1 (cl-Cas-1)[13]. Cleavage of Cas-1 occurred not only following ionizing radiation but also after all other stimuli causing acute thymus injury: corticosteroid-induced stress, cytoreductive chemotherapy and lipopolysaccharide (LPS) treatment (Fig. 1a and Extended Data Fig. 1a), all of which have been shown to induce acute thymic involution[23]. cl-Cas-1 mediates the proteolytic cleavage of the immature, inactive forms of IL-1β and IL-18 into their mature, inflammatory states. Accordingly, increased cl-Cas-1 levels corresponded

with increased activation of IL-1β and IL-18 within the thymus following each of these acute damage models (Fig. 1b), and the levels of active IL-18 did not increase in mice lacking the catalytic domain of Cas-1 (Fig. 1c). IL-18 binding protein (IL-18BP), an endogenous antagonist of IL-18, was also upregulated following HCT conditioning (Fig. 1d), possibly in response to the upregulation of activated IL-18 (ref. 24). Despite an early upregulation of IL-18BP, the ratio of IL-18 to IL-18BP increased by day 3 following SL-TBI, suggesting higher levels of free IL-18 (Fig. 1e).

To explore the functional involvement of these cytokines in regulating thymus regeneration, we assessed thymus recovery after SL-TBI in mice with germline deletions in IL-1β ($Il1r1^{-/-}$) or IL-18 signaling ($Il18^{-/-}$ and $Il18r1^{-/-}$). While $Il1r1^{-/-}$ mice showed no changes in thymus cellularity, suggesting a minimal role in IL-1 signaling, mice deficient in either IL-18 itself or its primary receptor, IL-18R1, exhibited improved thymus regeneration relative to wild-type (WT) controls (Fig. 1f). Mice lacking the catalytic domain of Cas-1 also showed increased thymus regeneration (Fig. 1g)[16,17]. Administration of recombinant IL-18 (rIL-18) 3 days following SL-TBI—the point at which organ cellularity reaches a nadir and regenerative processes begin to take effect—delayed thymic reconstitution (Fig. 1h). Mice lacking IL-18 but not IL-18R1 demonstrated higher cellularity than WT controls at baseline (Extended Data Fig. 1b), but all strains showed similar degrees of thymus involution 3 days after SL-TBI (Extended Data Fig. 1b). Inflammatory cytokines can stimulate the hypothalamic–pituitary axis, resulting in increased levels of glucocorticoids, which are known to trigger thymus involution[23,25]. Notably, mice deficient in IL-18 showed no changes in cortisol levels, suggesting a glucocorticoid-independent mechanism of action (Extended Data Fig. 1c). Taken together, these data demonstrate that damage-induced increases in IL-18 levels suppress endogenous thymus repair.

We next assessed the therapeutic potential of abrogating IL-18 signaling by treating HCT-recipient mice with an anti-IL-18 monoclonal antibody (mAb) for 2–3 weeks following HCT to capture both the acute spike in active IL-18 and its homeostatic presence during thymus recovery. Mice receiving anti-IL-18 mAb treatment showed greater thymus cellularity 50 days after HCT (Fig. 1i). Having established that IL-18 regulates endogenous thymus recovery and can be therapeutically targeted in the context of HCT, we set out to identify its source(s) and mechanism of action within the organ.

## IL-18 is produced by discrete populations of hematopoietic and nonhematopoietic stromal cells

Unlike IL-1β, which is upregulated following inflammasome stimulation, IL-18 is constitutively expressed in its proform within the cytoplasm of several cell types, awaiting activation by proteolytic cleavage[19,24]. To identify the source of IL-18 following acute damage, we investigated $Il18$ gene expression from previously published gene expression datasets (Extended Data Fig. 2a)[26]. At baseline, $Il18$ was not expressed by thymocytes beyond the diverse CD4$^-$CD8$^-$CD44$^+$CD25$^-$ DN1 (double-negative 1) population, which includes early T cell precursors, myeloid cells, B cells and innate lymphoid cells (ILCs)[27]. Given that $Il18$ expression was not found in mature thymocytes, we used single-cell RNA sequencing (scRNAseq) on all nonthymocyte populations by using $Rag2^{GFP}$ mice to exclude all $Rag2$-GFP$^+$ T cell lineage-committed cells (Fig. 2a and Extended Data Fig. 2b)[12,28]. Using this comprehensive gene expression dataset, $Il18$ expression was isolated to nonhematopoietic mesothelial cells (MECs) and capsular fibroblasts, as well as type 1 classical dendritic cells (cDC1s) and macrophages (Fig. 2b and Extended Data Fig. 2c). An increased amount of cl-Cas-1 was detected within cDC1s, macrophages, fibroblasts and MECs early after SL-TBI, but there was no change in cl-Cas-1 expression within TECs or endothelial cells (Fig. 2c and Extended Data Fig. 3). This indicates that myeloid cDC1s and rare CD45$^-$ capsular MEC/fibroblast populations meet the qualifications of (1) expressing $Il18$ at baseline and (2) increasing

cl-Cas-1 expression following injury, which is necessary for the proteolytic cleavage of immature pro-IL-18. To functionally investigate these sources, we generated mice with a specific deletion of $Il18$ in cDCs using the $Zbtb46$-Cre line ($Il18^{\Delta cDC}$)[29]. WT but not $Il18^{\Delta cDC}$ mice showed significantly increased levels of IL-18 on day 1 after TBI (Fig. 2d). To assess the contribution of nonhematopoietic stromal cells, such as MECs and fibroblasts, we generated bone marrow chimeras using WT (WT→WT) or $Il18^{-/-}$ (WT→$Il18^{-/-}$) mice as recipients of WT bone marrow. Recipient mice were allowed to recover for 10 weeks following transplantation, at which point they were subjected to SL-TBI. Similarly to mice deficient in IL-18 in cDCs, these chimeric mice demonstrated an increase in IL-18 levels in WT recipients but not in $Il18^{-/-}$ recipients (Fig. 2e). Taken together, these data suggest that multiple populations including myeloid cells such as cDCs and macrophages, along with nonhematopoietic stromal cells, contribute to the release of active IL-18 following acute thymus damage.

## IL-18 suppression of thymus regeneration is not mediated through a direct effect on TECs or hematopoiesis

To determine the potential cellular targets of IL-18, we first examined previously described transcriptome datasets for the expression of IL-18R subunit-encoding genes ($Il18r1$ and the coreceptor-encoding gene $Il18rap$)[12,26,28]. At baseline, the $Il18r1$ subunit was expressed by multiple populations, including cortical TECs (cTECs), medullary TECs (mTECs), regulatory T cells, ILCs, NK cells and NKT cells (Fig. 3a and Extended Data Fig. 2a). $Il18r1$ expression in TECs was notable given the role of these cells as master regulators of thymus function[30]; however, there was minimal IL-18R protein expression in TECs (Fig. 3b,c and Extended Data Figs. 3 and 4). Moreover, the deletion of $Il18r1$ in TECs using the $Foxn1$-cre line ($Il18r1^{\Delta TEC}$) did not alter regeneration (Fig. 3d). Prior work has established that IL-18 can induce hematopoietic stem cell quiescence[31–33]. While we did not observe IL-18R expression in most thymocytes or bone marrow-resident precursor populations (Fig. 3c,e,f and Extended Data Figs. 3 and 4), a low level of IL-18R expression was noted in early thymic progenitors, which represent the earliest stage of thymocyte development (Fig. 3c and Extended Data Fig. 4). We explored whether $Il18r1^{-/-}$ thymocytes exhibit increased reconstitution capacity in a competitive transplantation model (Fig. 3g). At 2 weeks following transplantation, a time point representing early thymus recovery, $Il18r1^{-/-}$ cells had no competitive advantage in seeding the thymus (Fig. 3h). Measuring the longitudinal contribution of donor-derived hematopoiesis by peripheral blood monitoring over 120 days, we found similar reconstitution in overall hematopoietic cells and T cells from WT and $Il18r1^{-/-}$ donor populations (Fig. 3i). Following additional SL-TBI at this late time point, $Il18r1^{-/-}$ donor cells again showed similar reparative capacity (Fig. 3j,k). Despite having little impact in vivo, IL-18 increased thymocyte differentiation and proliferation in coculture studies of ex vivo bone marrow-derived hematopoietic precursors and the OP9-DLL1 system (Extended Data Fig. 5a–c), consistent with prior literature[34]. From these data, we conclude that IL-18 does not directly suppress thymus function through TECs or T lineage progenitors.

## Damage-resistant NK cells suppress thymus regeneration after acute injury

IL-18 can signal through the IL-18R1 subunit but is potentiated by the coexpression of the IL-18 receptor accessory protein (IL-18RAP)[24,35]. Consistent with their lack of functional signaling after damage, the transcriptome datasets indicate that neither TECs nor thymocytes express $Il18rap$ (Fig. 3a and Extended Data Fig. 2a). The expression of both $Il18r1$ and $Il18rap$ was largely restricted to ILCs—including NK/ILC1 cells—and NKT cells (Fig. 3a). Consistent with this, protein expression suggested that NK1.1$^+$ populations, including NKT, NK and ILC1 cells, strongly expressed IL-18R at baseline within the thymus (Fig. 3b,c). In contrast to NKT cells, NK and ILC1 cells comprised only

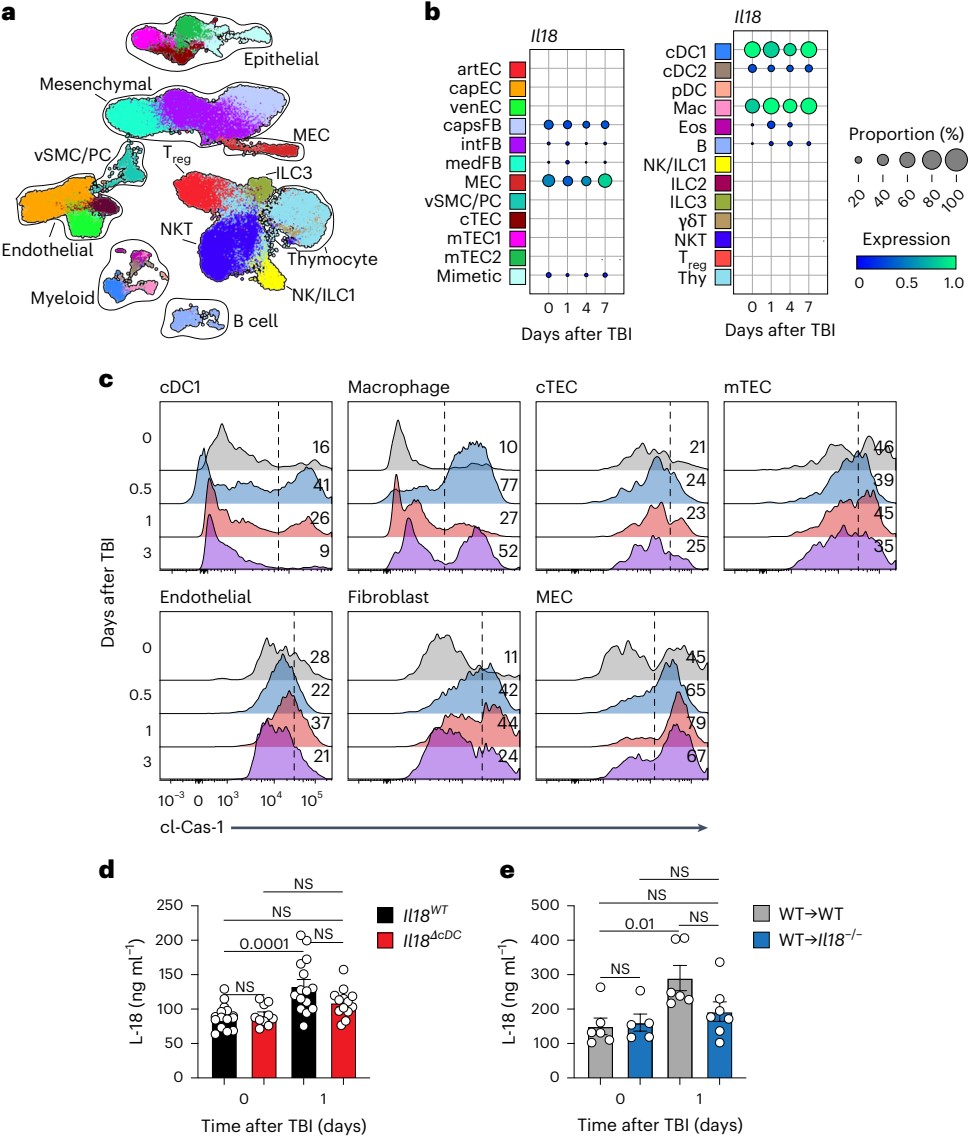

**Fig. 2 | Hematopoietic and nonhematopoietic sources of IL-18 after damage.**
**a,b**, scRNAseq was performed on (1) nonthymocyte CD45⁺ stromal cells (CD45⁺
*Rag2*-GFP⁻ cells isolated from female 1- to 2-month-old *Rag2^GFP* mice) and (2)
CD45⁻ stromal cells isolated from thymuses of female 1- to 2-month-old C57BL/6
mice at baseline and on days 1, 4 and 7 following SL-TBI. Data were previously
integrated and published in ref. 12. **a**, Integrated UMAP of both hematopoietic
and nonhematopoietic cells from datasets, showing undamaged cells, major
clusters in the thymus at baseline and annotation. **b**, Expression of *Il18* by
population. artEC, arterial endothelial cell; capEC, capillary endothelial cell;
venEC, venous endothelial cell; capsFB, capsular fibroblast; intFB, intermediate
fibroblast; medFB, medullary fibroblast; vSMC/PC, vascular smooth muscle/
pericyte; cTEC, cortical TEC; mTEC1 and mTEC2, medullary TECs; cDC, classical
dendritic cell; pDC, plasmacytoid dendritic cell; Mac, macrophage; Eos,
eosinophil; B, B cell; NK/ILC1, NK and type 1 ILC; ILC2, type 2 ILC; ILC3, type 3
ILC; γδT, γδ T cell; NKT, NK T cell; T_reg, regulatory T cell; Thy, thymocyte. Colors

represent unbiased clusters. **c**, cl-Cas-1 expression measured using fluorescently
conjugated FAM-YVAD-FMK in cDC1s, macrophages, cTECs, mTECs, endothelial
cells, fibroblasts and MECs on days 0, 0.5, 1 and 3 after SL-TBI (*n* = 3–5 per group).
Gating and phenotypes can be found in Extended Data Fig. 3. **d**, Amount of
active IL-18 measured by ELISA in female 1- to 2-month-old *Il18^fl/fl*:*Zbtb*-Cre⁻
(*Il18^WT*, *n* = 15 per group) and *Il18^fl/fl*:*Zbtb*-Cre⁺ (*Il18^ΔcDC*; day 0, *n* = 10; day 1, *n* = 13)
mice on day 0 or 1 following SL-TBI. **e**, Female 1- to 2-month-old C57/BL6 WT
(WT→WT) or *Il18^−/−* (WT→*Il18^−/−*) mice were lethally irradiated (2 × 550 cGy) and
transplanted with 5 × 10⁶ CD45.1⁺ WT bone marrow hematopoietic cells. At 10
weeks after transplantation, recipient mice were administered a second dose
of SL-TBI (550 cGy), and active IL-18 was measured at baseline and on day 1 after
this subsequent damage (WT→WT: *n* = 6 per group; WT→*Il18^−/−*: day 0, *n* = 5;
day 1, *n* = 7). Graphs represent mean ± s.e.m.; each dot represents an individual
biological replicate. Statistics were generated for **d** and **e** using one-way ANOVA
with Tukey's correction for multiple comparisons.

IL-18R^hi and IL-18R^lo−neg populations (Fig. 3b). Cell–cell interaction mod-
eling revealed that the targets of IL-18 included the NK/ILC1, NKT and
ILC3 subsets, with NK/ILC1 cells exhibiting the strongest aggregate
interactome score (Fig. 4a). To assess the radioresistance of NK and
NKT cells, we performed congenic HCT following myeloablative
conditioning and tracked host-derived NK and NKT cells. Although
recipient IL-18R⁺ NKT and NK cells were both more resistant to damage
compared to the more abundant thymocyte populations, resulting
in an increase in their relative frequency within the organ in the days

following transplantation, NKT cells decreased in absolute number
while NK cells maintained their number and even transiently expanded
early after HCT (Fig. 4b–d). These findings were confirmed by imag-
ing studies that showed an increased frequency of NKp46⁺ cells in the
thymus early after damage in both the cortex and medulla (Fig. 4e).

To assess the role of NK1.1⁺ cells in limiting thymic recovery, we
treated mice with an anti-NK1.1 mAb and exposed them to SL-TBI.
Treatment with the anti-NK1.1 mAb achieved near-complete ablation
of thymic IL-18R⁺NK1.1⁺ cells (Extended Data Fig. 5d). Mice depleted

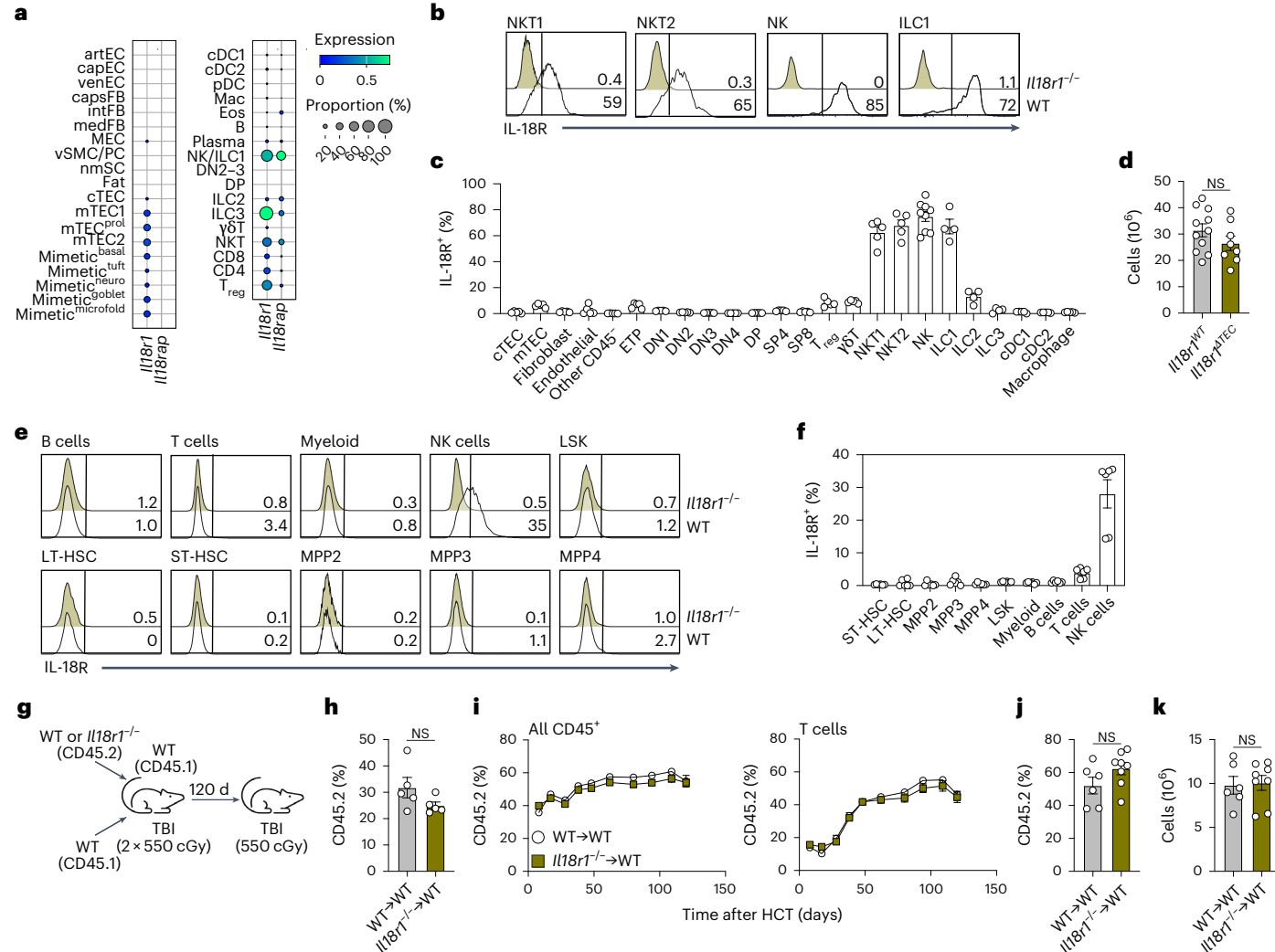

**Fig. 3 | IL-18 suppression of thymus function after damage is not mediated directly through TECs or hematopoietic progenitors. a,** Standard scaled dot plot of *Il18r1* and *Il18rap* gene expression by population of cells from thymuses of female 1- to 2-month-old C57BL/6 mice at baseline, taken from the scRNAseq dataset described in Fig. 2a. nmSC, nonmyelinating Schwann cells; mTEC^prol, proliferating mTECs. **b,** Concatenated flow cytometry plots showing the expression of IL-18R in CD45⁺NK1.1⁺TCRβ⁺ CD1d-αGalCer tetramer⁺ (NKT1) and CD1d-αGalCer tetramer⁻ (NKT2) invariant NK cells, CD45⁺NK1.1⁺TCRβ⁻CD49b⁺ NK cells and CD45⁺NK1.1⁺TCRβ⁻CD49a⁺ ILC1s (*n* = 5 per group). Gates were based on expression in *Il18r1⁻/⁻* mice. **c,** Percentage of IL-18R-expressing cTECs, mTECs, fibroblasts, endothelial cells, other CD45⁻ cells, early thymic progenitors (ETP), thymocytes (DN1–4, double-positive (DP)), and single-positive CD4⁺ (SP4) and CD8⁺ (SP8) cells), T_reg cells, γδ T cells, NK cells (*n* = 9), ILC1s (*n* = 4), ILC2s (*n* = 4), ILC3s (*n* = 4), cDC1s, cDC2s and macrophages (*n* = 5 per group unless otherwise specified). **d,** Female 1- to 2-month-old *Il18r1^fl/fl^:Foxn1*-Cre⁻ (*Il18r1^WT^*, *n* = 11) and *Il18r1^fl/fl^:Foxn1*-Cre⁺ (*Il18r1^ΔTEC^*, *n* = 8) mice were exposed to SL-TBI, and thymus cellularity was assessed 7 days later. **e,f,** Bone marrow populations

were measured for IL-18R expression (*n* = 6 per group), shown as flow cytometry plots (**e**) and percentage of positive cells (**f**). LSK, lineage (Lin)⁻Sca-1⁺c-Kit⁺ cells; LT-HSC, long-term hematopoietic stem cells; ST-HSC, short-term hematopoietic stem cells; MMP2–4, multipotent progenitors. **g,** Female 1- to 2-month-old WT CD45.1⁺ mice were lethally irradiated and transplanted (i.v.) with 2.5 × 10⁶ WT CD45.1⁺ bone marrow cells and 2.5 × 10⁶ bone marrow cells from either CD45.2⁺ WT or *Il18r1⁻/⁻* mice. **h,** Contribution of CD45.2⁺ cells in the thymus at 2 weeks following transplant. **i,** Contribution of CD45.2⁺ cells to the total CD45⁺ cell (left) or T cell (right) reconstitution in peripheral blood over 17 weeks after transplantation (WT→WT *n* = 6; *Il18r1⁻/⁻*→WT *n* = 8). **j,k,** At 17 weeks after transplantation, recipient mice were administered a subsequent dose of SL-TBI (550 cGy). Thymuses were collected after 7 days, and the percentage of CD45.2⁺ cells relative to all thymic CD45⁺ cells (**j**) and the total thymus cellularity (**k**) were measured (WT→WT: *n* = 6; *Il18r1⁻/⁻*→WT: *n* = 8). Graphs represent mean ± s.e.m.; each dot represents an individual biological replicate. Statistics were generated for **d, h, j** and **k** using unpaired two-tailed *t* tests.

of NK1.1⁺ cells exhibited increased thymus cellularity compared to controls (Fig. 4f). Notably, the improved regeneration observed upon anti-NK1.1 mAb treatment in WT thymuses was not recapitulated when the treatment was performed in *Il18⁻/⁻* or *Il18r1⁻/⁻* mice, suggesting that NK1.1⁺ cell control of thymus recovery is dependent on IL-18 (Fig. 4g). To distinguish between the roles of NK1.1⁺IL-18R⁺ NK/ILC1 cells and NKT cells in regulating thymus regeneration, *Cd1d⁻/⁻* mice lacking the antigen-presenting machinery for NKT cell development were subjected to SL-TBI to test the effects of thymus suppression in the absence of NKT cells[36]. *Cd1d⁻/⁻* mice exhibited similar thymus cellularity

to controls for up to 35 days following injury (Fig. 4h and Extended Data Fig. 5e), indicating that IL-18R⁺ NKT cells, while more abundant than IL-18R⁺ NK/ILC1 cells, do not mediate the suppression of regeneration. Consistent with this, mice in which *Il18r1* was deleted from early in thymocyte development using an *Lck-cre* driver (*Il18r1^ΔT/NKT^*), leading to deletion in NKT and T cells but not in NK/ILC1 cells, showed no difference in thymic repair after TBI (Fig. 4i and Extended Data Fig. 5f). In contrast, mice generated with a specific deletion of *Il18r1* in NK/ILC1 cells using the *Ncr1-cre* strain (*Il18r1^ΔNK/ILC1^*), restricting the deletion to NKp46-expressing cells[37], exhibited an increase in thymus regeneration,

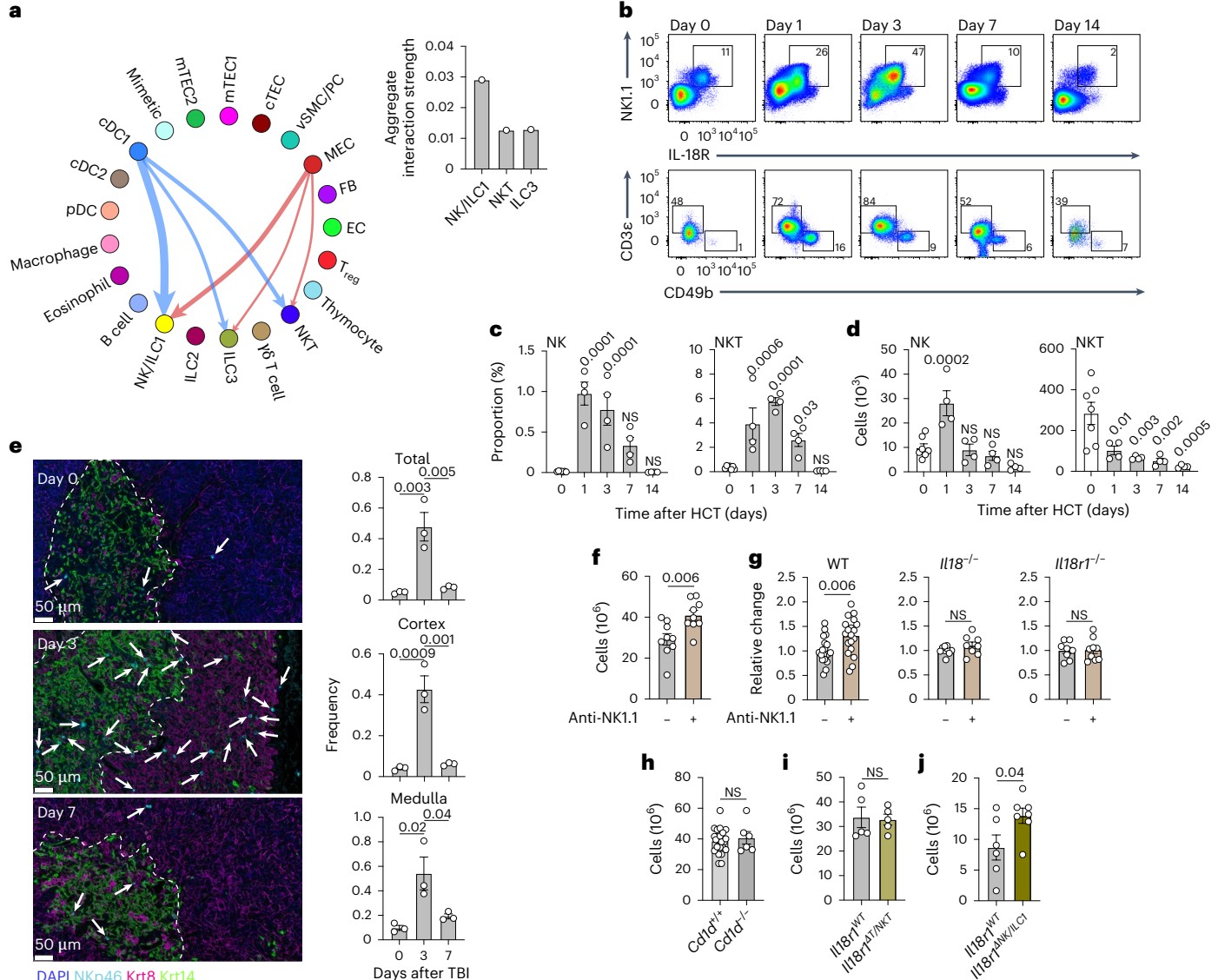

**Fig. 4 | Damage-resistant IL-18R⁺ NK cells suppress thymus repair. a**, CellChat interaction analysis for IL-18 at baseline and following SL-TBI, taken from the scRNAseq dataset described in Fig. 2a, with quantification of the aggregate signal strength for each IL-18 target cell. **b–d**, Female 1- to 2-month-old C57BL/6 CD45.2⁺ mice were lethally irradiated and transplanted (i.v.) with $5 \times 10^6$ WT CD45.1⁺ bone marrow cells. **b**, Concatenated flow cytometry plots showing CD45⁺CD45.1⁻CD4⁻CD8⁻ cells (top) and CD45⁺CD45.1⁻CD4⁻CD8⁻NK1.1⁺IL-18R⁺ cells gated on CD3⁺ NKT cells and CD49b⁺ NK cells (bottom) from thymus-recipient mice at the indicated time points after HCT ($n = 4$–7 per time point). **c,d**, Proportion (**c**) and total number (**d**) of recipient NK or NKT cells before HCT (day 0; $n = 7$) and on days 1, 3, 7 and 14 after HCT ($n = 4$ per group). **e**, Thymuses of female 1- to 2-month-old C57BL/6 mice were visualized at steady state or on day 3 or 7 after SL-TBI at 12×, assessing keratin-14-positive (Krt14⁺) mTECs (green), keratin-8-positive (Krt8⁺) cTECs (pink) and NKp46⁺ NK cells (arrows). The NKp46⁺ NK/ILC1 cell distribution within the thymus cortex or medulla at 0, 3 and 7 days after SL-TBI is shown ($n = 3$ per group). **f**, Female 1- to 2-month-old WT C57BL/6 mice were administered 200 μg of anti-NK1.1 mAb or control PBS (i.p.) on days −1, 1 and 3 after SL-TBI, and thymus cellularity was assessed on day 7

($n = 9$ per group). **g**, Female 1- to 2-month-old C57BL/6 WT, $Il18^{-/-}$ and $Il18r1^{-/-}$ mice were administered 200 μg of anti-NK1.1 mAb or isotype/PBS (i.p.) as above. The relative change in thymus cellularity is shown, comparing control-treated (WT, $n = 17$; $Il18^{-/-}$, $n = 9$; $Il18r1^{-/-}$, $n = 8$) and anti-NK1.1 mAb-treated (WT, $n = 21$; $Il18^{-/-}$, $n = 8$; $Il18r1^{-/-}$, $n = 9$) mice within each strain 7 days after SL-TBI. **h**, Female 1- to 2-month-old C57BL/6 WT ($Cd1d^{+/+}$, $n = 9$) and $Cd1d^{-/-}$ ($n = 6$) mice were exposed to SL-TBI, and thymus cellularity was measured 7 days later. **i**, Female 1- to 2-month-old $Il18r1^{fl/fl}$:Lck-Cre⁻ ($Il18r1^{WT}$, $n = 5$) and $Il18r1^{fl/fl}$:Lck-Cre⁺ ($Il18r1^{ΔT/NKT}$, $n = 5$) mice were exposed to SL-TBI and administered rIL-18 (s.c., 2.5 mg kg⁻¹) on day 3. Thymus cellularity was measured on day 7 after SL-TBI. **j**, Female 1- to 2-month-old $Il18r1^{fl/fl}$:Ncr1-Cre⁻ ($Il18r1^{WT}$, $n = 6$) and $Il18r1^{fl/fl}$:Ncr1-Cre⁺ ($Il18r1^{ΔNK/ILC1}$, $n = 7$) mice were exposed to SL-TBI and administered rIL-18 (s.c., 2.5 mg kg⁻¹) on day 3. Thymus cellularity was measured on day 7 after SL-TBI. Graphs represent mean ± s.e.m.; each dot represents an individual biological replicate. Statistics were generated for **c** and **d** using one-way ANOVA with Dunnet's correction for multiple comparisons, for **e** using one-way ANOVA with Tukey's correction for multiple comparisons, and for **f–j** using unpaired two-tailed $t$ tests.

but notably only with the introduction of exogenous rIL-18 on day 3 following TBI (Fig. 4j and Extended Data Fig. 5f). The increased thymic NK cells at baseline in $Il18r1^{ΔNK/ILC1}$ mice offers a potential explanation for this discrepancy (Extended Data Fig. 5g). Nonetheless, these findings indicate that NK/ILC1 cells can mediate the IL-18 response and are the most likely effector cells.

## Acute thymic damage activates NK cells and induces a cytotoxic response

Given our data demonstrating that NK/ILC1 cells are the main targets of IL-18 following acute thymic damage, we sought to further characterize their function. Analysis of our scRNAseq dataset following cytoreductive conditioning revealed upregulation of genes

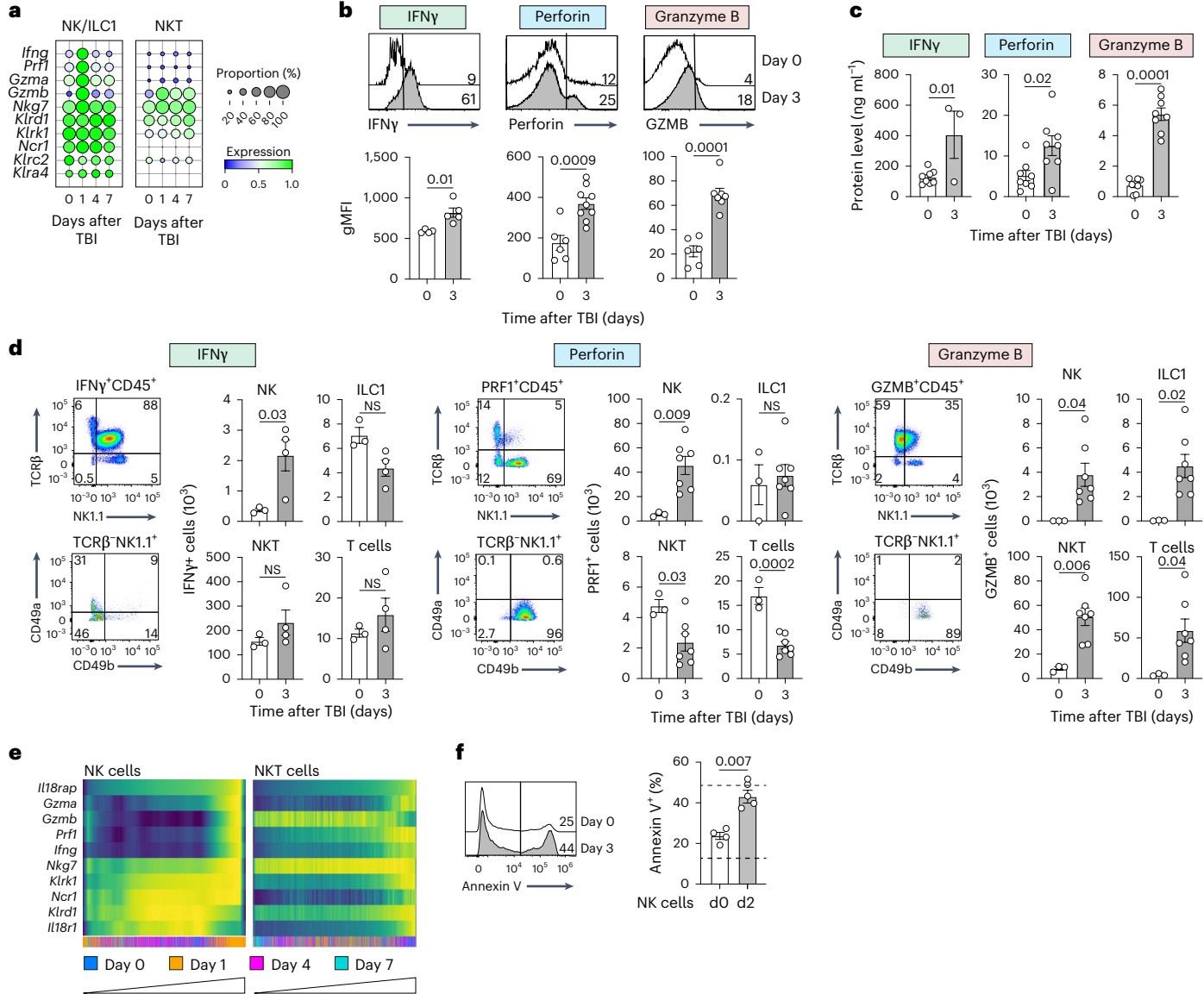

**Fig. 5 | Acute insult activates thymic NK cells. a**, Normalized gene expression in NK/ILC1 or NKT cells of the cytotoxicity factors *Ifng*, *Prf1*, *Gzma* and *Gzmb*, as well as the activation markers *Nkg7*, *Klrd1*, *Klrk1*, *Ncr1*, *Klrc2* and *Klra4*, on days 0, 1, 4 and 7 after SL-TBI, taken from the scRNAseq dataset described in Fig. 2a. **b**, Concatenated flow cytometry plots and corresponding geometric mean fluorescence intensity (gMFI) of CD45⁺NK1.1⁺TCRβ⁻ NK cell expression of *Ifng*-GFP (day 0, *n* = 4; day 3, *n* = 5), perforin (day 0, *n* = 6; day 3, *n* = 9) and granzyme B (GZMB; day 0, *n* = 4; day 3, *n* = 5) on days 0 and 3 following SL-TBI in female 1- to 2-month-old C57BL/6 WT or *Ifng*-reporter mice. **c**, Amount of thymic IFNγ (day 0, *n* = 8; day 3, *n* = 3), perforin (*n* = 8 per group) and granzyme B (*n* = 8 per group) measured by ELISA in female 1- to 2-month-old C57BL/6 mice on days 0 and 3 after SL-TBI. **d**, Thymuses were collected from female 1- to 2-month-old C57BL/6 mice on day 3 after SL-TBI. Concatenated flow cytometry plots gated on all CD45⁺ *Ifng*-GFP (left), CD45⁺perforin⁺ (middle) and CD45⁺GZMB⁺ (right) cells, as well as the total thymus cellularity of *Ifng*-GFP (left; day 0, *n* = 3; day 3, *n* = 4), perforin⁺ (middle; day 0, *n* = 3; day 3, *n* = 7) and GZMB⁺ (right; day 0, *n* = 3;

day 3, *n* = 7) CD45⁺NK1.1⁺TCRβ⁻CD49b⁺ NK cells, CD45⁺NK1.1⁺TCRβ⁻CD49a⁺ ILC1s, CD45⁺NK1.1⁺TCRβ⁺CD49b⁻ NKT cells and CD45⁺TCRβ⁺NK1.1⁻ T cells, are shown. **e**, Gene expression heat map of thymic NK/ILC1 and NKT cells for *Gzma*, *Gzmb*, *Prf1*, *Ifng*, *Nkg7*, *Klrk1*, *Ncr1*, *Klrd1* and *Il18r1* at 1, 4 and 7 days following SL-TBI. Each column represents a cell, with the cells ordered based on the expression of *Il18rap* (in ascending order from left to right). The time after TBI is indicated by the colors at the bottom. **f**, NK1.1⁺IL-18R⁺TCRβ⁻CD49b⁺ NK cells from female 1- to 2-month-old C57BL/6 mice were purified using FACS at baseline (d0) or 2 days after SL-TBI (d2) and cocultured with CellTrace-labeled RMA-S target cells at a 2:1 effector-to-target ratio. RMA-S target cell Annexin V expression was measured 5 h after coculture, and cell death was assessed (*n* = 4 biological replicates per group, representative of three independent experiments). Dashed lines represent RMA-S alone (bottom) or the positive control (top). Graphs represent mean ± s.e.m.; each dot represents an individual biological replicate. Statistics were generated for **b**–**d** and **f** using unpaired two-tailed *t* tests.

encoding NK/ILC1 effectors, including *Ifng*, *Prf1* and markers of NK cell activation (Fig. 5a). Comparison to other potential IL-18R⁺ targets, such as NKT cells, revealed no such program (Fig. 5a). Protein analysis of thymic T cell receptor (TCR)⁻NK1.1⁺ cells (which encompass both NK cells and ILC1s) 3 days following HCT conditioning supported this transcriptome analysis, showing increased expression of IFNγ,

granzyme B and perforin (Fig. 5b). Consistent with this, there were increases in the global thymic levels of IFNγ, granzyme B and perforin early after HCT conditioning (Fig. 5c). While the annotation of scRNAseq datasets was unable to distinguish ILC1s from NK cells based on the expression of the genes *Itga1* and *Itga2* (Extended Data Fig. 6a), we were able to differentiate these populations based on the expression

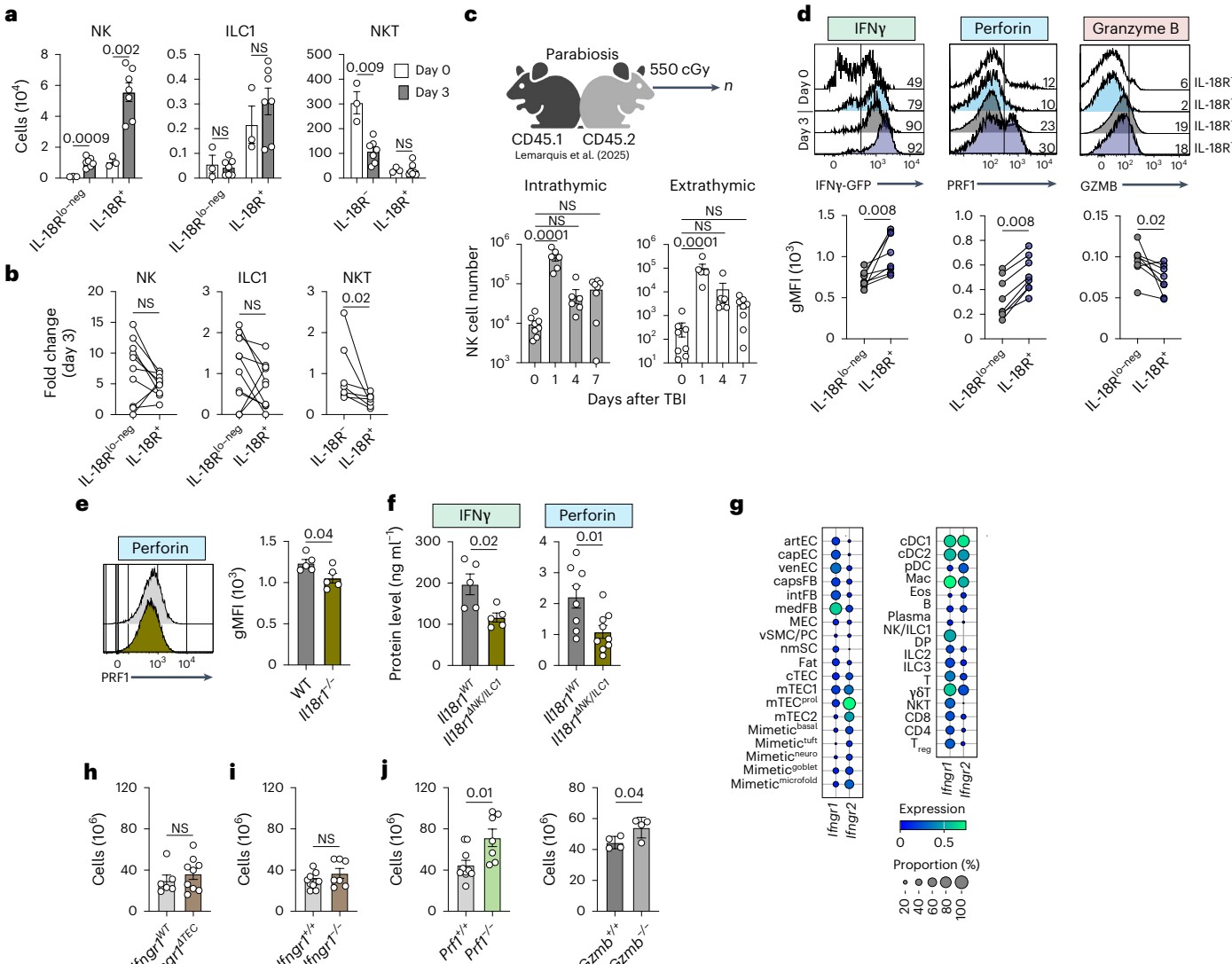

**Fig. 6 | IL-18 stimulation of cytotoxic NK cells suppresses thymus regeneration. a,b,** Female 1- to 2-month-old C57BL/6 WT or *Ifng*-reporter mice were exposed to SL-TBI, and thymuses were collected on days 0 and 3 after irradiation. IL-18R⁺ and IL-18R^lo−neg CD45⁺NK1.1⁺TCRβ⁻CD49b⁺ NK cells, CD45⁺NK1.1⁺TCRβ⁻CD49a⁺ ILC1s and CD45⁺NK1.1⁺TCRβ⁺CD49b⁻ NKT cells were compared. **a,** NK, ILC1 and NKT cellularity on day 0 (*n* = 3) and day 3 (*n* = 7) after SL-TBI. **b,** Fold change in NK cells (*n* = 10), ILC1s (*n* = 10) and NKT cells (*n* = 7) between days 0 and 3 after SL-TBI. **c,** Female C57BL/6 CD45.1⁺ and CD45.2⁺ mice were surgically conjoined to establish parabiotic pairs when both members of the pair were subjected to SL-TBI. Thymuses were collected, and chimerism was calculated on day 0 (*n* = 8), day 1 (*n* = 6), day 4 (*n* = 4) or day 7 (*n* = 8) after SL-TBI. Congenic markers (CD45.1 and CD45.2) were used to determine the mouse of origin (that is, cells expressing the same CD45 isoform as the mouse-pair thymus were classified as 'intrathymically' derived, while cells expressing the alternate isoform were considered 'extrathymically' derived). The numbers of intrathymic or extrathymic CD45⁺NK1.1⁺CD3⁻ NK/ILC1 cells at the indicated time points are shown graphically. **d,** Concatenated flow cytometry plots showing *Ifng*-GFP, perforin and granzyme B expression within CD45⁺NK1.1⁺TCRβ⁻CD49b⁺ NK cells. The gMFI of *Ifng*-GFP, perforin and granzyme B expression in IL-18R^lo−neg and

IL-18R⁺ NK cells on day 3 after SL-TBI (*n* = 8 per group) is shown. **e,** Female 1- to 2-month-old C57BL/6 WT or *Il18r1*⁻/⁻ mice were exposed to SL-TBI, and thymuses were isolated 3 days later. Concatenated flow cytometry plots and gMFI of perforin expression within CD45⁺NK1.1⁺TCRβ⁻CD49b⁺ NK cells are shown (*n* = 5 per group). **f,** Female 1- to 2-month-old *Il18r1^fl/fl:Ncr1*-Cre⁻ (*Il18r1^WT*, *n* = 8) and *Il18r1^fl/fl:Ncr1*-Cre⁺ (*Il18r1^ΔNK/ILC1*, *n* = 9) mice were exposed to SL-TBI, and the total thymic IFNγ and perforin levels were measured 3 days later. **g,** *Ifngr1* and *Ifngr2* expression at baseline, taken from the scRNAseq dataset described in Fig. 2a. **h,** Female 1- to 2-month-old C57BL/6 *Ifngr^fl/fl:Foxn1*-Cre⁻ (*Ifngr^WT*, *n* = 6) and *Ifngr^fl/fl: Foxn1*-Cre⁺ (*Ifngr^ΔTEC*, *n* = 9) mice were exposed to SL-TBI, and thymus cellularity was measured on day 7. **i,** Female 1- to 2-month-old C57BL/6 WT (*Ifngr1^+/+*, *n* = 10) or *Ifngr1*⁻/⁻ (*n* = 7) mice were exposed to SL-TBI, and thymus cellularity was measured on day 7. **j,** Female 1- to 2-month-old C57BL/6 WT (*Prf^+/+*, *n* = 10 and *Gzmb^+/+*, *n* = 5), *Prf*⁻/⁻ (*n* = 7) and *Gzmb*⁻/⁻ (*n* = 4) mice were exposed to SL-TBI, and thymus cellularity was measured on day 7. Graphs represent mean ± s.e.m.; each dot represents an individual biological replicate. Statistics were generated for **a**, **e**, **f** and **h–j** using unpaired two-tailed *t* tests, for **b** and **d** using paired two-tailed *t* tests, and for **c** using one-way ANOVA with Dunnet's correction for multiple comparisons. Panel **c** created with BioRender.com.

of their transcribed proteins, CD49a and CD49b, respectively (Extended Data Fig. 3)[38,39]. Unbiased analysis of perforin-, IFNγ- and granzyme B-expressing populations before and after damage revealed that only NK cells exhibited increased expression of perforin and IFNγ after damage (Fig. 5d). In fact, NK cells comprised the largest population of perforin-expressing cells within the thymus following damage (Fig. 5d).

Notably, there was an increase in all granzyme-expressing cells, including NK, ILC1, NKT and T cells (Fig. 5d). Consistent with these findings, we observed that, following cytoreductive conditioning, the expression of the activation and effector genes *Gzma*, *Gzmb*, *Pfr1* and *Ifng* in NK/ILC1 but not NKT cells correlated with the expression of the restricted coreceptor *Il18rap* shortly after acute damage (Fig. 5e).

To determine whether this increased NK-specific activation profile directly resulted in increased cytotoxicity, we cocultured RMA-S target cells with fluorescence-activated cell sorting (FACS)-purified IL-18R+ NK cells from the thymuses of mice that were either undamaged or had undergone SL-TBI 2 days prior. This approach revealed higher rates of RMA-S target cell death in cocultures with NK cells isolated from damaged thymuses (Fig. 5f). Taken together, these data suggest that acute damage, such as that caused by HCT conditioning, activates NK cells, increasing their cytotoxicity and capacity to kill nearby target cells.

### IL-18 mediates the NK cell effector program after acute damage

We next sought to characterize the IL-18-dependent characteristics of the main IL-18R-expressing populations. Only NK cells increased in number after damage (Fig. 6a,b). Both IL-18R$^{lo–neg}$ and IL-18R$^{hi}$ NK cells increased following damage, and mice deficient in IL-18R demonstrated similarly increased NK cell numbers on day 3 after SL-TBI (Extended Data Fig. 6b), suggesting a possible IL-18-independent mechanism for NK cell expansion. ILC1s were largely resistant to damage on day 3, with no difference observed between IL-18R$^{hi}$ and IL-18R$^{lo–neg}$ cells (Fig. 6a,b). In contrast, only IL-18R+ NKT cells were resistant to damage, as IL-18R– NKT cells were depleted (Fig. 6a,b).

Parabiosis experiments performed as part of another study[12] showed that, while almost all thymic NK cells were resident at baseline, there was an influx of newly recruited NK cells after damage (Fig. 6c and Extended Data Fig. 6c,d). The fold increase in extrathymic NK cell numbers was similar to that of intrathymic cell numbers, although intrathymic cells still accounted for the majority of absolute NK cells (Fig. 6c and Extended Data Fig. 6c,d). However, as we define extrathymic cells by their expression of the opposite congenic genotype, it is worth noting that this will undercount extrathymically derived NK cells by about half, as a similar number of host congenic marker-bearing NK cells presumably also enter from the circulation. This suggests that the increase in the number of NK cells reflects both an influx and local expansion. Consistent with its putative role in broadly activating effector functions in NK cells, the expression of IFNγ, perforin and NKG2D was significantly higher on day 3 in IL-18R+ than in IL-18R$^{lo–neg}$ NK cells (Fig. 6d and Extended Data Fig. 6e). Similarly, we found that the expression of perforin was reduced on day 3 in $Il18r1^{-/-}$ NK cells, and reduced levels of perforin and IFNγ in $Il18r1^{ΔNK/ILC1}$ thymuses were observed at the same time point (Fig. 6e,f). Notably, in contrast to this IL-18 dependence, we found that granzyme B expression was lower in IL-18R+ NK cells (Fig. 6d).

To functionally determine whether IL-18 directly induces the activation of thymic NK cells, we administered rIL-18 to WT mice that were not exposed to any damaging stimuli and assessed NK cell-derived cytotoxic factors 2 days later. Thymus size was unaffected 2 days after administration; however, exogenous rIL-18 administration increased the expression of the effector molecules perforin and IFNγ in NK cells. Functionally, these NK cells showed increased cytotoxicity in RMA-S cocultures (Extended Data Fig. 6f–i). Together, these findings suggest that IL-18 release following HCT conditioning increases the cytotoxicity of thymic NK cells.

Our scRNAseq dataset suggested widespread expression of IFNγ receptor (IFNγR)-encoding genes ($Ifngr$), including in TECs (Fig. 6g)—a pathway that has previously been implicated in mediating TEC cell death in acute graft-versus-host disease (GVHD) after HCT[18,40]. This led us to hypothesize that IL-18-mediated NK cell production of IFNγ results in IFNγ-induced TEC cell death. To address this, we generated mice with a TEC-specific deletion of $Ifngr1$; however, the absence of IFNγR in TECs ($Ifngr1^{ΔTEC}$) did not identify any difference in thymic regenerative capacity, suggesting that IFNγ does not signal TECs to dampen regeneration (Fig. 6h). To assess whether IFNγ could be affecting other cells, we examined regeneration in mice with a germline deletion of $Ifngr$ ($Ifngr^{-/-}$), which similarly did not alter regenerative

capacity (Fig. 6i). NK cell-mediated killing involves degranulation and the release of preformed cytotoxic proteins, mainly granzymes and perforin[41]. Mice lacking perforin ($Prf^{-/-}$) or granzyme B ($Gzmb^{-/-}$) showed significantly improved thymus regeneration compared to WT control mice (Fig. 6j). Therefore, cytotoxic granules released by NK cells following damage suppress thymus repair. However, while the absence of either perforin or granzyme B can improve regeneration, our data suggest that only NK cell perforin expression is regulated by IL-18 (Fig. 6d).

### Cytotoxic NK cells aberrantly target TECs

Having established that NK cell cytotoxicity suppresses thymus regeneration, we sought to identify populations that may be targeted by NK cells following cytoreductive conditioning. NK cells recognize stochastically expressed major histocompatibility complex class I (MHC-I) molecules in cells through Ly49 family inhibitory receptors[42]. NK cells can target self-cells in settings of MHC-I downregulation, such as virally infected cells or cancer cells evading CD8+ T cell immune surveillance[41–43]. Analysis of scRNAseq data identified decreased expression of MHC-I-encoding genes ($H2-D1$, $H2-K1$, $B2m$) that was almost exclusive to TEC subsets following SL-TBI (Fig. 7a). Accordingly, H-2K$^b$ expression was decreased within all TEC subsets (except tuft cells) but not in other stromal populations of endothelial, mesothelial or fibroblast cells following conditioning (Fig. 7b). H-2K$^b$ increased within CD45+ cells, driven by most thymocyte subsets and myeloid cells (Fig. 7b and Extended Data Fig. 7a). The expression of the NKG2D-activating ligand RAE-1 (ref. 42) was also upregulated in TECs after damage (Fig. 7c). Given the crucial function of TECs during normal T cell development as well as thymic regeneration[30], we hypothesized that TECs were targeted by IL-18-triggered NK cells following cytoreductive conditioning. Consistent with this hypothesis, there were significantly fewer viable cTECs and mTECs (isolated on day 3 after SL-TBI) when cocultured with activated NK cells (Fig. 7d). In contrast, the addition of NK cells had less effect on the viability of CD45– non-TEC cells (Extended Data Fig. 7b). Mice deficient in IL-18R had similar TEC numbers and viability at baseline compared to WT controls; however, there was significantly increased viability in both cTECs and mTECs following damage, which translated into increased cell numbers early after damage (Fig. 7e,f and Extended Data Fig. 7c). Similar to in vitro assays, we found no change in CD45– non-TEC viability between WT and $Il18r1^{-/-}$ mice (Extended Data Fig. 7d). Consistent with the hypothesis that activated NK cells mediate the killing of TECs after damage, nearest-neighbor analysis of NKp46+ cells in imaging studies (Fig. 4e) revealed a significant decrease in the distance between NKp46+ cells and both the cortical and medullary epithelium (Fig. 7g). Together, these findings suggest that MHC-I downregulation and RAE-1 upregulation in TECs make these cells vulnerable to activated NK cells throughout the organ.

We identified that, following cytoreductive conditioning, increased cl-Cas-1 expression leads to the release of activated IL-18, which triggers the cytotoxicity of organ-resident NK cells. These NK cells target both cTECs and mTECs to suppress thymus recovery and T cell reconstitution. Furthermore, these data demonstrate that IL-18 abrogation holds promise as a therapeutically feasible strategy for improving thymus recovery following HCT.

## Discussion

The thymus is extremely sensitive to acute injury, particularly during pre-HCT cytoreductive conditioning. In this study, we identified a crucial role for damage-induced IL-18 in limiting thymic regeneration through the stimulation of NK cells, which target the TEC stromal population.

IL-18 has been shown to regulate intestinal barrier function through epithelial cell maturation and function[44–46]. Given the existing parallels between the intestinal epithelium and TECs, as well as the importance of TECs for thymic function and repair, we first explored

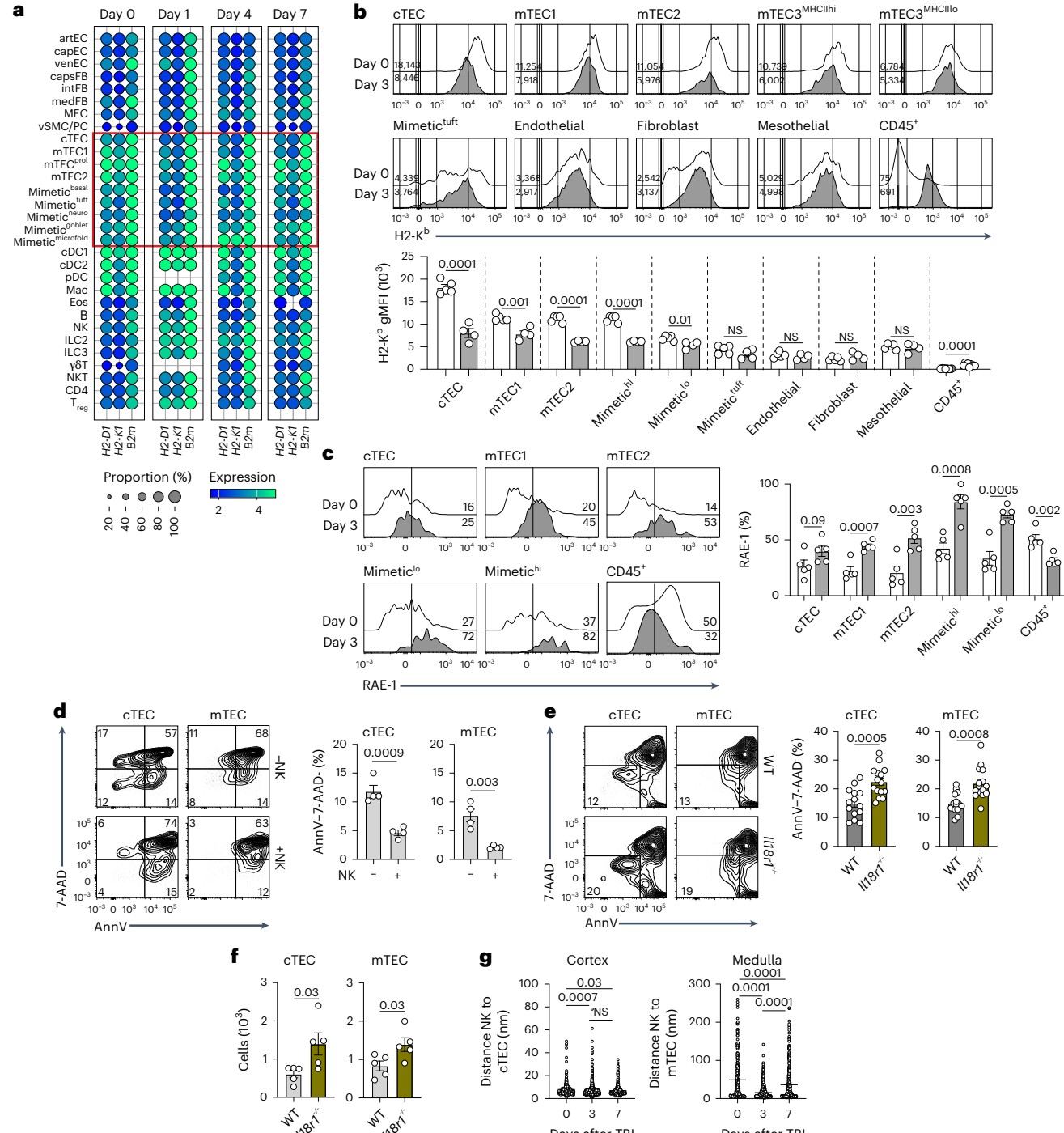

**Fig. 7 | Cytotoxic NK cells aberrantly target TECs. a,** Normalized expression of the MHC-I genes *H2-D1, H2-K1* and *B2m* following SL-TBI, taken from the scRNAseq dataset described in Fig. 2a. Red box highlights epithelial populations. **b,** Thymuses from female 1- to 2-month-old C57BL/6 mice were collected at baseline (*n* = 4) or 3 days after SL-TBI (*n* = 5). Concatenated flow cytometry plots showing H-2K^b expression in stromal subsets (gating and phenotypes are provided in Extended Data Fig. 3) and the CD45^+ population (*n* = 10) are presented. **c,** Thymuses from female 1- to 2-month-old C57BL/6 mice were collected at baseline or 3 days after SL-TBI. Concatenated flow cytometry plots and quantification of RAE-1 expression in stromal subsets (*n* = 5 per group) are shown. **d,** Female 1- to 2-month-old C57BL/6 mice were exposed to SL-TBI, and thymuses were collected 3 days later and enriched for nonhematopoietic stromal cells, which were cultured with or without poly(I:C)-stimulated NK cells. The expression of Annexin V (AnnV) and 7-aminoactinomycin D (7-AAD) in CD45^−EpCAM^+MHC-II^+Ly51^+ cTECs and CD45^−EpCAM^+MHC-II^+UEA-1^+ mTECs

was measured 5 h after coculture (*n* = 4 per group). **e,** Female 1- to 2-month-old C57BL/6 WT (*n* = 14) or *Il18r1*^−/−^ (*n* = 15) mice were exposed to SL-TBI. The expression of Annexin V and 7-AAD in CD45^−EpCAM^+MHC-II^+Ly51^+ cTECs, CD45^−EpCAM^+MHC-II^+UEA-1^+ mTECs was measured 5 days later. **f,** Female 1- to 2-month-old C57BL/6 WT or *Il18r1*^−/−^ mice were exposed to SL-TBI, and CD45^−EpCAM^+MHC-II^+Ly51^+ cTEC and CD45^−EpCAM^+MHC-II^+UEA-1^+ mTEC cellularity was measured 3 days later (*n* = 5 biological replicates per group, representative of two independent experiments). **g,** Data extrapolated from the images in Fig. 4e. The distance between NKp46^+ cells and either keratin-14^+ mTECs or keratin-8^+ cTECs was estimated by nearest-neighbor analysis and shown as a waterfall plot (day 0, *n* = 278; day 3, *n* = 1,663; day 7, *n* = 426). Graphs represent mean ± s.e.m.; each dot represents an individual biological replicate. Statistics were generated for **b–f** using unpaired two-tailed *t* tests and for **g** using one-way ANOVA with Tukey's correction for multiple comparisons.

the possibility that IL-18 signaling through TECs may directly inhibit thymic function. However, the conditional deletion of *Il18r1* within *Foxn1*-expressing TECs did not affect thymus recovery. As IL-18 has been reported to be a regulator of hematopoietic stem cell quiescence[32,33], we also performed competitive transplantations that demonstrated no differences in the capacity of *Il18r1*-deficient hematopoietic stem and progenitor cells to reseed the recovering thymus or reconstitute overall T cell populations longitudinally. Based on these findings, we conclude that IL-18 does not directly regulate thymus regeneration through TEC or thymocyte progenitor signaling.

IL-18 canonically signals through T, NKT, ILC1 and NK cells to mediate a T helper 1 response, primarily by inducing IFNγ expression[19,24]. We found that thymic IL-18R expression was largely restricted to NK1.1⁺ NKT, NK and ILC1 cells, and that the depletion of these populations improved thymus recovery. In contrast to studies implicating NKT cells in regeneration[10], we observed that mice deficient in NKT cells did not show any alterations in thymic repair, possibly due to differences in the background strain. Importantly, we found that the specific deletion of *Il18r1* in NK/ILC1 cells could improve thymus regeneration. Within the thymus, exogenous IL-18 and IL-12 act synergistically to promote the expansion and extravasation of ILC1s[47]. Notably, recent work found that IL-18 can stimulate the production of ILC1s to secrete GM-CSF, which skews thymic hematopoietic precursors toward granulopoiesis[47,48]. However, we found that only NK cells responded to injury by expanding and upregulating IFNγ and perforin—both in an IL-18-dependent manner—and that highly purified NK cells exhibited increased cytotoxicity. Despite previous studies linking IFNγ with TEC cell death during GVHD[40], surprisingly, IFNγ did not limit thymic regeneration. Instead, deficiency in either perforin or granzyme B was sufficient to improve thymic repair; however, our findings also suggest that only perforin is central to the NK/IL-18 axis, supporting prior work implicating IL-18 in the regulation of perforin- but not granzyme-dependent mechanisms of NK cell cytotoxicity[49]. These data demonstrate that cl-Cas-1-dependent IL-18 release suppresses thymus regeneration, supported by previous work showing that mice deficient in the NLRP3 inflammasome, which is upstream of Cas-1, exhibit improved thymus function[50].

Donor NK cells are reportedly beneficial in the setting of HCT, promoting engraftment, reducing GVHD by targeting HLA-mismatched antigen-presenting cells, and increasing TEC proliferation[51,52]. However, we suggest a different mechanism whereby radioresistant recipient NK cells are activated as a by-product of pyroptosis-triggered regenerative responses, counteracting thymus regeneration. Therefore, there is likely a distinction between the proreparative and antireparative functions of donor and recipient NK cells, respectively. Ionizing radiation induces the expression of MHC in tissues such as the intestine, largely through the upregulation of IFNγ[53,54]. However, although we observed increased IFNγ expression after TBI and increased MHC-I expression in CD45⁺ cells, TECs showed decreased MHC-I expression and increased levels of the NKG2D ligand RAE-1, making them vulnerable to cytotoxicity in an HLA-mismatch-independent manner. Therefore, there is a distinction between the proreparative and antireparative roles of donor and recipient NK cells, respectively. Furthermore, our work is consistent with reports showing that IL-18-stimulated NK cells target the epithelium during viral infection, delaying reepithelization, and that NK cells also target hematopoietic stem cells that upregulate NKG2D ligands in response to genotoxic stress[55–58]. Our study, therefore, contributes to a growing body of literature that reveals a role for NK cells in the regulation of tissue injury and repair. However, given the downregulation of MHC-I in TECs specifically, an interesting possibility exists that this IL-18/NK/MHC-I axis is an evolutionarily conserved mechanism to eliminate TECs that have undergone genotoxic stress in an effort to prioritize the quality of T cell selection over quantity.

This work positions IL-18 as a potential therapeutic target for improving thymus function after exposures causing acute injury, such as HCT conditioning. However, given the reported context-dependent effects of IL-18 in GVHD, along with its emerging promise in immunotherapy, careful examination of its abrogation will be required to balance its proreparative and graft-versus-tumor effects[19,59–64]. In summary, this study identifies a new pathway regulating T cell immune reconstitution following acute thymus damage and presents multiple opportunities for potential therapeutic targets to improve T cell reconstitution not only in patients undergoing HCT but also in those exposed to other forms of acute thymus injury due to chemotherapy, stress and infection.

## Online content

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

## Methods

### Mice

Inbred male and female C57BL/6J (000664) and B6 CD45.1 (002014) mice were obtained from The Jackson Laboratory. $Il1r1^{-/-}$ (003245), $Il18^{-/-}$ (004130), $Il18r1^{-/-}$ (004131), $Cas1^{\Delta10}$ (032662), $Cd1d^{-/-}$ (008881), $Ifngr^{-/-}$ (003288), $Prf1^{-/-}$ (002407), $Rag2$-eGFP (005688) and GREAT (IFNγ reporter with endogenous poly(A) transcript) mice were obtained from The Jackson Laboratory and bred in-house. $Gzmb^{-/-}$ mice were obtained from G. Hill (Fred Hutchinson Cancer Center) and bred in-house. $Il18$ flox mice ($Il18^{fl/fl}$) were obtained from R. Nowarski (Harvard Medical School) and R. Flavell (Yale School of Medicine) and crossed in-house with $Zbtb46$-Cre$^+$ mice obtained from The Jackson Laboratory (032662) to generate $Il18^{fl/fl}$:$Zbtb$-Cre$^+$ ($Il18^{\Delta cDC}$) mice. $Il18r1^{fl/fl}$ mice were obtained from G. Trinchieri (National Cancer Institute) and crossed with $Foxn1$-Cre$^+$ mice obtained from The Jackson Laboratory (018448) and $Ncr1$-Cre$^+$ mice obtained from K. Barry (Fred Hutchinson Cancer Center) to generate $Il18r1^{fl/fl}$:$Foxn1$-Cre$^+$ ($Il18r1^{\Delta TEC}$) and $Il18r1^{fl/fl}$:$Ncr1$-Cre$^+$ ($Il18r1^{\Delta NK}$) mice, respectively. $Ifngr^{fl/fl}$ (025394) and $Foxn1$-Cre$^+$ (018448) mice were obtained from The Jackson Laboratory and were crossed to generate $Ifngr^{fl/fl}$:$Foxn1$-Cre$^+$ ($Ifngr^{\Delta TEC}$) mice. All experimental mice were between 6 and 10 weeks old. Mice were maintained at the Fred Hutchinson Cancer Research Center and acclimatized for at least 2 days before experimentation, which was performed according to the Institutional Animal Care and Use Committee guidelines.

### Cell isolation

Single-cell suspensions of freshly dissected thymuses were obtained and enzymatically digested using 0.15% collagenase D (Sigma 11088882001) and 0.1% DNase I (Sigma 10104159001) in DMEM, as previously described[8]. Cellularity was calculated using the Z2 Coulter Particle and Size Analyzer (Beckman Coulter). For studies sorting rare populations of cells in the thymus, multiple identically treated thymuses were pooled to isolate sufficient numbers of cells; however, in these instances, separate pools of cells were established to maintain individual samples as biological replicates. The bone marrow was flushed from the femurs and tibias and then passed through a 70-μm filter. Peripheral blood samples were collected into EDTA capillary pipettes (Fisher Scientific). Red blood cell lysis was performed using ACK lysis buffer (A1049201, Fisher Scientific).

### Flow cytometry

Cells were stained with antibodies to the following proteins for analysis: CD45 (565967, BD Biosciences), CD31 (102434, BioLegend), CD140a (135907, BioLegend), MHC-II (107620, BioLegend), EpCAM (46-5791-82, BD Biosciences), Ly51 (740882, BD Biosciences), UEA-1 (ZC0426, Vector Laboratories), DLCK-1 (NBP1-77127F, Novus Biologicals), Ly6D (138605, BioLegend), CD104 (123615, BioLegend), CD140a (135921, BioLegend), CD31 (102427, BioLegend), PDPN (127425, BioLegend), CD8a (100714, BioLegend), CD4 (565709, BD Biosciences), TCRβ (109239, BioLegend), CD3ε (100232, BioLegend), CD25 (102030, BioLegend), CD44 (612799, BD Biosciences), NK1.1 (108753, BioLegend), CD49b (561067, BD Biosciences), c-Kit (105811, BioLegend), TCRγδ (118107, BioLegend), CD1d PBS-57 tetramer (National Institutes of Health Tetramer Core), CD11c (35-0114, Tonbo), CD11c (612796, BD Biosciences), CD11b (741722, BD Biosciences), XCR1 (148225, BioLegend), B220 (103232, BioLegend), CD127 (50-1271, Tonbo), Sca-1 (122527, BioLegend), CD135 (135305, BioLegend), CD150 (46-1502-82, eBioscience), CD48 (103427, BioLegend), NKG2D (562800, BD Biosciences), CD49a (741976, BD Biosciences), KLRG1 (138425, BioLegend), CCR-6 (129814, BioLegend), IL-23R (150907, BioLegend), ST2 (566310, BD Biosciences), H-2K$^b$ (116525, BioLegend), IL-18R (25-5183-82, Thermo Fisher), IL-18R (25-5183-82, Thermo Fisher), RAE-1 (130-111-467, Miltenyi Biotec) and streptavidin-APC high concentration (405243, BioLegend). Following fixation and permeabilization (554714, BD Biosciences), cells were stained with antibodies to perforin

(154315, BioLegend) and granzyme B (MHGB04, Thermo Fisher). Annexin V and 7-AAD staining (640920, BioLegend) was performed in Annexin V binding buffer (422201, BioLegend). Flow cytometric analysis was performed on a Symphony S6 instrument (BD Biosciences), and cells were sorted on an Aria II cell sorter (BD Biosciences) using FACS-Diva (BD Biosciences) or FlowJo (TreeStar) software.

### In vivo acute damage models

To induce thymus damage, we subjected mice to SL-TBI at a dose of 550 cGy from a Cs-137 γ-radiation source without hematopoietic rescue. Other models of thymus damage included i.p. injection of 20 mg kg$^{-1}$ dexamethasone (Sigma-Aldrich D2915), 200 mg kg$^{-1}$ cyclophosphamide (University of Washington Medical Pharmacy) and 1.5 mg kg$^{-1}$ LPS (InvivoGen tlrl-eblps). For in vivo studies of rIL-18 administration, C57BL/6J or $Ifng$-GFP mice were administered 2.5 mg kg$^{-1}$ rIL-18 (s.c.) either in the absence of other thymus-damaging treatments (day 0) or at 3 days after SL-TBI.

### In vivo depletion and transplantation studies

To perform NK1.1$^+$ cell depletion studies, we injected mice with 200 μg (i.p., 10 mg kg$^{-1}$) of anti-NK1.1 mAb (BioXCell BE0036) on days −1, 1 and 3 following SL-TBI. B6 HCT recipients were subjected to 1,100 cGy TBI (2 × 550 cGy) before transplantation; then, within 24 h, they received an i.v. injection of $5 \times 10^6$ to $10 \times 10^6$ bone marrow cells. For IL-18 abrogation experiments, mice were dosed with 200 μg (i.p., 10 mg kg$^{-1}$) of anti-IL-18 mAb (BioXCell BE0237) on days −1, 1, 3, 6, 9, 12, 15 and 18 following transplantation.

### Parabiosis

Female C57BL/6 CD45.1$^+$ and CD45.2$^+$ mice were surgically conjoined to establish parabiotic pairs using a modified protocol as previously described[12]. Briefly, mice were cohoused for 10 days before parabiosis surgery and maintained in a parabiotic state until both members of the pair were subjected to SL-TBI (550 cGy) on experimental day 21, 24 or 27, corresponding to 7, 4 or 1 day(s) before tissue collection. All mice were killed on day 28 after surgery. To distinguish between circulating and tissue-resident cells at the time of collection, we administered 3 μg of anti-CD45 antibody (APC-EF780, BioLegend) into the mice by retro-orbital injection 3 min before killing. Thymuses were collected from both parabionts and analyzed by flow cytometry. Circulating cells labeled by i.v. administration of anti-CD45 antibody were excluded from analysis. Congenic markers (CD45.1 and CD45.2) were used to determine the origin of thymic cells: cells expressing the same CD45 isoform as the assessed mouse-pair thymus donor were classified as intrathymically derived, whereas cells expressing the alternate isoform were considered extrathymically derived.

### Protein quantification

For the detection of active IL-1β, active IL-18, granzyme B, perforin, IFNγ and mature Cas-1 (Figs. 1b,c and 5c and Extended Data Fig. 1) in supernatants, thymic tissue was mechanically dissociated in defined volumes of buffer. The resulting supernatant was analyzed using cytokine-specific ELISA kits (IL-1β, Invitrogen #88-7013-22; IL-18, Thermo Fisher #BMS618-3; granzyme B, R&D #DY1865; perforin, Novus Biologicals #NBP3-00452; IFNγ, Thermo Fisher #KMC4021; mature Cas-1, Adipogen #AG-45B-0002-KI01), and absorbance was measured on a Spark 10M plate reader (Tecan).

For the detection of active IL-18 and IL-18BP (Fig. 1b: only IL-18 after cyclophosphamide treatment; Figs. 1d and 2d,e) in whole organs, thymuses were homogenized in RIPA buffer (25 mM Tris (pH 7.6), 150 nM NaCl, 1% NaCl, 1% NP-40, 0.1% SDS, 0.05% sodium deoxycholate, 0.5 mM EDTA) with protease inhibitors (Thermo Fisher A32955) using a Homogenizer 150 (Fisher Scientific) and normalized by mass at a concentration of 20 mg thymus tissue per ml of RIPA buffer. The resulting lysates were analyzed using cytokine-specific ELISA kits (IL-18, Thermo Fisher

#BMS618-3; IL-18BP, Abcam ab254509), and absorbance was measured on the Spark 10M plate reader (Tecan).

## In vitro cell culture

Coculture experiments were performed by plating 50,000 ex vivo FACS-purified bone marrow Lin$^-$ selected or Lin$^-$Sca-1$^+$c-Kit$^+$ FACS-purified cells onto six-well plates confluent with OP9-DLL1 cells in OP9 medium, as previously described[17,65]. Cocultures were performed in the presence of 5 ng ml$^{-1}$ Flt-3L (Peprotech, 250-31L) and 1 ng ml$^{-1}$ IL-7 (Peprotech 217-17), along with either 0, 1 or 10 ng ml$^{-1}$ rIL-18 (BioLegend 767008). Equal volumes of nonadherent cells were assessed by flow cytometry for differentiation at 10, 14 and 21 days following coculture.

## Cytotoxicity assays

Cytotoxicity assays of thymic NK cells (Figs. 5f and 6h) were performed by coculturing thymus-derived FACS-purified NK1.1$^+$ IL-18R$^+$CD49b$^+$TCRβ$^-$ NK cells with CellTrace Yellow (Thermo Fisher C34573)-labeled RMA-S cells at either a 2:1 or 5:1 effector-to-target ratio in RPMI/10% FBS supplemented with 10 ng ml$^{-1}$ rIL-15 (BioLegend 566302). Splenic FACS-purified NK1.1$^+$IL-18R$^+$CD49b$^+$TCRβ$^-$ NK cells derived from C57BL/6 mice treated with 0.3 mg poly(I:C) (i.p.) (InvivoGen tlrl-picw) 1 day earlier served as positive controls. Cocultures were incubated at 37 °C for 5 h, after which cell death of CellTrace Yellow-labeled RMA-S cells was assessed by flow cytometry according to the expression of Annexin V (BioLegend 640920).

Cytotoxicity assays of thymus-derived TECs as target cells (Fig. 7c) were performed by magnetically enriching thymus-derived CD45$^-$ cells and coculturing them with CellTrace Violet (Thermo Fisher C34571)-labeled splenic NK-enriched cells (Miltenyi Biotec 130-115-818) derived from C57BL/6 mice treated with 0.3 mg poly(I:C) (i.p.) (InvivoGen tlrl-picw) 1 day earlier at a 4:1 effector-to-target ratio in RPMI/10% FBS supplemented with 10 ng ml$^{-1}$ rIL-15 (BioLegend 566302). Cocultures were incubated at 37 °C for 5 h, after which cell death of CD45$^-$EpCAM$^+$MHC-II$^+$Ly51$^+$ cTECs and CD45$^-$EpCAM$^+$MHC-II$^+$UEA-1$^+$ mTECs was assessed by flow cytometry based on the expression of Annexin V and 7-AAD (BioLegend 640920).

## Imaging

Formalin-fixed, paraffin-embedded tissues were cut into 4-μm sections, mounted onto positively charged slides and baked for 1 h at 60 °C. The slides were then dewaxed and stained on the BOND RX stainer (Leica) using Leica BOND reagents for dewaxing (Dewax Solution), antigen retrieval and antibody stripping (Epitope Retrieval Solution 2), and rinsing after each step (BOND Wash Solution). The antigen retrieval and antibody stripping steps were performed at 100 °C, while all other steps were conducted at ambient temperature. Endogenous peroxidase was blocked with 3% $H_2O_2$, followed by protein blocking with TCT buffer (0.05 M Tris, 0.15 M NaCl, 0.25% casein, 0.1% Tween-20, 0.05% ProClin 300, pH 7.6). Primary antibodies (rabbit polyclonal anti-mouse keratin-14, BioLegend 905301; rat anti-mouse keratin-8, Troma-I Developmental Studies Hybridoma Bank; rabbit anti-mouse NKp46, Abcam 233558) were applied sequentially, followed by the application of the secondary antibodies and the tertiary TSA amplification reagent (Akoya OPAL fluorophore). A high-stringency wash was performed after the secondary and tertiary antibody applications using high-salt TBST solution (0.05 M Tris, 0.3 M NaCl and 0.1% Tween-20, pH 7.2–7.6). Species-specific polymer HRP was used for all secondary antibody applications, including either anti-rabbit HRP (Akoya Opal) or goat anti-rat IgG polymer detection kit (Vector ImmPress). Following the application of the final antibody, the slides were stained with DAPI and coverslipped with Prolong Gold Antifade reagent (Invitrogen/Life Technologies). Slides were cured at room temperature, and whole-slide images were acquired on the Vectra Polaris Quantitative Pathology Imaging System (Akoya Biosciences), spectrally unmixed using Phenoptics Inform software and exported as multi-image TIF files. Tiles were fused, and cellular analysis of the images was performed using the HALO image analysis software (Indica Labs). The cells were first identified based on nuclear recognition of the DAPI stain, and membrane segmentation was assisted by referencing the two cytokeratin stains. Thresholds were set to identify positive cells based on the mean intensity within the cytoplasmic and membrane regions of each cell. Cortical and medullary regions were defined by a random forest classifier, followed by a manual review. Cell populations were quantified within each region, and a nearest-neighbor analysis was performed to determine spatial relationships and provide measurements between cells.

## scRNAseq and qPCR

Previously generated and published scRNAseq datasets of thymic CD45$^-$ nonhematopoietic cells (GSE240016; 50,890 cells) and $Rag2^{GFP}$ CD45$^+$ hematopoietic cells (GSE244673; 37,879 cells) from 2-month-old mice at steady state and on days 1, 4 and 7 after SL-TBI were used for this study[12,28]. The CD45$^-$ dataset can be viewed at https://thymosight.org/, along with all previously published thymus single-cell sequencing datasets. CellChat (v1.4.0)[66] was used with default parameters to predict cell–cell interactions between all subsets using the combined dataset at steady state and on days 1, 4 and 7 after damage, focusing on the IL-18 signaling pathway. Aggregate signal strength was calculated for each IL-18 target cell by combining the CellChat signal quantification for each IL-18 source to an individual target. RNA was extracted from sorted cells using the RNeasy Plus Micro kit (74034, Qiagen). cDNA was synthesized using the iScript gDNA Clear cDNA Synthesis kit (1725035, Bio-Rad) and a Bio-Rad C1000 Touch ThermoCycler (Bio-Rad). RNA expression was assessed using the Bio-Rad CFX96 Real Time System (Bio-Rad), with iTaq Universal SYBR Green Supermix (1725122, Bio-Rad) and the *Il18* primer (qMmuCED0061252, Bio-Rad).

## Statistics

Statistical analysis between two groups was performed using an unpaired two-tailed *t* test (Figs. 1c,g–i, 3d,h,j–k, 4f–j, 5b–d,f, 6a,e,f,h–j and 7b–f and Extended Data Figs. 1c, 5e,f, 6b,d,f–i and 7a–d) or a paired two-tailed *t* test (Fig. 6b,d and Extended Data Fig. 6e). Statistical comparisons among three or more groups in the figures were performed using a one-way ANOVA with Dunnett's multiple comparison test (Figs. 1a,b,d–f and 6c and Extended Data Figs. 1a and 6c) or Tukey's multiple comparison test (Figs. 2d,e, 4c–d and 7g and Extended Data Fig. 1b). All statistics were calculated using GraphPad Prism, and display graphs were generated in either GraphPad Prism or R. Information on replicates, error bars and statistical significance can be found in the figures and their corresponding legends.

## Reporting summary

Further information on research design is available in the Nature Portfolio Reporting Summary linked to this article.

# Data availability

The datasets generated and/or analyzed during the current study are provided with this article. Sequencing data used in this study have been deposited in the National Center for Biotechnology Information's Gene Expression Omnibus (GEO) and can be accessed through GEO numbers GSE240016 (CD45$^-$ nonhematopoietic cells) and GSE244673 (CD45$^+$ hematopoietic cells). Any additional data are available from the corresponding author. Source data are provided with this paper.

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

## Acknowledgements

We gratefully acknowledge the assistance of the Fred Hutchinson Flow Cytometry Core (RRID:SCR_022613) and the Experimental Histopathology (RRID:SCR_022612) Shared Resource of the Fred Hutchinson/University of Washington Cancer Consortium (P30-CA015704). We gratefully acknowledge the support of the Immunotherapy Integrated Research Center at the Fred Hutchinson Cancer Research Center. We are grateful to G. Trinchieri (National Institutes of Health), R. Flavell (Yale University), R. Nowarski (Harvard University) and K. Barry (Fred Hutchinson) for providing mice, as well as to G. Hill (Fred Hutchinson) and J. Trapani (Peter Mac Cancer Center, Australia) for helpful discussions. This research was supported by the National Institutes of Health award numbers R35-HL171556, R01-HL145276 (J.A.D.), R01-HL165673 (J.A.D.), U01-AI70035 (J.A.D.), Project 1 of P01-AG052359 (J.A.D.), and the National Cancer Institute Cancer Center Support Grant (P30-CA015704). D.G. was supported by F30-HL165761. D.A. was supported by T32-GM007270.

## Author contributions

D.G. and J.A.D. conceived of the idea for this article. D.G. and J.A.D. designed, analyzed and interpreted experiments, as well as drafted the manuscript. K.C., D.A., M.W., V.H., L.I., P.d.R., E.E.L., S.S.-S., S.K. and C.E. performed experiments and/or analyzed and helped interpret data. A.K., A.L. and M.R.M.v.d.B. aided in project discussions and sequencing data analysis. J.A.D. supervised the project.

## Competing interests

J.A.D. and M.R.M.v.d.B. are founders of and receive stock options from Thymofox and Thymogenesis, and both receive royalties from Wolters Kluwer. The other authors declare no competing interests.

## Additional information

**Extended data** is available for this paper at https://doi.org/10.1038/s41590-025-02270-z.

**Correspondence and requests for materials** should be addressed to David Granadier or Jarrod A. Dudakov.

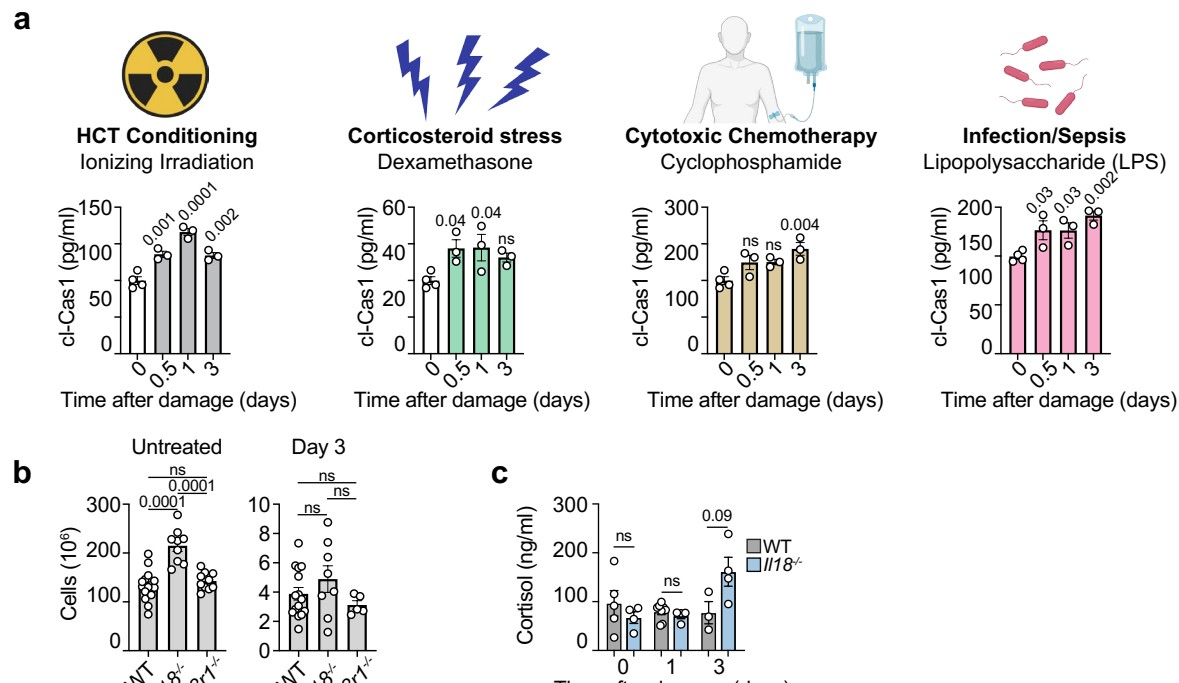

**Extended Data Fig. 1 | Increased pyroptosis after distinct modalities of thymic damage. a**, Thymus supernatants generated from female 1-2 mo C57BL/6 mice given SL-TBI (550 cGy), Dexamethasone (i.p., 20 mg/kg), Cyclophosphamide (i.p., 200 mg/kg) or LPS (i.p., 1.5 mg/kg) and mature Caspase-1 assayed by ELISA on days 0 (n = 4), 0.5 (n = 3), 1 (n = 3) and 3 (n = 3). **b**, Female 1-2 mo C57BL/6 WT (d0, n = 18; day 3, n = 15) or $Il18^{-/-}$ (d0, n = 9; day 3, n = 8) and $Il18r1^{-/-}$ (d0, n = 10; d3, n = 5) mice were given SL-TBI and thymus cellularity was assessed at baseline (day 0) or 3 days post-SL-TBI. **c**, Thymus supernatants generated from female 1-2 mo C57BL/6 WT (d0, n = 5; d1, n = 8; d3, n = 3) and $Il18^{-/-}$ (n = 4/group) mice given SL-TBI (550 cGy) and Cortisol assayed by ELISA on days 0, 1 and 3. Graphs represent mean ± SEM; each dot represents an individual biological replicate; ns = not significant. Statistics were generated for a using one-way ANOVA with Dunnet's correction for multiple comparisons, b using Tukey's correction for multiple comparisons, and for **c** using an unpaired two-tailed t-test. Panel **a** icons created with BioRender.com.

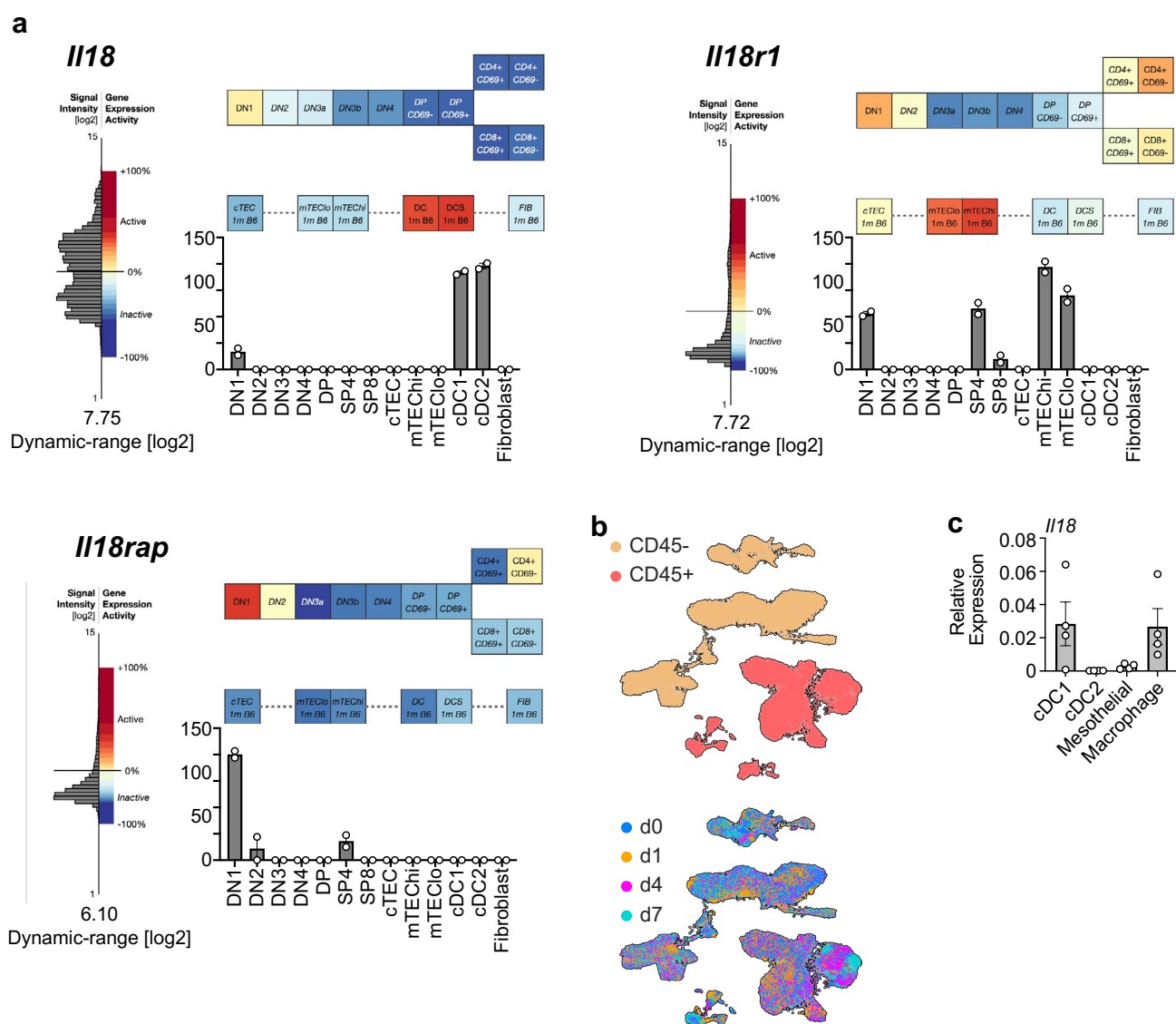

**Extended Data Fig. 2 | Expression of IL-18 related genes across subsets in the thymus. a**, Relative expression of *Il18*, *Il18r1* and *Il18rap* on subsets of thymus populations at baseline in 1mo mice (n = 2). Data extracted from Gene Expression Commons using the "Complete thymocyte:stromal interaction model dataset" (https://gexc.riken.jp/models/475/) which is based on GSE56928 (ref. 26). **b**, UMAPs from scRNAseq described in Fig. 2a with color overlays indicating

dataset (Left, *Rag2*-GFP⁻ CD45⁺ or CD45-) or timepoint (right, d0, 1, 4, 7 after SL-TBI). **c**, *Il18* relative expression measured by qPCR within cDC1, cDC2, MECs, and Macrophages FACS isolated (see Extended Data Fig. 3 for gating) from thymuses of female 1-2 mo C57/BL6 WT mice (n = 4/group). Graphs represent mean ± SEM; each dot represents an individual biological replicate.

### Thymic Epithelial Cells

Gated: CD45⁻EpCAM⁺
MHCII⁺UEA1ʰⁱLy51ˡᵒ

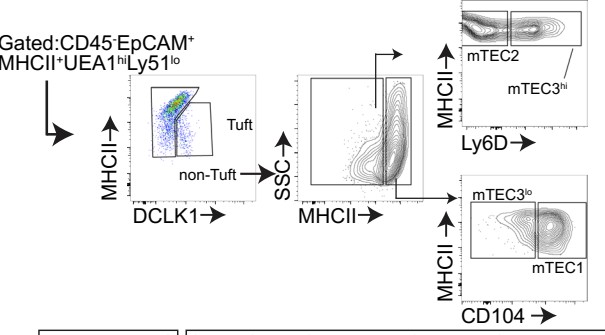

| cTEC | CD45⁻CD31⁻CD140a⁻EpCAM⁺MHC-II⁺Ly51⁺UEA-1ˡᵒ |
|---|---|
| mTEC | CD45⁻CD31⁻CD140a⁻EpCAM⁺MHC-II⁺Ly51⁻UEA-1ʰⁱ |
| mTEC1 | ∟mTEC gated: DCLK1⁻MHCIIˡᵒCD104⁺ |
| mTEC2 | ∟mTEC gated: DCLK1⁻MHCIIʰⁱLy6D⁻ |
| mTEC3ʰⁱ | ∟mTEC gated: DCLK1⁻MHCIIʰⁱLy6D⁺ |
| mTEC3ˡᵒ | ∟mTEC gated: DCLK1⁻MHCIIˡᵒCD104⁻ |
| Tuft | ∟mTEC gated: DCLK1⁺ |

### Non-Epithelial Stroma

Gated: Viable⁺CD45⁻EpCAM⁻MHCII⁻

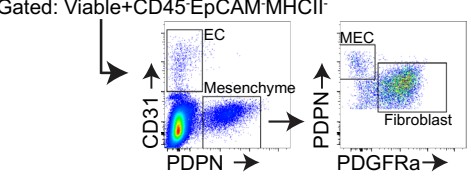

| Endothelial | CD45⁻EpCAM⁻MHC-II⁻CD31⁺CD140a⁻ |
|---|---|
| Fibroblast | CD45⁻EpCAM⁻MHC-II⁻CD31⁻CD140a⁺ |
| Mesothelial | CD45-EpCAM⁻MHC-II⁻CD31⁻CD140a⁻CD140b⁺GP38⁺ |

### Developing T cells

| ETP | CD45+CD4-CD8-CD3-TCRb-CD44+CD25-c-Kit+ |
|---|---|
| DN1 | CD45+CD4-CD8-CD3-TCRb-CD44+CD25- |
| DN2 | CD45+CD4-CD8-CD3-TCRb-CD44+CD25+ |
| DN3 | CD45+CD4-CD8-CD3-TCRb-CD44-CD25+ |
| DN4 | CD45+CD4-CD8-CD3-TCRb-CD44-CD25- |
| DP | CD45+CD4+CD8+ |
| SP8 | CD45+CD4-CD8+CD3+TCRb+ |
| SP4 | CD45+CD4+CD8-CD3+TCRb+ |
| Treg | CD45+CD4+CD8-CD3+TCRb+ CD25+ |
| gdT | CD45+ CD4- CD8- TCRb- TCRgd+ |

### Bone Marrow

| LSK | CD45⁺Lineage⁻c-Kit+Sca-1+ |
|---|---|
| LT-HSC | CD45⁺Lineage⁻c-Kit+CD135-CD150+CD48- |
| ST-HSC | CD45⁺Lineage⁻c-Kit+CD135-CD150-CD48- |
| MPP2 | CD45⁺Lineage⁻c-Kit+CD135-CD150+CD48+ |
| MPP3 | CD45⁺Lineage⁻c-Kit+CD135-CD150-CD48- |
| MPP4 | CD45⁺Lineage⁻c-Kit+CD135+CD150- |
| T cell | CD45⁺B220⁻CD11b-CD11c-NK1.1-CD3+ |
| NK Cell | CD45⁺B220⁻CD11b-CD11c-NK1.1+CD3- |
| B cell | CD45⁺B220+CD11b- |
| Myeloid | CD45⁺B220-CD11b+ |

### Myeloid Cells

| cDC1 | CD45+ CD11c+ MHC-II+ XCR1+ CD11b- |
|---|---|
| cDC2 | CD45+ CD11c+ MHC-II+ XCR1- CD11b+ |
| Other Myeloid | CD45+ Non-cDC CD11c+ and/or CD11b+ |

### NK and ILC1

Gated: Viable+CD45+CD8-
TCRb-TCRgd-CD11c-

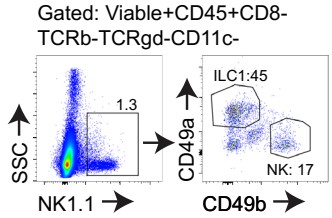

### ILC2

Gated: Viable+CD45+CD8-
TCRb-TCRgd-CD11c-

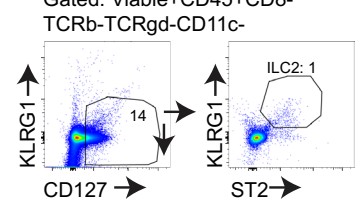

### NKT

Gated: Viable+CD45+

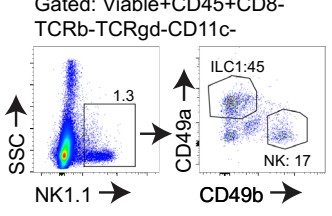

### ILC3
Gated: Viable+CD45+CD8-
TCRb-TCRgd-CD11c-

| NKT1 | CD45⁺CD4⁻CD8⁻TCRb⁺NK1.1⁺PBS-57 Tetramer⁺ |
|---|---|
| NKT2 | CD45⁺CD4⁻CD8⁻TCRb⁺NK1.1⁺PBS-57 Tetramer⁻ |
| NK | CD45⁺CD4⁻CD8⁻CD3⁻TCRb⁻TCRgd⁻NK1.1⁺CD49a⁻CD49b⁺ |
| ILC1 | CD45⁺CD4⁻CD8⁻CD3⁻TCRb⁻TCRgd⁻NK1.1⁺CD49a⁺CD49b⁻ |
| ILC2 | CD45⁺CD4⁻CD8⁻CD3⁻TCRb⁻TCRgd⁻CD11c⁻CD127⁺KLRG1⁺ST2⁺ |
| ILC3 | CD45⁺CD4⁻CD8⁻CD3⁻TCRb⁻TCRgd⁻CD11c⁻CD127⁺MHCII⁺CCR6⁺IL-23R⁺ |

**Extended Data Fig. 3 | Gating strategies for cell subsets in bone marrow and thymus.** Bone marrow and thymuses of untreated female 1-2 mo C57BL/6 mice were isolated and digested for follow cytometric identification of the populations described. Concatenated flow plots of gating strategies for mTEC1, mTEC2, mTEc3hi/lo, Tuft, Endothelial, Mesothelial and Fibroblast cells shown above. Gating strategies for NKT1, NKT2, NK, ILC-1, ILC-2, and ILC3 cells shown below. Tables show phenotypes used to gate individual populations.

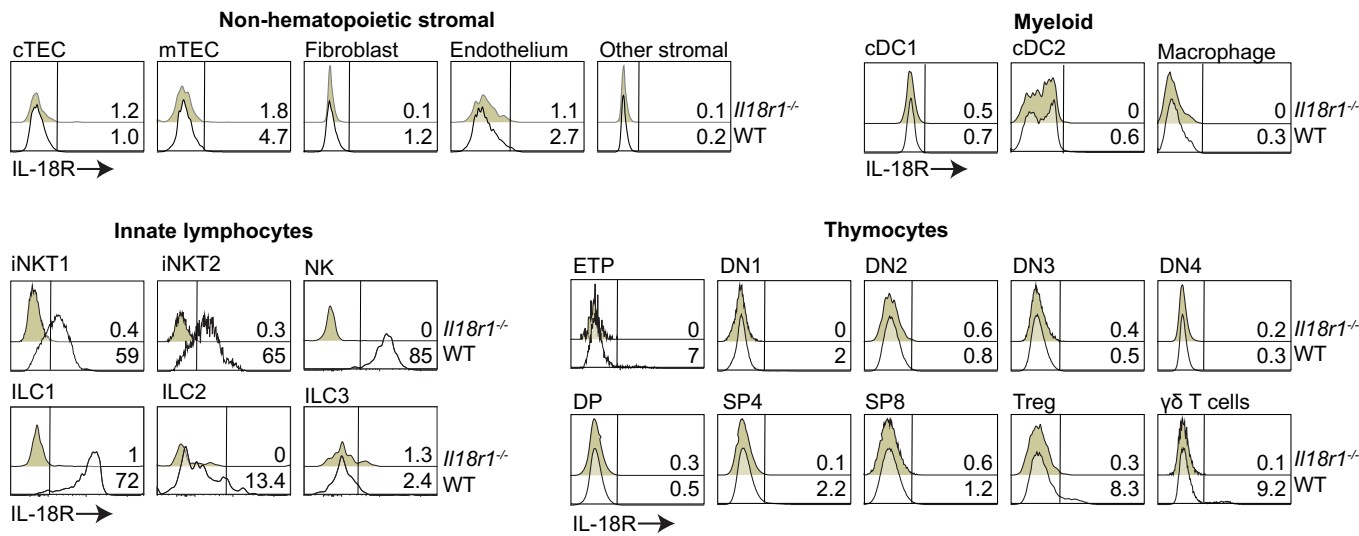

**Extended Data Fig. 4 | IL-18R expression across thymus cell subsets.** IL-18R expression on thymic cellular populations (see Extended Data Fig. 3 for gating) taken from female 1-2 mo C57BL/6 WT or *Il18r1*⁻/⁻ mice (n = 3-5/group).

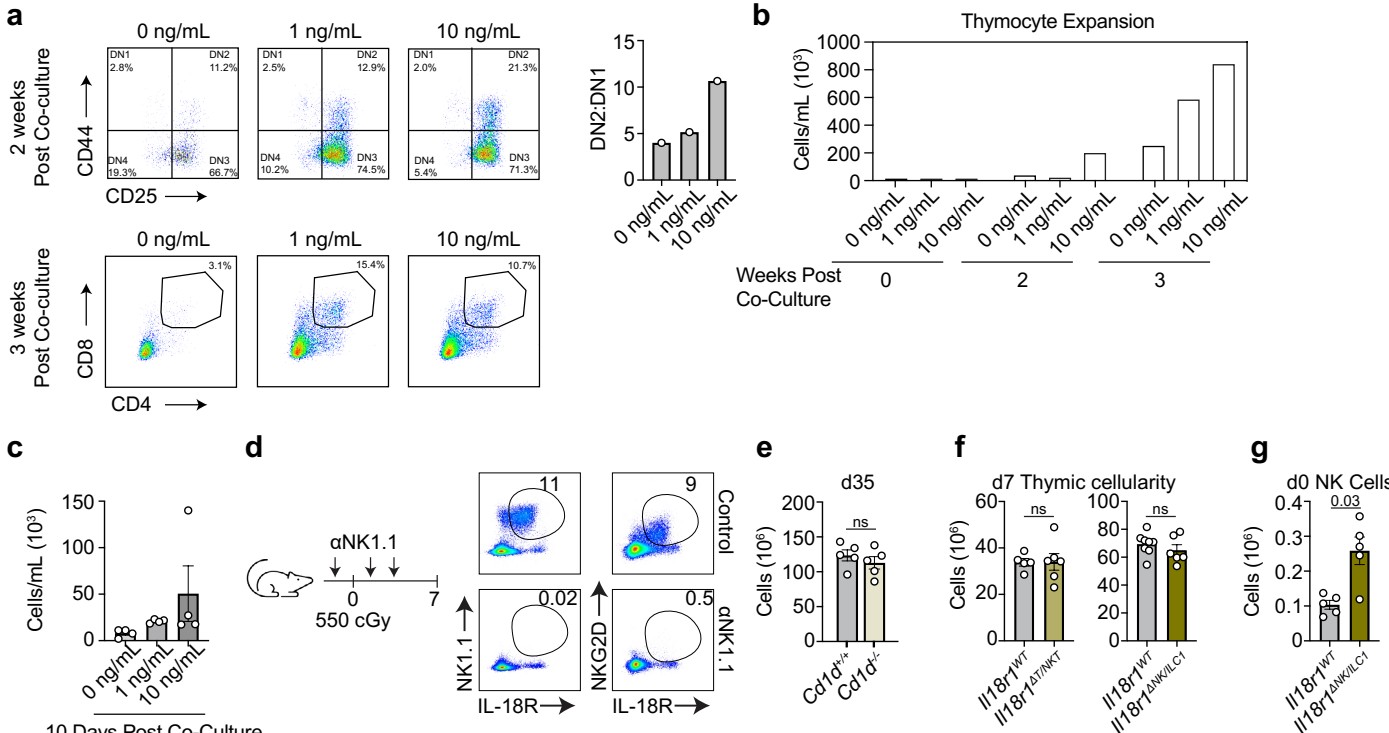

**Extended Data Fig. 5 | Effect of IL-18 on T cell development in vitro.**
**a**–**c**, 50,000 lineage depleted bone marrow cells were co-cultured with
OP9-DLL1$^{GFP}$ fibroblasts confluent in a 6-well dish with 5 ng/mL Flt3L plus 1 ng/mL
IL-7 and 0, 1 or 10 ng/mL rIL-18 (n = 1/group). **a**, (Top) CD45$^+$ Thy1$^+$ CD4$^-$ CD8$^-$
DN1-4 thymocyte differentiation measured according to CD44 and CD25
expression 2 weeks post co-culture and (Bottom) CD45$^+$Thy1$^+$ thymocyte
differentiation into CD4$^+$ CD8$^+$ Double Positive population 3 weeks post
co-culture. (Right) Ratio of DN2:DN1 thymocyte differentiation 2 weeks post
co-culture. **b**, Thymocyte expansion measured by non-adherent cell count
quantified 2 and 3 weeks post co-culture with OP9-DLL1$^{GFP}$ adherent cells.
**c**, 50,000 bone marrow CD45+ Lineage- cKit+ Sca-1+ LSKs were FACS purified
and co-cultured with OP9-DLL1$^{GFP}$ fibroblasts confluent in a 6-well dish with
5 ng/mL Ftl3L plus 1 ng/mL IL-7 and 0, 1 or 10 ng/mL rIL-18 (n = 4/group) and

10 days later, thymocyte expansion was quantified by measuring non-adherent
cell expansion. **d**, Female 1-2 mo C57BL/6 mice were administered 200μg αNK1.1
monoclonal antibody or control PBS (i.p) at days -1, 1 and 3 days post SL-TBI and
thymuses assessed at day 7. Thymus NK1.1$^+$IL-18R$^+$ (Left) and NKG2D$^+$IL-18R$^+$
(Right) cells (parent gated on viable CD45$^+$CD4$^-$CD8$^-$ cells) (n = 9/group).
**e**, Female 1-2 mo C57BL/6 WT (*Cd1d$^{+/+}$*) and *Cd1d$^{-/-}$* mice were given SL-TBI and
thymus cellularity was measured 35 days later (n = 5/group). **f**, Female 1-2 mo
*Il18r1$^{fl/fl}$:Ncr1-Cre$^-$* (*Il18r1$^{WT}$*, n = 5) and *Il18r1$^{fl/fl}$:Ncr1-Cre$^+$* (*Il18r1$^{ΔNK/ILC1}$*, n = 6) mice
were given SL-TBI and thymus cellularity measured on day 7 post SL-TBI. **g**, Female
1-2 mo *l18r1$^{WT}$* and *Il18r1$^{ΔNK/ILC1}$* thymus CD45$^+$ NK1.1$^+$ TCRβ$^-$CD49b$^+$ NK cells at
baseline (n = 5/group). Graphs represent mean ± SEM; each dot represents an
individual biological replicate; ns = not significant. Statistics were generated for
**e**, **f** using unpaired two-tailed t-tests.

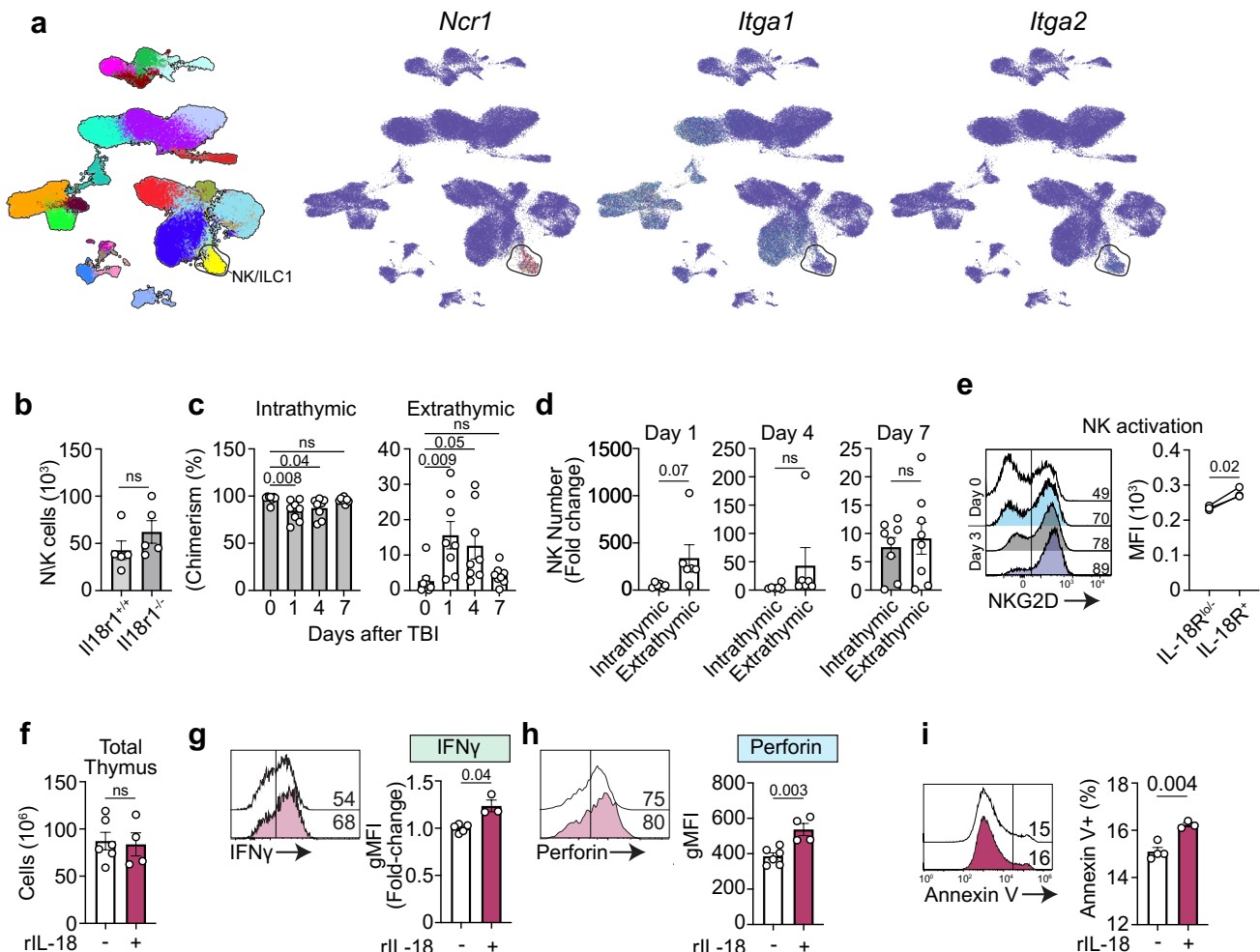

**Extended Data Fig. 6 | Increased NK cells is a function of both extrathymic recruitment and intrathymic expansion. a**, Taken from scRNAseq described in from Fig. 2a: integrated UMAP showing undamaged, baseline thymus major clusters and annotations (left) and heatmap expression *Ncr1, Itga1* and *Itga1*. **b**, Number of NK cells at day 3 in WT or *Il18r1*$^{-/-}$ mice (n = 5/group). **c**, **d**, Female C57BL/6 CD45.1$^+$ and CD45.2$^+$ mice were surgically conjoined to establish parabiotic pairs when both members of the pair received sublethal total body irradiation (SL-TBI; 550 cGy). Intrathymic/extrathymic chimerism was calculated on days 0, 1, 4, or 7 days after SL-TBI using congenic markers (CD45.1 and CD45.2) to determine mouse of origin (that is, cells expressing the same CD45 isoform as assessed mouse pair thymus = "intrathymic"; cells expressing the alternate isoform = "extrathymic"-derived). Absolute chimerism (**c**) (n = 8/group) and fold-change in NK cell number (**d**) across timepoints (d1, n = 6; d4, n = 6; d7, n = 8). **e**, Concatenated flow cytometry plots shown of NKG2D expression on CD45$^+$

NK1.1$^+$ TCRβ$^-$CD49b$^+$ NK cells and gMFI of NKG2D expression of IL-18R$^-$ and IL-18R$^+$ NK cells at day 3 post SL-TBI (n = 3/group). **f**–**h**, Female 1-2 mo C57BL/6 WT (-rIL-18, n = 6; +rIL-18, n = 4) or *Ifng*-reporter (-rIL-18, n = 5; +rIL-18, n = 3) mice were given rIL-18 (s.c., 2.5 mg/kg) or PBS. **f**, Total thymus cellularity assessed on day 3. **g**–**h**, Expression of *Ifng*-GFP (**g**) or Perforin (**h**) at day 3. **i**, Female 1-2mo C57BL/6 mice were given SL-TBI and 3 days later administered with rIL-18 (s.c., 2.5 mg/kg, n = 3) or PBS (n = 4) as in Fig. 4k. On day 7 NK1.1$^+$IL-18R$^+$TCRβ$^-$CD49b$^+$ NK cells were then FACS purified and cocultured with cell-dye labeled RMA-S target cells at a 5:1 Effector to Target ratio. RMA-S target cell Annexin V expression was measured 5 h post co-culture. Graphs represent mean ± SEM; each dot represents an individual biological replicate; ns = not significant. Statistics were generated for **b**–**d** and **f**–**i** using unpaired two-tailed t-tests, e using paired two-tailed t-tests, and for c using one-way ANOVA with Dunnet's correction for multiple comparisons.

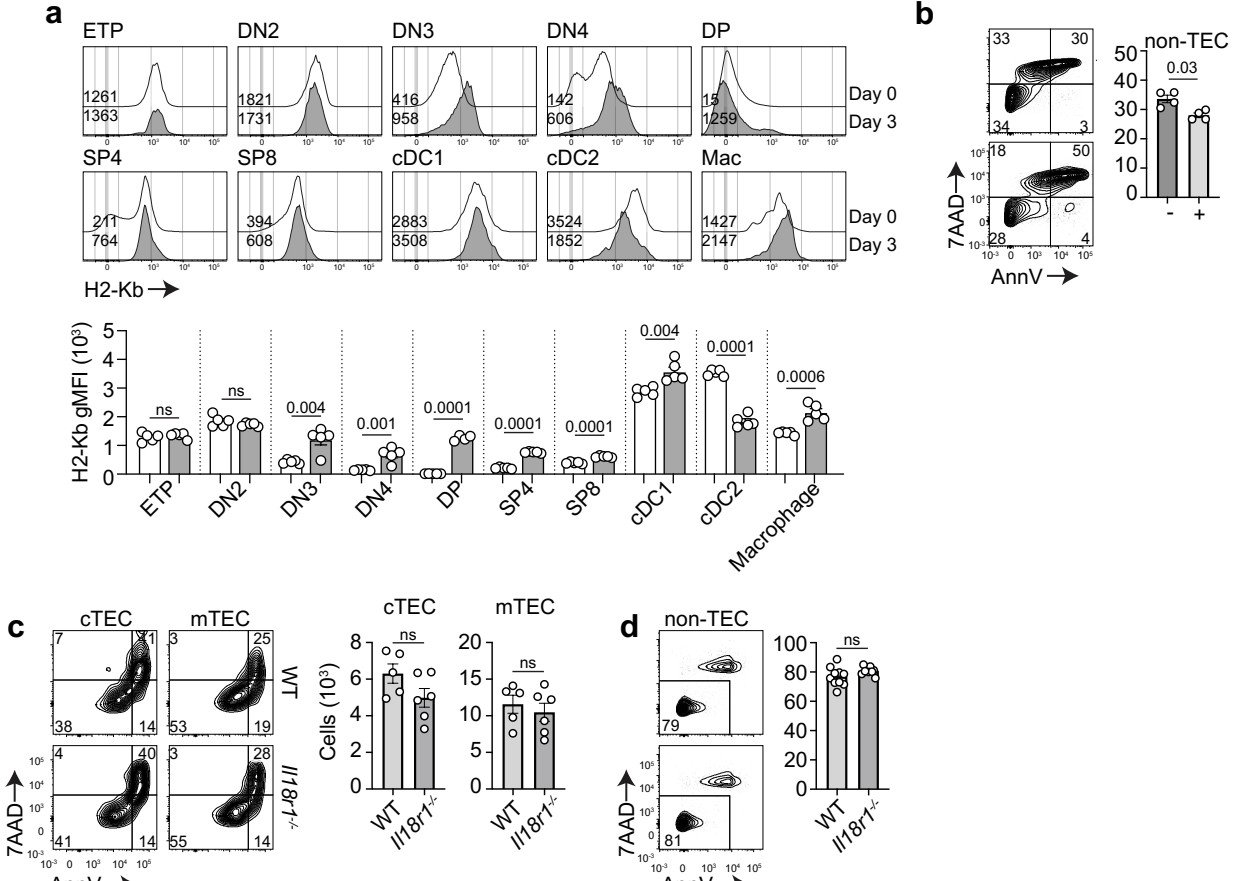

**Extended Data Fig. 7 | MHC I expression across thymocyte subsets after damage. a**, Female 1-2 mo C57BL/6 mice were given SL-TBI and thymuses collected 0 and 3 days later. Concatenated flow plots showing H2kb expression of early thymic progenitors (ETPs), DN2-4, DP, SP4, SP8 cDC1, cDC2 cells (see Extended Data Fig. 3 for gating) and CD45+CD11b+Tim-4+ Macrophages and their respective gMFI 0 and 3 days post SL-TBI (n = 5/group). **b**, Female 1-2mo C57BL/6 mice were given SL-TBI and thymuses collected 3 days later and enriched for non-hematopoietic stromal cells which were cultured with or without poly(I:C) stimulated NK cells. Annexin V and 7-AAD expression of CD45⁻ EpCAM⁻ MHC-

II⁻ non-TECs was measured at 5 h post co-culture (n = 4/group). **c**, Female 1-2 mo C57BL/6 WT (n = 5) or *Il18r1⁻/⁻* (n = 6) thymuses collected at baseline and Annexin V and 7-AAD expression of cTECs and mTECs assessed. d, Female 1-2 mo C57BL/6 WT (n = 10) or *Il18r1⁻/⁻* (n = 9) were given SL-TBI and thymuses collected at 5 days later. Annexin V and 7-AAD expression of CD45⁻EpCAM⁻MHC-II⁻ non-TECs was measured. Graphs represent mean ± SEM; each dot represents an individual biological replicate; ns = not significant. Statistics were generated for **a**–**d** using unpaired two-tailed t-tests.

Marcel van den Brink

# Reporting Summary

## Statistics

For all statistical analyses, confirm that the following items are present in the figure legend, table legend, main text, or Methods section.

| n/a | Confirmed | |
|---|---|---|
| ☐ | ☒ | The exact sample size (*n*) for each experimental group/condition, given as a discrete number and unit of measurement |
| ☐ | ☒ | A statement on whether measurements were taken from distinct samples or whether the same sample was measured repeatedly |
| ☐ | ☒ | The statistical test(s) used AND whether they are one- or two-sided<br>*Only common tests should be described solely by name; describe more complex techniques in the Methods section.* |
| ☐ | ☒ | A description of all covariates tested |
| ☐ | ☒ | A description of any assumptions or corrections, such as tests of normality and adjustment for multiple comparisons |
| ☐ | ☒ | A full description of the statistical parameters including central tendency (e.g. means) or other basic estimates (e.g. regression coefficient) AND variation (e.g. standard deviation) or associated estimates of uncertainty (e.g. confidence intervals) |
| ☒ | ☐ | For null hypothesis testing, the test statistic (e.g. *F*, *t*, *r*) with confidence intervals, effect sizes, degrees of freedom and *P* value noted<br>*Give P values as exact values whenever suitable.* |
| ☒ | ☐ | For Bayesian analysis, information on the choice of priors and Markov chain Monte Carlo settings |
| ☒ | ☐ | For hierarchical and complex designs, identification of the appropriate level for tests and full reporting of outcomes |
| ☒ | ☐ | Estimates of effect sizes (e.g. Cohen's *d*, Pearson's *r*), indicating how they were calculated |

*Our web collection on statistics for biologists contains articles on many of the points above.*

## Software and code

Policy information about availability of computer code

| Data collection | Cells were sorted on an Aria II or Symphony S6 (BD Biosciences) using FACSDiva (BD Biosciences, Franklin Lakes, NJ). Flow cytometry analysis was performed on a Symphony A6 using FACSDiva Software (BD Biosciences). Sequencing data were collected on Illumina Nextseq or Novaseq platforms. Tissue sections were imaged on a Zeiss LSM 880 confocal microscope. Formalin-fixed paraffin-embedded thymus tissue section images were acquired on the Vectra Polaris Quantitative Pathology Imaging System (Akoya Biosciences, Marlborough, MA). |
|---|---|
| Data analysis | #1. Flow cytometry data were analyzed using FlowJo (v10.7.1, TreeStar Software) and GraphPad Prism (v10; GraphPad Software, LLC). Data was collected and cells sorted on BD cytometers using FACSDiva software.<br><br>#2. All single cell RNA-seq data were analyzed using Cell Ranger (v7.0.1; 10x Genomics). The shunPykeR adapted Jupyter notebooks, R notebooks, and the assorted conda (.yml) and renv (renv.lock) environment files to reproduce analyses and figure creation for this manuscript can be found at https://github.com/kousaa/Kousa-et-al-2024-NI. The app.R code that launches the ThymoSight app, together with the python notebooks used to create consistent annotation fields, reanalyze and integrate the public datasets with ours have been submitted on GitHub at https://github.com/FredHutch/thymosight. The server hosting the interactive app can be accessed at www.thymosight.org. Environments and packages can be found listed below.<br><br>#3. Other software: Cytoscape (v3.10.0); HALO image analysis software (v4.0)<br><br>Environment names and requirements for scRNAseq analsysis:<br><br>name: Kousa-et-al-2024-NI (main), channels: conda-forge, defaults, dependencies: -alabaster=0.7.12, -anndata=0.8.0, -appnope=0.1.2, -argon2-cffi=21.3.0, -argon2-cffi-bindings=21.2.0, -arpack=3.7.0, -asttokens=2.0.5, -attrs=21.4.0, -babel=2.10.3, -backcall=0.2.0, - |

beautifulsoup4=4.11.1, -bleach=4.1.0, -brotli=1.0.9, -brotli-bin=1.0.9, -brotlipy=0.7.0, -ca-certificates=2022.6.15.1, -certifi=2022.6.15.1, -cffi=1.15.1, -charset-normalizer=2.1.1, -colorama=0.4.5, -cryptography=37.0.4, -cycler=0.11.0, -debugpy=1.5.1, -decorator=5.1.1, -defusedxml=0.7.1, -docutils=0.19, -entrypoints=0.4, -executing=0.8.3, -fonttools=4.37.1, -freetype=2.10.4, -glpk=4.65, -gmp=6.2.1, -h5py=3.7.0, -hdf5=1.10.6, -icu=58.2, -igraph=0.9.10, -imagesize=1.4.1, -importlib-metadata=4.11.4, -importlib_metadata=4.11.4, -importlib_resources=5.2.0, -ipykernel=6.15.2, -ipython=8.4.0, -ipython_genutils=0.2.0, -jedi=0.18.1, -jinja2=3.0.3, -joblib=1.1.0, -jpeg=9e, -jsonschema=4.4.0, -jupyter_client=7.3.5, -jupyter_core=4.10.0, -jupyterlab_pygments=0.1.2, -kiwisolver=1.4.4, -lcms2=2.12, -leidenalg=0.8.10, -libblas=3.9.0, -libbrotlicommon=1.0.9, -libbrotlidec=1.0.9, -libbrotlienc=1.0.9, -libcblas=3.9.0, -libcxx=14.0.6, -libffi=3.3, -libgfortran=4.0.0, -libgfortran4=7.5.0, -libiconv=1.17, -liblapack=3.9.0, -libllvm11=11.1.0, -libopenblas=0.3.12, -libpng=1.6.37, -libsodium=1.0.18, -libtiff=4.2.0, -libwebp-base=1.2.4, -libxml2=2.9.14, -llvm-openmp=14.0.4, -llvmlite=0.38.0, -lz4-c=1.9.3, -markupsafe=2.1.1, -matplotlib-base=3.5.3, -matplotlib-inline=0.1.6, -metis=5.1.0, -mistune=0.8.4, -mpfr=4.1.0, -munkres=1.1.4, -natsort=8.2.0, -nbclient=0.5.13, -nbconvert=6.4.4, -nbformat=5.3.0, -ncurses=6.3, -nest-asyncio=1.5.5, -networkx=2.8.6, -notebook=6.4.12, -numba=0.55.1, -olefile=0.46, -openssl=1.1.1q, -packaging=21.3, -pandas=1.4.4, -pandocfilters=1.5.0, -parso=0.8.3, -patsy=0.5.2, -pexpect=4.8.0, -pickleshare=0.7.5, -pip=22.1.2, -prometheus_client=0.14.1, -prompt-toolkit=3.0.20, -psutil=5.9.0, -ptyprocess=0.7.0, -pure_eval=0.2.2, -pycparser=2.21, -pygments=2.11.2, -pynndescent=0.5.7, -pyopenssl=22.0.0, -pyparsing=3.0.9, -pyrsistent=0.18.0, -pysocks=1.7.1, -python=3.8.13, -python-dateutil=2.8.2, -python-fastjsonschema=2.16.2, -python-igraph=0.9.11, -python_abi=3.8, -pytz=2022.2.1, -pyzmq=23.2.0, -readline=8.1.2, -scanpy=1.9.1, -scikit-learn=1.1.2, -scipy=1.5.3, -seaborn=0.12.0, -seaborn-base=0.12.0, -send2trash=1.8.0, -session-info=1.0.0, -setuptools=63.4.1, -six=1.16.0, -snowballstemmer=2.2.0, -soupsieve=2.3.1, -sphinx=5.1.1, -sphinxcontrib-applehelp=1.0.2, -sphinxcontrib-devhelp=1.0.2, -sphinxcontrib-htmlhelp=2.0.0, -sphinxcontrib-jsmath=1.0.1, -sphinxcontrib-qthelp=1.0.3, -sphinxcontrib-serializinghtml=1.1.5, -sqlite=3.39.2, -stack_data=0.2.0, -statsmodels=0.13.2, -stdlib-list=0.7.0, -suitesparse=5.10.1, -tbb=2021.5.0, -terminado=0.13.1, -testpath=0.6.0, -texttable=1.6.4, -threadpoolctl=3.1.0, -tk=8.6.12, -tornado=6.2, -tqdm=4.64.1, -traitlets=5.1.1, -typing-extensions=4.3.0, -typing_extensions=4.3.0, -umap-learn=0.5.3, -unicodedata2=14.0.0, -urllib3=1.26.11, -wcwidth=0.2.5, -webencodings=0.5.1, -wheel=0.37.1, -xz=5.2.5, -zeromq=4.3.4, -zipp=3.8.0, -zlib=1.2.12, -zstd=1.4.9, -pip:, -aniso8601==9.0.1, -anndata2ri==1.1, -annoy==1.17.1, -backports-zoneinfo==0.2.1, -biomart==0.9.2, -chardet==4.0.0, -click==8.1.3, -cmake==3.24.1.1, -cython==0.29.32, -dunamai==1.13.0, -et-xmlfile==1.1.0, -fa2==0.3.5, -fbpca==1.0, -fcsparser==0.2.4, -feather-format==0.4.1, -flask==1.1.2, -flask-cors==3.0.10, -flask-restful==0.3.9, -geosketch==1.2, -get-version==3.5.4, -harmonypy==0.0.6, -harmonyts==0.1.4, -idna==2.10, -imageio==2.22.1, -intervaltree==3.1.0, -itsdangerous==2.1.2, -loompy==3.0.7, -mousipy==0.0.5, -nbstripout==0.6.1, -numexpr==2.8.3, -numpy==1.20.2, -numpy-groupies==0.9.19, -openpyxl==3.0.10, -palantir==1.0.1, -phenograph==1.5.7, -pillow==9.2.0, -pyarrow==9.0.0, -pyreadr==0.4.6, -python-slugify==5.0.2, -pytz-deprecation-shim==0.1.0.post0, -pywavelets==1.3.0, -requests==2.25.1, -rpy2==3.5.4, -scanorama==1.7.2, -scikit-image==0.19.3, -scrublet==0.2.3, -scvelo==0.2.4, -sklearn==0.0, -sortedcontainers==2.4.0, -tables==3.7.0, -text-unidecode==1.3, -tifffile==2022.8.12, -tzdata==2022.2, -tzlocal==4.2, -watermark==2.3.1, -werkzeug==2.2.2, -wrapt==1.15.0, -xlrd==2.0.1, -xlsxwriter==3.0.3

name: Kousa-et-al-2024-NI (dynamo), channels: defaults, dependencies: -anyio=4.2.0, -appnope=0.1.2, -argon2-cffi=21.3.0, -argon2-cffi-bindings=21.2.0, -asttokens=2.0.5, -async-lru=2.0.4, -babel=2.11.0, -beautifulsoup4=4.12.2, -bleach=4.1.0, -brotli-python=1.0.9, -bzip2=1.0.8, -ca-certificates=2023.12.12, -certifi=2024.2.2, -cffi=1.16.0, -charset-normalizer=2.0.4, -comm=0.1.2, -debugpy=1.6.7, -decorator=5.1.1, -defusedxml=0.7.1, -executing=0.8.3, -ipykernel=6.28.0, -ipython=8.20.0, -jedi=0.18.1, -json5=0.9.6, -jsonschema=4.19.2, -jsonschema-specifications=2023.7.1, -jupyter-lsp=2.2.0, -jupyter_client=8.6.0, -jupyter_core=5.5.0, -jupyter_events=0.8.0, -jupyter_server=2.10.0, -jupyter_server_terminals=0.4.4, -jupyterlab=4.0.11, -jupyterlab_pygments=0.1.2, -jupyterlab_server=2.25.1, -libcxx=14.0.6, -libffi=3.4.4, -libsodium=1.0.18, -matplotlib-inline=0.1.6, -mistune=2.0.4, -nbclient=0.8.0, -nbconvert=7.10.0, -nbformat=5.9.2, -ncurses=6.4, -nest-asyncio=1.6.0, -notebook=7.0.8, -notebook-shim=0.2.3, -openssl=3.0.13, -overrides=7.4.0, -pandocfilters=1.5.0, -parso=0.8.3, -pexpect=4.8.0, -pip=23.3.1, -prometheus_client=0.14.1, -prompt-toolkit=3.0.43, -prompt_toolkit=3.0.43, -psutil=5.9.0, -ptyprocess=0.7.0, -pure_eval=0.2.2, -pycparser=2.21, -pysocks=1.7.1, -python=3.11.8, -python-fastjsonschema=2.16.2, -python-json-logger=2.0.7, -pyyaml=6.0.1, -pyzmq=25.1.2, -readline=8.2, -referencing=0.30.2, -requests=2.31.0, -rfc3339-validator=0.1.4, -rfc3986-validator=0.1.1, -rpds-py=0.10.6, -send2trash=1.8.2, -setuptools=68.2.2, -six=1.16.0, -sniffio=1.3.0, -soupsieve=2.5, -sqlite=3.41.2, -stack_data=0.2.0, -terminado=0.17.1, -tinycss2=1.2.1, -tk=8.6.12, -tornado=6.3.3, -traitlets=5.7.1, -typing_extensions=4.9.0, -urllib3=2.1.0, -wcwidth=0.2.5, -webencodings=0.5.1, -websocket-client=0.58.0, -wheel=0.41.2, -xz=5.4.6, -yaml=0.2.5, -zeromq=4.3.5, -zlib=1.2.13, -pip:, -absl-py==2.1.0, -aiohttp==3.9.3, -aiosignal==1.3.1, -anndata==0.10.5.post1, -array-api-compat==1.4.1, -attrs==23.2.0, -cellrank==2.0.3.dev5+g721c59f, -cfgv==3.4.0, -chex==0.1.85, -click==8.1.7, -colorcet==3.1.0, -contextlib2==21.6.0, -contourpy==1.2.0, -cycler==0.12.1, -distlib==0.3.8, -docrep==0.3.2, -dunamai==1.19.2, -dynamo-release==1.4.0, -et-xmlfile==1.1.0, -etils==1.7.0, -filelock==3.13.1, -flax==0.8.1, -fonttools==4.49.0, -frozenlist==1.4.1, -fsspec==2024.2.0, -get-version==3.5.5, -h5py==3.10.0, -identify==2.5.35, -idna==3.6, -igraph==0.10.8, -importlib-resources==6.1.2, -jax==0.4.25, -jaxlib==0.4.25, -jinja2==3.0.3, -joblib==1.3.2, -kiwisolver==1.4.5, -lightning==2.1.4, -lightning-utilities==0.10.1, -llvmlite==0.42.0, -loompy==3.0.7, -louvain==0.8.0, -markdown-it-py==3.0.0, -markupsafe==2.1.5, -matplotlib==3.6.3, -mdurl==0.1.2, -ml-collections==0.1.1, -ml-dtypes==0.3.2, -mpmath==1.3.0, -msgpack==1.0.8, -mudata==0.2.3, -multidict==6.0.5, -multipledispatch==1.0.0, -natsort==8.4.0, -networkx==3.2.1, -nodeenv==1.8.0, -numba==0.59.0, -numdifftools==0.9.41, -numpy==1.26.4, -numpy-groupies==0.10.2, -numpyro==0.13.2, -openpyxl==3.1.2, -opt-einsum==3.3.0, -optax==0.1.9, -orbax-checkpoint==0.5.3, -packaging==23.2, -pandas==2.2.1, -patsy==0.5.6, -pillow==10.2.0, -platformdirs==4.2.0, -pre-commit==3.6.2, -progressbar2==4.4.1, -protobuf==4.25.3, -pygam==0.9.1, -pygments==2.17.2, -pygpcca==1.0.4, -pynndescent==0.5.11, -pyparsing==3.1.1, -pyro-api==0.1.2, -pyro-ppl==1.9.0, -python-dateutil==2.9.0, -python-utils==3.8.2, -pytorch-lightning==2.2.0.post0, -pytz==2024.1, -rich==13.7.1, -scanpy==1.9.8, -scikit-learn==1.1.3, -scipy==1.11.4, -scvelo==0.3.1, -scvi-tools==1.1.1, -seaborn==0.13.2, -session-info==1.0.0, -statsmodels==0.14.1, -stdlib-list==0.10.0, -sympy==1.12, -tensorstore==0.1.54, -texttable==1.7.0, -threadpoolctl==3.3.0, -toolz==0.12.1, -torch==2.2.1, -torchmetrics==1.3.1, -tqdm==4.66.2, -typing-extensions==4.10.0, -tzdata==2024.1, -umap-learn==0.5.5, -virtualenv==20.25.1, -wrapt==1.16.0, -yarl==1.9.4, -zipp==3.17.0

name: Kousa-et-al-2024-NI (thymosight), channels: -conda-forge, -defaults, dependencies: -anndata=0.10.5.post1, -anyio=4.2.0, -appnope=0.1.2, -argon2-cffi=21.3.0, -argon2-cffi-bindings=21.2.0, -arpack=3.8.0, -array-api-compat=1.4.1, -asttokens=2.0.5, -async-lru=2.0.4, -attrs=23.1.0, -babel=2.11.0, -beautifulsoup4=4.12.2, -bleach=4.1.0, -brotli=1.1.0, -brotli-bin=1.1.0, -brotli-python=1.0.9, -bzip2=1.0.8, -c-ares=1.27.0, -ca-certificates=2024.2.2, -cached-property=1.5.2, -cached_property=1.5.2, -certifi=2024.2.2, -cffi=1.16.0, -charset-normalizer=2.0.4, -colorama=0.4.6, -contourpy=1.2.0, -cycler=0.12.1, -debugpy=1.6.7, -decorator=5.1.1, -defusedxml=0.7.1, -exceptiongroup=1.2.0, -executing=0.8.3, -fonttools=4.25.0, -freetype=2.12.1, -get-annotations=0.1.2, -glpk=5.0, -gmp=6.3.0, -h5py=3.10.0, -hdf5=1.14.3, -icu=73.2, -idna=3.4, -igraph=0.10.10, -ipykernel=6.28.0, -ipython=8.20.0, -jedi=0.18.1, -jinja2=3.1.3, -joblib=1.3.2, -json5=0.9.6, -jsonschema=4.19.2, -jsonschema-specifications=2023.7.1, -jupyter-lsp=2.2.0, -jupyter_client=8.6.0, -jupyter_core=5.5.0, -jupyter_events=0.8.0, -jupyter_server=2.10.0, -jupyter_server_terminals=0.4.4, -jupyterlab=4.0.11, -jupyterlab_pygments=0.1.2, -jupyterlab_server=2.25.1, -kiwisolver=1.4.5, -krb5=1.21.2, -lcms2=2.16, -leidenalg=0.10.2, -lerc=4.0.0, -libaec=1.1.2, -libblas=3.9.0, -libbrotlicommon=1.1.0, -libbrotlidec=1.1.0, -libbrotlienc=1.1.0, -libcblas=3.9.0, -libcurl=8.5.0, -libcxx=16.0.6, -libdeflate=1.19, -libedit=3.1.20191231, -libev=4.33, -libexpat=2.5.0, -libffi=3.4.4, -libgfortran=5.0.0, -libgfortran5=13.2.0, -libhwloc=2.9.3, -libiconv=1.17, -libjpeg-turbo=3.0.0, -liblapack=3.9.0, -libleidenalg=0.11.1, -libllvm14=14.0.6, -libnghttp2=1.58.0, -libopenblas=0.3.26, -libpng=1.6.43, -libsodium=1.0.18, -libsqlite=3.45.1, -libssh2=1.11.0, -libtiff=4.6.0, -libwebp-base=1.3.2, -libxcb=1.15, -libxml2=2.12.5, -libzlib=1.2.13, -llvm-openmp=17.0.6, -llvmlite=0.42.0, -markupsafe=2.1.3, -matplotlib-base=3.8.3, -matplotlib-inline=0.1.6, -mistune=2.0.4, -munkres=1.1.4, -

natsort=8.4.0, -nbclient=0.8.0, -nbconvert=7.10.0, -nbformat=5.9.2, -ncurses=6.4, -nest-asyncio=1.6.0, -networkx=3.2.1, -notebook=7.0.8, -notebook-shim=0.2.3, -numba=0.59.0, -numpy=1.26.4, -openjpeg=2.5.0, -openssl=3.2.1, -overrides=7.4.0, -packaging=23.1, -pandas=2.2.1, -pandocfilters=1.5.0, -parso=0.8.3, -patsy=0.5.6, -pexpect=4.8.0, -pillow=10.2.0, -pip=23.3.1, -platformdirs=3.10.0, -prometheus_client=0.14.1, -prompt-toolkit=3.0.43, -prompt_toolkit=3.0.43, -psutil=5.9.0, -pthread-stubs=0.4, -ptyprocess=0.7.0, -pure_eval=0.2.2, -pycparser=2.21, -pygments=2.15.1, -pynndescent=0.5.11, -pyparsing=3.1.1, -pysocks=1.7.1, -python=3.11.8, -python-dateutil=2.8.2, -python-fastjsonschema=2.16.2, -python-igraph=0.11.4, -python-json-logger=2.0.7, -python-tzdata=2024.1, -python_abi=3.11, -pytz=2023.3.post1, -pyyaml=6.0.1, -pyzmq=25.1.2, -readline=8.2, -referencing=0.30.2, -requests=2.31.0, -rfc3339-validator=0.1.4, -rfc3986-validator=0.1.1, -rpds-py=0.10.6, -scanpy=1.9.8, -scikit-learn=1.4.1.post1, -scipy=1.12.0, -seaborn=0.13.2, -seaborn-base=0.13.2, -send2trash=1.8.2, -session-info=1.0.0, -setuptools=68.2.2, -six=1.16.0, -sniffio=1.3.0, -soupsieve=2.5, -sqlite=3.41.2, -stack_data=0.2.0, -statsmodels=0.14.1, -stdlib-list=0.10.0, -tbb=2021.11.0, -terminado=0.17.1, -texttable=1.7.0, -threadpoolctl=3.3.0, -tinycss2=1.2.1, -tk=8.6.13, -tornado=6.3.3, -tqdm=4.66.2, -traitlets=5.7.1, -typing-extensions=4.9.0, -typing_extensions=4.9.0, -tzdata=2024a, -umap-learn=0.5.5, -urllib3=2.1.0, -wcwidth=0.2.5, -webencodings=0.5.1, -websocket-client=0.58.0, -wheel=0.41.2, -xorg-libxau=1.0.11, -xorg-libxdmcp=1.1.3, -xz=5.4.5, -yaml=0.2.5, -zeromq=4.3.5, -zlib=1.2.13, -zstd=1.5.5, -pip:, -annoy=1.17.3, -biomart==0.9.2, -comm==0.2.1, -cython==3.0.8, -et-xmlfile==1.1.0, -harmonypy==0.0.6, -imageio==2.34.0, -ipywidgets==8.1.2, -jupyterlab-widgets==3.0.10, -lazy-loader==0.3, -openpyxl==3.1.2, -scikit-image==0.22.0, -scrublet==0.2.3, -tifffile==2024.2.12, -widgetsnbextension==4.0.10

name: Kousa-et-al-NI (cellchat), -R=4.2.2, -Bioconductor=3.15, Python: 3.8.2 (virtualenv), packages: -BBmisc=1.13, -BH=1.78.0-0, -Biobase=2.56.0, -BiocGenerics=0.42.0, -BiocManager=1.30.19, -BiocParallel=1.30.4, -BiocVersion=3.15.2, -CellChat=1.4.0, -ComplexHeatmap=2.12.1, -DBI=1.1.3, -DelayedArray=0.22.0, -DelayedMatrixStats=1.18.2, -DropletUtils=1.16.0, -EnhancedVolcano=1.14.0, -FNN=1.1.3.1, -Formula=1.2-4, -GenomeInfoDb=1.32.4, -GenomeInfoDbData=1.2.8, -GenomicRanges=1.48.0, -GetoptLong=1.0.5, -GlobalOptions=0.1.2, -HDF5Array=1.24.2, -Hmisc=4.7-1, -IRanges=2.30.1, -KernSmooth=2.23-20, -MASS=7.3-58.1, -MAST=1.22.0, -Matrix=1.5-3, -MatrixGenerics=1.8.1, -MatrixModels=0.5-1, -NMF=0.24.0, -R.methodsS3=1.8.2, -R.oo=1.25.0, -R.utils=2.12.2, -R6=2.5.1, -RANN=2.6.1, -RColorBrewer=1.1-3, -RCurl=1.98-1.9, -ROCR=1.0-11, -RSpectra=0.16-1, -Rcpp=1.0.9, -RcppAnnoy=0.0.20, -RcppArmadillo=0.11.4.2.1, -RcppEigen=0.3.3.9.3, -RcppProgress=0.4.2, -RcppTOML=0.1.7, -Rhdf5lib=1.18.2, -Rtsne=0.16, -S4Vectors=0.34.0, -Seurat=4.3.0, -SeuratObject=4.1.3, -SingleCellExperiment=1.18.1, -SparseM=1.81, -SummarizedExperiment=1.26.1, -TSP=1.2-1, -UpSetR=1.4.0, -XVector=0.36.0, -abind=1.4-5, -anndata=0.7.5.3, -arm=1.13-1, -arrow=10.0.0, -askpass=1.1, -assertthat=0.2.1, -backports=1.4.1, -base64enc=0.1-3, -beachmat=2.12.0, -bit=4.0.5, -bit64=4.0.5, -bitops=1.0-7, -blob=1.2.3, -boot=1.3-28.1, -brew=1.0-7, -brio=1.1.3, -broom=1.0.1, -bslib=0.4.1, -ca=0.71.1, -caTools=1.18.2, -cachem=1.0.6, -callr=3.7.3, -car=3.1-1, -carData=3.0-5, -cellranger=1.1.0, -checkmate=2.1.0, -circlize=0.4.15, -cli=3.4.1, -clipr=0.8.0, -clue=0.3-63, -cluster=2.1.4, -coda=0.19-4, -codetools=0.2-18, -colorspace=2.0-3, -commonmark=1.8.1, -corrplot=0.92, -cowplot=1.1.1, -cpp11=0.4.3, -crayon=1.5.2, -credentials=1.3.2, -crosstalk=1.2.0, -curl=4.3.3, -data.table=1.14.6, -dbplyr=2.1.1, -deldir=1.0-6, -dendextend=1.16.0, -desc=1.4.2, -devtools=2.4.3, -dichromat=2.0-0.1, -diffobj=0.3.5, -digest=0.6.31, -doParallel=1.0.17, -dplyr=1.0.10, -dqrng=0.3.0, -dtplyr=1.2.2, -edgeR=3.38.4, -egg=0.4.5, -ellipsis=0.3.2, -emmeans=1.8.2, -estimability=1.4.1, -evaluate=0.19, -expm=0.999-6, -fansi=1.0.3, -farver=2.1.1, -fastmap=1.1.0, -feather=0.3.5, -figpatch=0.2, -fitdistrplus=1.1-8, -fontawesome=0.4.0, -forcats=0.5.2, -foreach=1.5.2, -foreign=0.8-83, -formatR=1.12, -formattable=0.2.1, -fs=1.5.2, -futile.logger=1.4.3, -futile.options=1.0.1, -future=1.29.0, -future.apply=1.10.0, -gargle=1.2.1, -gclus=1.3.2, -generics=0.1.3, -gert=1.6.0, -ggalluvial=0.12.3, -ggcorrplot=0.1.4, -ggplot2=3.4.0, -ggplotify=0.1.0, -ggpubr=0.5.0, -ggrepel=0.9.2, -ggridges=0.5.4, -ggsci=2.9, -ggsignif=0.6.4, -gh=1.3.0=0.1.1, -globals=0.16.2, -glue=1.6.2, -goftest=1.2-3, -googledrive=2.0.0, -googlesheets4=1.0.1, -gplots=3.1.3, -gprofiler2=0.2.1, -gridBase=0.4-7, -gridExtra=2.3, -gridGraphics=0.5-1, -gtable=0.3.1, -gtools=3.9.4, -haven=2.5.1, -heatmaply=1.4.0, -here=1.0.1, -highr=0.9, -hms=1.1.2, -htmlTable=2.4.1, -htmltools=0.5.4, -htmlwidgets=1.5.4, -httpuv=1.6.6, -httr=1.4.4, -ica=1.0-3, -ids=1.0.1, -igraph=1.3.5, -ini=0.3.1, -interp=1.1-3, -irlba=2.3.5.1, -isoband=0.2.6, -iterators=1.0.14, -jpeg=0.1-9, -jquerylib=0.1.4, -jsonlite=1.8.4, -knitr=1.41, -labeling=0.4.2, -lambda.r=1.2.4, -later=1.3.0, -lattice=0.20-45, -latticeExtra=0.6-30, -lazyeval=0.2.2, -leiden=0.4.3, -lifecycle=1.0.3, -limma=3.52.4, -listenv=0.8.0, -lme4=1.1-31, -lmtest=0.9-40, -locfit=1.5-9.6, -lubridate=1.8.0, -magick=2.7.3, -magrittr=2.0.3, -mapproj=1.2.9, -maps=3.4.1, -matrixStats=0.63.0, -memoise=2.0.1, -mgcv=1.8-41, -mime=0.12, -miniUI=0.1.1.1, -minqa=1.2.5, -modelr=0.1.9, -munsell=0.5.0, -mvtnorm=1.1-3, -naniar=0.6.1, -network=1.17.2, -nlme=3.1-160, -nloptr=2.0.3, -nnet=7.3-18, -norm=1.0-10.0, -numDeriv=2016.8-1.1, -openssl=2.0.5, -openxlsx=4.2.5.1, -pacman=0.5.1, -pals=1.7, -parallelly=1.32.1, -patchwork=1.1.2, -pbapply=1.6-0, -pbkrtest=0.5.1, -pheatmap=1.0.12, -pillar=1.8.1, -pkgbuild=1.3.1, -pkgconfig=2.0.3, -pkgload=1.3.2, -pkgmaker=0.32.2, -plotly=4.10.1, -plyr=1.8.8, -png=0.1-8, -polyclip=1.10-4, -polynom=1.4-1, -praise=1.0.0, -prettyunits=1.1.1, -processx=3.8.0, -progress=1.2.2, -progressr=0.12.0, -promises=1.2.0.1, -ps=1.7.2, -purrr=0.3.5, -qap=0.1-2, -quantreg=5.94, -rappdirs=0.3.3, -rcmdcheck=1.4.0, -readr=2.1.3, -readxl=1.4.1, -registry=0.5-1, -rematch=1.0.1, -rematch2=2.1.2, -remotes=2.4.2, -renv=0.15.5, -reprex=2.0.2, -reshape2=1.4.4, -reticulate=1.26, -rhdf5=2.40.0, -rhdf5filters=1.8.0, -rjson=0.2.21, -rlang=1.0.6, -rmarkdown=2.17, -rngtools=1.5.2, -roxygen2=7.1.2, -rpart=4.1.19, -rprojroot=2.0.3, -rstatix=0.7.1, -rstudioapi=0.13, -rversions=2.1.1, -rvest=1.0.3, -sass=0.4.4, -scCB2=1.6.0, -scales=1.2.1, -scattermore=0.8, -sctransform=0.3.5, -scuttle=1.6.3, -selectr=0.4-2, -seriation=1.4.0, -sessioninfo=1.2.2, -shape=1.4.6, -shiny=1.7.3, -sitmo=2.0.2, -sna=2.7, -snow=0.4-4, -sourcetools=0.1.7, -sp=1.5-1, -sparseMatrixStats=1.8.0, -spatstat.data=3.0-0, -spatstat.explore=3.0-5, -spatstat.geom=3.0-3, -spatstat.random=3.0-1, -spatstat.sparse=3.0-0, -spatstat.utils=3.0-1, -statnet.common=4.6.0, -stringi=1.7.8, -stringr=1.5.0, -survival=3.4-0, -svglite=2.1.0, -sys=3.4.1, -systemfonts=1.0.4, -tensor=1.5, -testthat=3.1.6, -tibble=3.1.8, -tidyr=1.2.1, -tidyselect=1.2.0, -tidyverse=1.3.2, -tinytex=0.42, -tzdb=0.3.0, -usethis=2.1.5, -utf8=1.2.2, -uuid=1.1-0, -uwot=0.1.14, -vctrs=0.5.1, -viridis=0.6.2, -viridisLite=0.4.1, -visdat=0.5.3, -vroom=1.6.0, -waldo=0.4.0, -webshot=0.5.4, -whisker=0.4, -withr=2.5.0, -xfun=0.35, -xml2=1.3.3, -xopen=1.0.0, -xtable=1.8-4, -yaml=2.3.6, -yulab.utils=0.0.5, -zip=2.2.0, -zlibbioc=1.42.0, -zoo=1.8-11

For manuscripts utilizing custom algorithms or software that are central to the research but not yet described in published literature, software must be made available to editors and reviewers. We strongly encourage code deposition in a community repository (e.g. GitHub). See the Nature Portfolio underlined:guidelines for submitting code & software for further information.

# Data

Policy information about availability of data

All manuscripts must include a data availability statement. This statement should provide the following information, where applicable:

- Accession codes, unique identifiers, or web links for publicly available datasets
- A description of any restrictions on data availability
- For clinical datasets or third party data, please ensure that the statement adheres to our policy

Sequencing data generated in this study have been deposited in NCBI's Gene Expression Omnibus (GEO) and can be accessed through the GEO SuperSeries accession number GSE240020. Accession numbers for publicly available raw count data that have been re-analyzed for this study are listed here: (i) [Mouse] > Kernfeld et al. (2018) [GSE107910]; Bornstein et al. (2018) [GSE103967]; Dhalla et al. (2019) [https://www.ebi.ac.uk/biostudies/arrayexpress/studies/E-MTAB-8105#]; Baran-Gale et al. (2020) [https://bioconductor.org/packages/release/data/experiment/html/MouseThymusAgeing.html]; Wells et al. (2020)

[GSE137699]; Rota et al. (2021) [GSE162668]; Nusser et al. (2022) [GSE106856]; Michelson et al. (2022) [GSE194253]; Klein et al. (2023) [GSE215418]; Farley et al. (2023) [GSE232765]; Givony et al. (2023) [GSE236075]; Michelson et al. (2023) [GSE225661]; Horie et al. (2023) [GSE228198] and (ii) [Human] > Park et al. (2020) [https://zenodo.org/records/3711134]; Bautista et al. (2021) [GSE147520]; Ragazzini et al. (2023) [GSE220830, GSE220206, GSE220829]. The re-analyzed public datasets with added metadata can be accessed at 10.5281/zenodo.12516405.

## Research involving human participants, their data, or biological material

Policy information about studies with [human participants or human data](). See also policy information about [sex, gender (identity/presentation), and sexual orientation]() and [race, ethnicity and racism]().

| | |
|---|---|
| Reporting on sex and gender | N/A |
| Reporting on race, ethnicity, or other socially relevant groupings | N/A |
| Population characteristics | N/A |
| Recruitment | N/A |
| Ethics oversight | N/A |

Note that full information on the approval of the study protocol must also be provided in the manuscript.

# Field-specific reporting

Please select the one below that is the best fit for your research. If you are not sure, read the appropriate sections before making your selection.

☒ Life sciences ☐ Behavioural & social sciences ☐ Ecological, evolutionary & environmental sciences

For a reference copy of the document with all sections, see [nature.com/documents/nr-reporting-summary-flat.pdf]()

# Life sciences study design

All studies must disclose on these points even when the disclosure is negative.

| | |
|---|---|
| Sample size | Sample sizes were based on previously published work and preliminary studies, the effect size (difference between groups/standard deviation) for each study was estimated, and sample sizes were calculated using G*Power (v3.1.9.2) (Faul et al. 2009 Behavior Res Meth. 41:1149-1160) for an alpha=0.05 and beta=0.8. |
| Data exclusions | No samples were excluded from scRNA-seq, HALO or flow cytometry datasets. <br><br> scRNA-seq cell exclusions: Quality of the single cells was computationally assessed based on total counts, number of genes, mitochondrial and ribosomal fraction per cell, with low total counts, low number of genes ($\leq$1000) and high mitochondrial content ($\geq$0.2) as negative indicators of cell quality. Cells characterized by more than one negative indicator were considered as "bad" quality cells. Although cells were negatively sorted prior to sequencing for the CD45 marker, a small amount of CD45+ cells (expressing Ptprc), and also a few parathyroid cells (expressing Gcm2), were detected within our dataset. To remove bad quality cells and contaminants in an unbiased way, we assessed them in a cluster basis rather than individually. Leiden clusters with a "bad" quality profile and/or a high number of contaminating cells were removed. Finally, cells marked as doublets by scrublet were also filtered out. |
| Replication | In general variation within groups and between experiments was low, however, to take into account inter-experimental variation all experiments were performed at least twice and in no instance were experiments discarded due to conflicting findings. To account for intra-experimental variation, particularly for in vitro studies, several wells per conditions were assessed with primary sample material coming from at least two different mice. In vitro experiments were similarly independently repeated at least three times. |
| Randomization | No specific method of randomization was used. However, for all experiments, mouse cages were randomly allocated to each group after balancing of age and sex. |
| Blinding | Due to practical considerations in experimental design and the fact that aged mice are typically easy to distinguish from young mice, the performing investigator was not typically blinded to the group allocation during the experiments. |

# Reporting for specific materials, systems and methods

We require information from authors about some types of materials, experimental systems and methods used in many studies. Here, indicate whether each material, system or method listed is relevant to your study. If you are not sure if a list item applies to your research, read the appropriate section before selecting a response.

## Materials & experimental systems

| n/a | Involved in the study |
|---|---|
| ☐ | ☒ Antibodies |
| ☐ | ☒ Eukaryotic cell lines |
| ☒ | ☐ Palaeontology and archaeology |
| ☐ | ☒ Animals and other organisms |
| ☒ | ☐ Clinical data |
| ☒ | ☐ Dual use research of concern |
| ☒ | ☐ Plants |

## Methods

| n/a | Involved in the study |
|---|---|
| ☒ | ☐ ChIP-seq |
| ☐ | ☒ Flow cytometry |
| ☒ | ☐ MRI-based neuroimaging |

# Antibodies

| | |
|---|---|
| Antibodies used | The antibodies used were rabbit polyclonal anti-mouse Keratin 14 (905301, Biolegend), rat anti-mouse Keratin 8 (Troma-I; Developmental Studies Hybridoma Bank, Iowa City), and rabbit anti-mouse NKp46 (ab233558, Abcam). Secondary antibodies used were Opal 690 anti-rabbit (Akoya), Opal 520 anti-rat (Immpress), and Opal 570 anti-NKp46 (Akoya). For flow cytometry and cell sorting, surface antibodies against CD45 (565967, BD Biosciences; 103154 BioLegend ), CD31 (102434, Biolegend), CD140a (135907, Biolegend), MHC-II (107620, Biolegend), EpCAM (46-5791-82, BD Biosciences), Ly-51 (740882, BD Biosciences), UEA-1 (ZC0426, Vector Labs), DLCK-1 (NBP1-77127F, Novus Biologicals), Ly6D (138605, Biolegend), CD104 (123615, Biolegend), CD140a (135921, Biolegend), CD31 (102427, Biolegend), PDPN (127425, Biolegend), CD8a (100714, Biolegend), CD4 (565709, BD Biosciences), TCR-β (109239, Biolegend), CD3e (100232, Biolegend), CD25 (102030, Biolegend), CD44 (612799, BD Biosciences), NK1.1 (108753, Biolegend), CD49b (561067, BD Biosciences), c-Kit (105811, Biolegend), TCR-γδ (118107, Biolegend), CD1d PBS-57 Tetramer (NIH tetramer Core), CD11c (35-0114, Tonbo), CD11c (612796, BD Biosciences), CD11b (741722, BD Biosciences), XCR1 (148225, Biolegend), B220 (103232, Biolegend), CD127 (50-1271, Tonbo), Sca-1 (122527, Biolegend), CD135 (135305, Biolegend), CD150 (46-1502-82, eBioscience), CD48 (103427, Biolegend), NKG2D (562800, BD Biosciences), CD49a (741976, BD Biosciences), KLRG1 (138425, Biolegend), CCR-6 (129814, Biolegend), IL-23R (150907, Biolegend), ST2 (566310, BD Biosciences), H2-kb (116525, Biolegend), IL-18R (25-5183-82, Thermo Fisher), IL-18R (25-5183-82, Thermofisher) were used. Following fixation and permeabilization (554714, BD Biosciences), cells were stained with the following antibodies: Perforin (154315, Biolegend), Granzyme B (MHGB04, Thermofisher). Annexin V and 7-AAD staining (640920, BioLegend) was performed in Annexin V binding buffer (422201, BioLegend). A detailed table outlining specific vendors, fluorochromes, catalog numbers, lot numbers and dilutions has been included in a separate data table. |
| Validation | Isotype, fluorescent-minus one, and genetic knockout (when appropriate) controls were used to set gates. |

# Eukaryotic cell lines

Policy information about cell lines and Sex and Gender in Research

| | |
|---|---|
| Cell line source(s) | OP9-DLL1 and RMA(S) |
| Authentication | N/A |
| Mycoplasma contamination | N/A |
| Commonly misidentified lines (See ICLAC register) | N/A |

# Animals and other research organisms

Policy information about studies involving animals; ARRIVE guidelines recommended for reporting animal research, and Sex and Gender in Research

| | |
|---|---|
| Laboratory animals | Inbred male and female C57Bl/6J (000664) and B6 CD45.1 (002014) mice were obtained from Jackson Laboratories (Bar Harbor, USA). Il1r1-/- (003245), Il18-/- (004130), Il18r1-/- (004131), Casp1Δ10 (032662), CD1d-/- (008881), Ifngr-/- (003288), Prf1-/- (002407), Rag2-eGFP (005688) and GREAT ("interferon-gamma reporter with endogenous polyA transcript") mice were obtained from Jackson Laboratories and bred in house. Gzmb-/- mice were obtained from Geoffrey Hill (Fred Hutchinson Cancer Center) and bred in house. Il18 flox mice (Il18fl/fl) were obtained from R. Nowarski (Harvard Medical School) and R. Flavell (Yale School of Medicine) and crossed in house to Zbtb46 Cre+ obtained from Jackson Laboratories (032662) to generate Il18fl/fl Zbtb Cre+ (Il18ΔDC) mice. Il18r1fl/fl mice were obtained from G. Trinchieri (NCI) and crossed to Foxn1 Cre+ mice obtained from Jackson Laboratories (018448) and Ncr1 Cre+ obtained from K. Barry (Fred Hutch Cancer Center) to generate Il18r1fl/fl Foxn1 Cre+ (Il18r1ΔTEC) and Il18r1fl/fl Ncr1 Cre+ (Il18r1ΔNK), respectively. Ifngrfl/fl (025394) and Foxn1 Cre+ (018448) mice were obtained from Jackson Laboratories and crossed to generate Ifngrfl/fl Foxn1 Cre+ (IfngrΔTEC) mice. All experimental mice were between 6-10 weeks old. Animal rooms were kept in a constant temperature of 72 degrees (+/- 3 degrees) Fahrenheit with humidity set between 30%-70% as recommended in the "Guide for the Care and Use of Lab Animals". Light cycles were set for 12 hours on and 12 hours off. |
| Wild animals | No wild animals were used in this study. |
| Reporting on sex | All mouse ages and sexes are reported in figure legends |

| Field-collected samples | No field collected samples were used in this study. |
|---|---|
| Ethics oversight | All studies were performed under approved Institutional Animal Care and Use Committee (or equivalent) protocols at Fred Hutchinson Cancer Center and Memorial Sloan Kettering Cancer Center. |

Note that full information on the approval of the study protocol must also be provided in the manuscript.

# Flow Cytometry

## Plots

Confirm that:

☒ The axis labels state the marker and fluorochrome used (e.g. CD4-FITC).

☒ The axis scales are clearly visible. Include numbers along axes only for bottom left plot of group (a 'group' is an analysis of identical markers).

☒ All plots are contour plots with outliers or pseudocolor plots.

☒ A numerical value for number of cells or percentage (with statistics) is provided.

## Methodology

| Sample preparation | Thymus was enzymatically digested following and adapted protocol. Briefly, thymi were mechanically dissociated into 1-2 mm pieces. Tissue pieces were incubated with a digestion buffer (either, RPMI with 10% FCS, 62.5 um/mL liberase TM, 0.4 mg/ml DNase I; or RPMI with 25mM HEPES, 20µg/µL DNAse1, and 1mg/mL Collagenase/Dispase). Between incubation steps, supernatant containing dissociated cells were transferred to tubes equipped with 100 um filter. Cells were pelleted by centrifugation at 400g for 5 min. All steps were performed at 4ºC unless indicated. For sequencing experiments, cell pellets were incubated with anti-mouse CD45 microbeads and CD45+ cells were depleted from cell suspension using magnetic-associated cell sorting (MACS) on LS columns according to manufacturer's protocol. Following red blood cell lysis using ACK buffer, the CD45-depleted cell fraction was incubated with an antibody cocktail for 15 min at 4 C and cells of interest were purified by fluorescent-associated cell sorting (FACS) on a BD Biosciences Aria II using a 100 micro meter nozzle. Cells were sorted into tubes containing RPMI supplemented with 2% BSA. FACS-purified cells were spun down at 400g for 5 min and resuspended in PBS supplemented with 0.04 % BSA for generation of single-cell suspensions. Isolation of thymus NK and CD45- cells for ex vivo cytotoxicity assays is detailed in figure legends. |
|---|---|
| Instrument | Cells sorting was performed on a BD Biosciences Aria II or S6 using a 100 um or 70um nozzle. Flow cytometry analysis was performed on BD Symphony  A6. |
| Software | Flow cytometry was analyzed using FlowJo (Treestar Software). |
| Cell population abundance | For sort experiments, cell purity was typically >99% as checked using a post-sort purity check by flow cytometry. |
| Gating strategy | All gating have been outlined in the manuscript. |

☒ Tick this box to confirm that a figure exemplifying the gating strategy is provided in the Supplementary Information.

