## [Peer Review File · Nature Immunology]

Damage-induced IL-18 stimulates thymic NK Cells limiting endogenous tissue regeneration

Corresponding Author: Dr Jarrod Dudakov

Version 0:

Decision Letter:

2nd Apr 2024

Dear Dr Dudakov,

Your Article, "Damage-induced IL-18 stimulates thymic NK Cells limiting endogenous tissue regeneration" has now been seen by 2 referees. You will see from their comments copied below that while they find your work of considerable potential interest, they have raised quite substantial concerns that must be addressed. In light of these comments, we cannot accept the manuscript for publication, but would be very interested in considering a revised version that addresses these serious concerns.

We hope you will find the referees' comments useful as you decide how to proceed. If you wish to submit a substantially revised manuscript, please bear in mind that we will be reluctant to approach the referees again in the absence of major revisions.

We hope that you will find the prioritised set of referee points to be useful when revising your study. Please do not hesitate to get in touch if you would like to discuss these issues further.

Please ensure to address reviewers comments on IL-18 producing cells, the target of IL-18 and the mechanism by which NK cells limit thymus regeneration.

If you choose to revise your manuscript taking into account all reviewer and editor comments, please highlight all changes in the manuscript text file [OPTIONAL: in Microsoft Word format].

* If you have not done so already please begin to revise your manuscript so that it conforms to our Article format instructions at <http://www.nature.com/nl/authors/index.html>. Refer also to any guidelines provided in this letter.

The Reporting Summary can be found here:

When submitting the revised version of your manuscript, please pay close attention to our

<https://www.nature.com/nature-portfolio/editorial-policies/image-integrity>>Digital Image Integrity Guidelines. and

to the following points below:

Link Redacted

If you wish to submit a suitably revised manuscript we would hope to receive it within 6 months. If you cannot send it within this time, please let us know. We will be happy to consider your revision so long as nothing similar has been accepted for publication at Nature Immunology or published elsewhere.

Nature Immunology is committed to improving transparency in authorship. As part of our efforts in this direction, we are now requesting that all authors identified as 'corresponding author' on published papers create and link their Open Researcher and Contributor Identifier (ORCID) with their account on the Manuscript Tracking System (MTS), prior to acceptance. ORCID helps the scientific community achieve unambiguous attribution of all scholarly contributions. You can create and link your ORCID from the home page of the MTS by clicking on 'Modify my Springer Nature account'. For more information please visit please visit www.springernature.com/orcid.

Thank you for the opportunity to review your work.

Sincerely,

Stephanie Houston, PhD
Senior Editor
Nature Immunology

Reviewers' Comments:

Reviewer #1:

Remarks to the Author:

This manuscript examines complex mechanisms that control the thymus regeneration, predominantly following damage induced by sublethal irradiation (SI) or following lethal irradiation and stem cell transplant (SCT). The authors point out the relevance of thymus regeneration in promoting T-cell recovery in several settings, including cancer treatment.

The strength of the study is extensive use of well defined (SLI and SCT) models of thymus regeneration together with a wide range of genetically altered mouse models. From their work, the authors propose a model in which thymus damage promotes IL18 production by non-epithelial stroma, which stimulates activation of NK cells that target thymic epithelial cells to prevent regeneration.

A limitation is that several aspects of this proposed model are only partly supported by the presented data. These include clarity on the nature of IL18 producing cells, the specific cell target of IL18, and perhaps most important, the mechanism of action by which NK cells limit thymus regeneration. Specific comments below:

Producers of IL18

1. Data in Fig 2 shows analysis of cell types that produce IL18 at both steady state and following SI. This needs to be clearer. For example, how are all the stromal cell types in Fig2b defined? The legend to Fig 2 states panels a-b show data from thymus at baseline and 1, 4, 7 days following SI, but it is not clear how the presented data relates to this timecourse. Also, is the RNAseq data on IL18 expression verified by another assay eg qPCR?
2. Fig 2c shows that non-epithelial stroma contains approximately 30% cl-Cas1+ cells, which links to identification of the target of IL18. What are these cl-Cas1+ non-epithelial cells? As the authors show that non-epithelial cells are highly heterogeneous (various fibroblasts, various endothelial subsets), can the authors examine whether this cl-Cas1+ population maps to a particular subset?

Experiments on the target of IL18.

1. The authors propose NK cells are the major target of IL18, but the experiments performed do not allow for specific targeting of NK cells, so this key point remains unclear. First, use of anti-NK1.1 depletion (Fig 4) will target several cell types including NK and NKT. Second, there needs to be clarity on whether NK cells described here are NK cells or ILC1, which both belong to group 1 ILC. As NKp46 is expressed by NK and multiple ILC subsets, including ILC1 and ILC3 (eg Meininger et al Trends Imm 2020 41:10) using NCR1Cre to delete IL18R (Fig 4) will target multiple cell types.
2. How are NK and NKT1, NKT 2 (Fig 3) defined in this study? Also, clear heterogeneity in IL18R expression is seen in NK, NKT1 and NKT2 (Fig. 3), but this is not explored. Further development of this is important. For example, when NK and NKT are examined post-damage (Fig 4c, d), is it the IL18R+ subsets of these cells that preferentially survive? Alongside analysis of NK and NKT, data should be included on ILC numbers in Figure 4c, d, given their expression of IL18R.
3. In experiments in Fig 4i, where control and Ncr1Cre/IL18R floxed mice are irradiated, why is addition of IL18 included in this experiment? Presumably, compared to control mice, Ncr1Cre/IL18R floxed mice recover better without the need for addition of exogenous IL18? This experiment should be included.
4. Fig S4d shows thymus recovery at day 7 following SI in Cd1d KO mice is not impaired. This contrast to another study (ref 11) which showed reduced thymus recovery of Cd1d KO mice at day 35 following SI. Inclusion of additional experiments at this later time point would help in clarifying any possible role for NKT-cells.

Mechanisms of action of NK cells.

1. Fig 7 shows that MHC I is downregulated on both cTEC and mTEC after damage, while Rae1 is upregulated, and this is taken as evidence that TEC become NK targets after damage. Are these changes in MHC I and Rae1 specific to TEC, or do other thymic stromal populations (eg the ones identified in Fig 2 – various fibroblasts: mesothelial cells, intermediate fibroblasts, capsular fibroblasts, and various endothelial subtypes) also show changes? A detailed analysis of stromal cell expression of MHC I and Rae1 is required to put the data on TEC into context.
2. In Figure 5e, the authors show that NK cells target RMA-S cells, but the relevance of this to TEC as an NK target is not clear. Can the authors show that thymic NK cells from SI mice kill TEC from SI mice? Here, a comparison of NK targeting of TEC from control and SI mice is important, as it is suggested that changes in MHC I and Rae1 expression in TEC of SI mice (see point 4 below) renders them more susceptible to NK targeting.
3. In Fig 6i, perforin KO mice are shown to have improved thymus regeneration. To further support the proposed model, that NK cells target TEC and limit regeneration, inclusion of data on cell death within TEC is needed. Here, compared to controls, there should be less cell death detected within TEC in perforin KO mice after SI.
4. Fig 7c, d shows that absence of IL18, or anti NK1.1 treatment, enhances recovery of mTEC but not cTEC. This seems at odds with data in Fig 7a, where MHC I and Rae1 changes following SI are detectable in both mTEC and cTEC (Fig 7a), which would indicate that both mTEC and cTEC are NK targets. If cTEC are not a target of NK cells, and cTEC are the key regulators of CD4-8- and CD4+8+ thymocyte populations, why then is thymocyte development so much affected following SI?
5. In Fig 1f-i, improvements in thymus regeneration as indicated by total thymus cellularity are shown when IL18 function is perturbed (eg IL18 KO mice, IL18R KO mice, anti-IL18 injection). Do these improvements in thymus cellularity relate to increases in TEC numbers, and reduced cell death in TEC? If this could be shown, then it would fit very well with the proposed model.

Minor points

1. Fig 1H shows that recombinant IL18 impairs regeneration following SI, while anti-IL18 improves regeneration following SCT. Does anti-IL18 also improve regeneration following SI?
2. Fig 1F shows thymus cellularity of WT and various KO mice 7 days after SI. Thymus cellularities at day 0 (ie pre SI) should also be included to ensure a similar starting point.

Reviewer #2:

Remarks to the Author:

In this study, the authors report that inflammatory IL-18 produced following thymic tissue damage impairs thymic regeneration. They used a combination of mouse models and cell culture assays to demonstrate the origin of IL-18 secretion and showed that it induced a cytotoxic response by resident NK cells that limits tissue regeneration. Finally, they identified mTECs as potential targets of this cytotoxic response. The manuscript is well written, the methods are appropriate, and the work is significant as there is a need to improve strategies for thymus regeneration. The study also has implications when considering IL-18 as an immunotherapeutic agent for cancer.

One of the weaknesses of the study is the lack of data on spatial distribution of the different cell types involved in this pathway (IL-18 producing cells, cytotoxic NK cells, and TECs targeted by NK cells). For example, are the stromal cells and dendritic cells that produce IL-18 near IL18R+ NK cells? Where are the granzyme+ /perforin+ NK cells located in the thymus (cortex vs medulla)? Demonstrating proximity between cytotoxic NK cells and dying TECs would help support the conclusions.

Other comments:

Do IL18R-negative NK cells also upregulate granzyme and perforin following injury?

Do thymic NK cells in IL18R1-delta-NK mice upregulate granzyme and perforin after injury?

Fig. 6D – why is basal expression of IFN and perforin much higher than in Fig. 5B? Which cell population is shown?

It would be helpful to have a better phenotyping of MHC and Rae1 expression in TECs. For example, is expression changed in all subsets of mTECs (Aire+, Ccl21+, Dclk1+, etc.)?

Version 1:

Decision Letter:

2nd May 2025

Dear Dr Dudakov,

Thank you for providing your response to reviewers concerns on your article "Damage-induced IL-18 stimulates thymic NK Cells limiting endogenous tissue regeneration". We are very interested in the possibility of publishing your study in Nature Immunology and suggest that reviewers concerns over the presentation of ex vivo data analyzing cytotoxic capacity of thymic NK cells should be addressed and additional discussion of data which suggests other hematopoietic cells may produce IL-18 should be added. We agree that experiments for in situ assessment of TEC cell death, BM chimeras and anti-IL18 in sublethal TBI model would not change the overall message of the study and therefore will not be required for resubmission.

We therefore invite you to revise your manuscript taking into account all reviewer and editor comments. Please highlight all changes in the manuscript text file in Microsoft Word format.

* If you have not done so already please begin to revise your manuscript so that it conforms to our Article format instructions at <http://www.nature.com/ni/authors/index.html>. Refer also to any guidelines provided in this letter.

* Please include a revised version of any required reporting checklist. It will be available to referees to aid in their evaluation of the manuscript goes back for peer review. They are available here:

Reporting summary:

When submitting the revised version of your manuscript, please pay close attention to our [href="https://www.nature.com/nature-portfolio/editorial-policies/image-integrity">Digital Image Integrity Guidelines.](https://www.nature.com/nature-portfolio/editorial-policies/image-integrity) and to the following points below:

Please note, Extended Data figures and tables are online-only (appearing in the online PDF and full-text HTML version of the paper), peer-reviewed display items that provide essential background to the Article but are not included in the printed version of the paper due to space constraints or being of interest only to a few specialists. A maximum of ten Extended Data display items (figures and tables) is typically permitted. When re-submitting your manuscript, please ensure that any supplementary figures and tables that are more critical to the manuscript's conclusions are converted to Extended data to increase these data's visibility.

Link Redacted

We hope to receive your revised manuscript within two weeks. If you cannot send it within this time, please let us know. We will be happy to consider your revision so long as nothing similar has been accepted for publication at Nature Immunology or published elsewhere.

Nature Immunology is committed to improving transparency in authorship. As part of our efforts in this direction, we are now requesting that all authors identified as 'corresponding author' on published papers create and link their Open Researcher and Contributor Identifier (ORCID) with their account on the Manuscript Tracking System (MTS), prior to acceptance. ORCID helps the scientific community achieve unambiguous attribution of all scholarly contributions. You can create and link your ORCID from the home page of the MTS by clicking on 'Modify my Springer Nature account'. For more information please visit please visit www.springernature.com/orcid.

Sincerely,

Stephanie Houston, PhD
Senior Editor
Nature Immunology

Reviewers' Comments:

Reviewer #2 (Remarks to the Author):

Thank you to the authors for addressing the concerns raised through additional experiments. While I am generally satisfied by the responses, I have comments on the new data on NK cell killing of TECs (Figure 7C and 7D). Why are the majority of the cells staining positive for 7AAD and Annexin V, even in the absence of NK cells? Is the data from this assay reliable if performed with cells that are mostly dead to start with? An alternative assay such as performing histology to look for apoptosis in TECs while co-staining with an NK cell marker would help support the conclusions. I would also recommend making the flow plots in extended Figure 3 bigger and modifying either the axis of the plots or the table to harmonize the names of the markers used.

Reviewer #3 (Remarks to the Author):

The manuscript presents interesting findings, effectively combining a range of germline and conditionally gene-deleted mouse models, along with antibody-mediated blocking strategies, to support its conclusions. This study demonstrates that acute thymic injury triggers caspase-1 activation and subsequent IL-18 production, which impairs thymus regeneration. IL-18 is produced by both hematopoietic and non-hematopoietic cells in response to damage. IL-18 receptor-expressing thymic NK cells are activated following injury and play a suppressive role in thymic repair. IL-18 stimulation enhances the cytotoxic activity of thymic NK cells, leading to aberrant targeting of thymic epithelial cells (TECs), thereby further hindering thymus regeneration.

A limitation of the study is the relatively modest and variable phenotype observed in thymic cellularity following damage, which necessitates the use of a large number of mice to support conclusions based on differences that are not particularly striking. That said, all the data point in the same direction, supporting the overall message of the study.

In my view, a major point of concern is the presentation of ex vivo data analyzing the cytotoxic capacity of thymic NK cells harvested from mice following SL-TBI. These experiments, involving targets such as RMA-S cells or thymic epithelial cells (TECs), are central to the authors' conclusions. Therefore, the clarity of their presentation and their capacity to convince (Fig 5F, Fig 6I, Fig 7E) should be improved to strengthen the study.

There are additional comments that should be addressed:

- Fig. 1F and 1I: IL-18^{-/-} mice display increased thymic cellularity at baseline (Extended Fig. 1B), which complicates the interpretation of data at day 7 post-SL-TBI. The use of anti-IL-18 treatment in Fig. 1I provides a valuable alternative approach to support the role of IL-18 in thymus regeneration. However, the experimental setting in Fig. 1I is markedly different and not directly comparable. To strengthen the conclusions, an experiment using anti-IL-18 antibody with analysis performed 7 days post-SL-TBI should be included.

-Fig 2E: The contribution of hematopoietic cells to IL-18 production following SL-TBI is explored in Fig. 2I using mice with IL-18-deficient dendritic cells. However, the absence of significant differences between IL-18^{WT} mice and those lacking IL-18 in dendritic cells suggests that additional hematopoietic cell types may be involved. To further support the conclusion that hematopoietic cells contribute to IL-18 production, including a control group in which IL-18^{-/-} bone marrow cells are transferred into WT recipients would support the overall argument.

-Fig 4I and 4J: In contrast to the previous experiments, the protocol used in these two figures includes recombinant IL-18 (rec IL-18) injections administered on day 3 post-TBI. What is the rationale for modifying the protocol to include IL-18 administration after TBI? Is the contribution of NKp46⁺ cells observable only in the context of this non-physiological presence of IL-18? It would be important to show the results of experiments assessing thymic cellularity 7 days post-SL-TBI without IL-18 injection to clarify this point.

-Fig 4J: It remains unclear whether the recruitment or increased number of NK1.1⁺ cells in the thymus following SL-TBI is

dependent on IL-18 signaling. Specifically, do NK1.1⁺ cell numbers increase in the thymus of IL-18^{-/-} or IL-18R1-deficient mice post-TBI? Addressing this point would clarify whether IL-18 is required for the accumulation or expansion of NK cells in the thymus in response to damage.

-Fig 5F. Presenting target cell killing as relative values is not very convincing. The figure should instead display non-normalized data as the percentage of specific lysis. Additionally, including a representative FACS plot of the co-culture after 5 hours would provide a clearer picture of killing efficiency. Furthermore, according to the Materials and Methods, splenic NK cells from mice treated with poly(I:C) were used as a positive control. This control should be shown in the figure to allow proper evaluation of the thymic NK cells' capacity to kill target cells.

-Lines 287-297 suggest that the increase in NK cell numbers in the thymus is due to cell expansion. However, the possibility that NK cells may be recruited from the periphery has not been addressed. This potential mechanism should be discussed

- Fig. 6H: The IFN- γ plots show only biological duplicates. To ensure statistical validity, the experiments should be reproduced with additional replicates.

- Fig. 6I: The same comment applies as for Fig. 5F. Presenting target cell killing as relative values is not very convincing. The figure should instead display non-normalized data as the percentage of specific lysis. Additionally, including a representative FACS plot of the co-culture after 5 hours would provide a clearer picture of the killing efficiency.

-Lines 326-328: Given the variability of the model, the conclusion that "these data support IL-18 triggering NK perforin upregulation/release more than granzyme B" should be either mitigated or described with greater caution.

-Fig. 7: Downmodulation of MHC I serves as a signal to trigger NK cell cytotoxicity in the context of activating ligand expression. In addition to evaluating MHC I expression, the expression of NK cell activating ligands, such as Rae1, should also be assessed on TECs.

Minor comments:

- Line 224: NK cells repeated twice

- Line 257 refers to an increased expression of NKG2D at the protein level following TBI; however, this data is not shown.

- Line 261 incorrectly references Extended Fig. 5F for the annotation of Itga1 and Itga2 in the scRNA-seq data. These annotations are actually shown in Extended Fig. 6A and should be correctly cited to avoid confusion.

Version 2:

Decision Letter:

Our ref: NI-A37414B

2nd Jul 2025

Dear Dr. Dudakov,

Thank you for submitting your revised manuscript "Damage-induced IL-18 stimulates thymic NK Cells limiting endogenous tissue regeneration" (NI-A37414B). It has now been seen by the original referees and their comments are below. The reviewers find that the paper has improved in revision, and therefore we'll be happy in principle to publish it in Nature Immunology, pending minor revisions to satisfy the referees' final requests and to comply with our editorial and formatting guidelines.

We will now perform detailed checks on your paper and will send you a checklist detailing our editorial and formatting requirements in about a week. Please do not upload the final materials and make any revisions until you receive this additional information from us.

If you had not uploaded a Word file for the current version of the manuscript, we will need one before beginning the editing process; please email that to immunology@us.nature.com at your earliest convenience.

Thank you again for your interest in Nature Immunology Please do not hesitate to contact me if you have any questions.

Sincerely,

Stephanie Houston, PhD
Senior Editor
Nature Immunology

Reviewer #2 (Remarks to the Author):

The authors have addressed my concerns.

Reviewer #3 (Remarks to the Author):

The authors have addressed or discussed the concerns raised during the previous round of revision. The manuscript has improved significantly and is now more mature.

We greatly appreciate the extremely helpful comments of both reviewers. New text in the revised manuscript has been colored blue and for ease of identification, new pieces of data in the revised figures have been indicated with a light-yellow box. We have also included new data in Response Figures in-line with the appropriate comment(s) in the text below.

Reviewer #1:

Remarks to the Author:

This manuscript examines complex mechanisms that control the thymus regeneration, predominantly following damage induced by sublethal irradiation (SI) or following lethal irradiation and stem cell transplant (SCT). The authors point out the relevance of thymus regeneration in promoting T-cell recovery in several settings, including cancer treatment.

The strength of the study is extensive use of well defined (SLI and SCT) models of thymus regeneration together with a wide range of genetically altered mouse models. From their work, the authors propose a model in which thymus damage promotes IL18 production by non-epithelial stroma, which stimulates activation of NK cells that target thymic epithelial cells to prevent regeneration.

A limitation is that several aspects of this proposed model are only partly supported by the presented data. These include clarity on the nature of IL18 producing cells, the specific cell target of IL18, and perhaps most important, the mechanism of action by which NK cells limit thymus regeneration. Specific comments below: We appreciate the generally enthusiastic comments about our study and have tried to address all comments and concerns. We would also like to thank the reviewer for insightful comments and suggestions (as well as a very well-organized review) We feel that the extensive new data in our manuscript far better supports the conclusions of our paper.

A. Producers of IL18

1. Data in Fig 2 shows analysis of cell types that produce IL18 at both steady state and following SI. This needs to be clearer. For example, how are all the stromal cell types in Fig2b defined? The legend to Fig 2 states panels a-b show data from thymus at baseline and 1, 4, 7 days following SI, but it is not clear how the presented data relates to this timecourse. Also, is the RNAseq data on IL18 expression verified by another assay eg qPCR?

We apologize for the confusion in how we presented the data. We have now reanalyzed our scRNAseq dataset to interrogate expression of *Il18* across subsets and across timepoint after damage. This analysis demonstrates that transcriptionally *Il18* expression does not change very much after damage and remains restricted to distinct populations (cDC1, Mesothelial, and macrophages being the most prominent. This was also further confirmed using qPCR on selected flow isolated subsets. This is now included as (Revised Fig. 2b-c and Response Fig. 1).

2. Fig 2c shows that non-epithelial stroma contains approximately 30% cl-Cas1+ cells, which links to identification of the target of IL18. What are these cl-Cas1+ non-epithelial cells? As the authors show that non-epithelial cells are highly heterogeneous (various fibroblasts, various endothelial subsets), can the authors examine whether this cl-Cas1+ population maps to a particular subset?

To address this, we have performed additional experiments to interrogate Cas1+ cells in greater detail (Revised Fig. 2d and Response Fig. 1). In particular, given the scRNAseq and PCR data suggesting expression of *Il18* by other stromal cells including macrophages and mesothelial cells, we have also now performed flow cytometry for cleaved caspase 1 to identify these cell populations. Using this approach we

Response Fig. 1

could indeed identifying that Mesothelial cells and Macrophages upregulate cl-Cas1 after damage but other stromal cells such as TECs did not.

B. Experiments on the target of IL18.

Q1. The authors propose NK cells are the major target of IL18, but the experiments performed do not allow for specific targeting of NK cells, so this key point remains unclear. First, use of anti-NK1.1 depletion (Fig 4) will target several cell types including NK and NKT. Second, there needs to be clarity on whether NK cells described here are NK cells or ILC1, which both belong to group 1 ILC. As NKp46 is expressed by NK and multiple ILC subsets, including ILC1 and ILC3 (eg Meininger et al Trends Imm 2020 41:10) using NCR1Cre to delete IL18R (Fig 4) will target multiple cell types.

We agree that the approach we first took did not distinguish ILC1 from NK cells and we perhaps should have been more careful with our language. We have found that distinguishing ILC1 from NK cells was quite difficult in our dataset so have now amended the manuscript to only refer to NK/ILC1 for any data derived from our scRNAseq dataset or where only NKT cells have been excluded. However, we have also now included extensive data looking at the functional distinction between NK and ILC1. Specifically, we now show UMAP overlay for the genes encoding CD49a and CD49b (**S5e and Response Fig. 2**), as well as the discrimination of these populations by flow cytometry (**S3**). While the reviewer correctly points out that Ncr1 driven cre will delete expression of Il-18R in both ILC1 and NK cells, to determine the specific roles of each of these related populations we first took an unbiased approach to determine the cells expressing key effector molecules IFN γ , Perforin and granzyme B after damage. This analysis suggested that while both IFN γ and Perforin were specifically upregulated in NK cells after damage and not by other NK1.1 expressing populations including ILC1, NKT and T cells. However, in the case of IFN γ , NK cells were only a small contributor at baseline and after damage. In contrast, NK cells were the main producer of Perforin in the thymus (**Fig. 5d and Response Fig. 2**). Interestingly, granzyme B was increased in all cells NK1.1⁺ analyzed and was not specific to NK cells. Where possible we have also looked for ILC1 in other analyses through the manuscript (Fig. 6a-b). Using this more refined flow gating, we could also show that NK cells specifically increase expression of IFN γ and Perforin in an IL-18R-dependent fashion (**Fig. 6c-d and Response Fig. 2**). Moreover, the cells used in the NK-cell killing assay were using this refined NK-specific sorting strategy. Together we feel this supports the notion

that NK cells and not ILC1 or NKT cells are the main driver of these responses after damage. While some ILC3 have been reported to express Nkp46 in other tissues, in our previous studies we have failed to identify any Nkp46⁺ ILC3s in the thymus (Dudakov 2012 Science; Dudakov 2017 Blood).

Q2. How are NK and NKT1, NKT 2 (Fig 3) defined in this study? Also, clear heterogeneity in IL18R expression is seen in NK, NKT1 and NKT2 (Fig. 3), but this is not explored. Further development of this is important. For example, when NK and NKT are examined post-damage (Fig 4c, d), is it the IL18R⁺ subsets of these cells that preferentially survive? Alongside analysis of NK and NKT, data should be included on ILC numbers in Figure 4c, d, given their expression of IL18R.

We distinguished NKT1 and NKT based on CD1d PB57 (analogue of α -galactosylceramide) tetramer staining, with NKT1 considered to be tetramer⁺ while also expressing NK1.1 and TCR β , while NKT2 were also NK1.1⁺ and TCR β but were tetramer⁻ (Fig. S3, and Response Fig. 3). To address the question about IL-18R, we have now performed analyses showing the expansion of IL-18R⁺ and IL-18R⁻ NK, ILC1 and NKT cells after damage. We found that that there was no difference in expansion between receptor positive or negative NK or ILC1 cells, and a reduction in IL-18R⁺ NKT cells (Fig. 6a-b, and Response Fig. 3). We also showed that activation of NK/ILC1 cells damage, as measured by expression of the effector molecules such as *Prf1*, *Gzma* and *Gzmb*, was largely restricted to Il18r1⁺ cells. However, we did not detect this correlation in the context of NKT cells. We have now performed functional analysis and show that NK cell IFN γ and Perforin expression is IL-18R dependent. This was not the case for Granzyme B, which was upregulated by NK cells independently of IL-18R (Fig. 6c, and Response Fig. 3). We find reduced NK cell expression of Perforin in mice with germline *Il18r1* deletion, and reduced levels of IFN γ and Perforin in mice with conditional knockout of *Il18r1* in NK/ILC1 cells (Fig. 6d-e, and Response Fig. 3). We feel that together this more refined analysis has led to a far more complete interrogation of the biology of NK/ILC1 and IL-18 signaling.

Q3. In experiments in Fig 4i, where control and Ncr1Cre/IL18R floxed mice are irradiated, why is addition of IL18 included in this experiment? Presumably, compared to control mice, Ncr1Cre/IL18R floxed mice recover better without the need for addition of exogenous IL18? This experiment should be included.

Although we see early responses in Ncr1-cre mice, namely reduced levels of Perforin and IFN γ in the thymus at day 3 (Revised Fig. 6e and Response Fig. 3), this is not translated into improved thymic cellularity at later timepoints unless we add exogenous IL-18 from day 3. For instance, although we have some evidence that there may be an increase in NK cell numbers in Cre⁺ mice at baseline, there was no change in NK cell number between Cre⁺ and Cre⁻ mice at day 3 or 5 (Response Figure 4a). Similarly, while do see a decrease in Perforin by NK cells in Cre⁺ mice at day 3, this effect is gone by day 5 with no addition of IL-18 (Response Figure 4b). We also found no change in levels of Perforin at day 5 in Cre⁺ mice (Response Figure 4c). Our conclusion is therefore, that the cytotoxicity promoting effects of IL-18 on thymus NK cells is most prominent in the first 2-5 days post SL-TB; supported by evidence (and described in a later response) that TEC viability and cellularity in *Il18r1*^{-/-} mice are increased in that early stage after (Fig. 7d-e and Response Figure 4) – note that there is no evidence increased baseline TEC cellularity in *Il18r1*^{-/-} mice (Extended Data Fig. 7c and

Response Fig. 4). So then, the conundrum of why a higher cellularity in *Il18r1^{fl/fl} Ncr1-Cre⁺* mice is not reflected at day 7 remains. One explanation may be that that higher number of NK cells at baseline in *Il18r1^{fl/fl} Ncr1-Cre⁺* vs *Cre⁻* mice mitigate some of the expected differences that we would expect based on our results in *Il18r1^{-/-}* mice. Another explanation may be that NK cell activation is partially IL-18 dependent and while its conditional knockout does affect NK cytotoxicity enough to impair NK cell targeting of TECs in the initial 2-5 days post SL-TBI that it is not enough to capture increased thymus cellularity beyond that. Therefore, while our data strongly suggests that IFN γ and Perforin are largely IL-18 dependent, evidence of IL-18 independent cytotoxicity can also be found in Granzyme B.

It should also be noted that our data also accounts for the possibility that the mechanism of IL-18 in delaying organ regeneration is attributable to its recently identified role (in conjunction with TL1A) of stimulating ILC1 production of GM-CSF and thymus granulopoiesis (*Ruiz Perez, 2024, PMID: 38839915*). Competitive bone marrow transplantations with *Il18r1^{-/-}* and WT marrow (described in Figure 2) addressed the possibility of increased innate thymocyte proliferative capacity that had been described in ST-HSCs (Howard 2021 PMID: 34798063). Thymocytes lacking IL-18R expression showed no greater makeup of the thymus at both short and long-term timepoints. To try and account for other IL-18R expressing cells, we also deleted *Il18r1* using *Lck* as a driver of *cre* expression. This led to deletion of *Il18r1* in all T cell committed thymocytes and their progeny, including mature T cells, NKT, and $\gamma\delta$ T cells, that include other potential NK1.1⁺ and NK1.1⁻ IL-18R expressing cells. Critically, we found no difference in thymus cellularity in these mice, even when giving them exogenous IL-18 in a manner like with the *Ncr1-cre* mice (**Fig. 4i** and **Response Fig. 5**), consistent with this being a non-NKT, non-T cell driven phenomena. Therefore, our best, albeit incomplete, explanation is that differences in baseline NK numbers (for unknown reasons outside the scope of this work) combined with IL-18 independent mechanisms of NK cell activation partially impair our ability to capture the same degree of regeneration seen in *Il18r1^{-/-}*, *Il18^{-/-}* and anti-NK1.1 mAb models. However, we strongly believe that despite this unexplained model, the preponderance of evidence in our manuscript, both new and old, support the model of IL-18 driven NK cell activation and killing of TECs after damage.

Q4. Fig S4d shows thymus recovery at day 7 following SI in *Cd1d* KO mice is not impaired. This contrast to another study (ref 11) which showed reduced thymus recovery of *Cd1d* KO mice at day 35 following SI. Inclusion of additional experiments at this later time point would help in clarifying any possible role for NKT-cells.

While indeed there are previous reports that NKT cells can actually be pro-regenerative and therefore their deletion may be expected to lead to poorer repair (Cosway et al. 2022), in our hands we could detect no difference in thymic regeneration in *CD1d* KO mice, even out to 35 days (**Revised Fig. 4h, Ext Data Fig. 5e,**

and **Response Fig. 5**). The lack of impact on regeneration may be due to the background of mice as NKT cells may play a greater role in regeneration in a BALB./c background (personal communication with G. Anderson). Similarly, new data we have also included suggest that deletion of *Il18r* on thymocytes (using *Lck-cre*, which would also delete *IL-18R* on T cells and NKT cells) showed no difference after damage (**Revised Fig. 4i**, and **Response Fig. 5**).

C. Mechanisms of action of NK cells.

C.1. Fig 7 shows that MHC I is downregulated on both cTEC and mTEC after damage, while *Rae1* is upregulated, and this is taken as evidence that TEC become NK targets after damage. Are these changes in MHC I and *Rae1* specific to TEC, or do other thymic stromal populations (eg the ones identified in Fig 2 – various fibroblasts: mesothelial cells, intermediate fibroblasts, capsular fibroblasts, and various endothelial subtypes) also show changes? A detailed analysis of stromal cell expression of MHC I and *Rae1* is required to put the data on TEC into context.

To address this comment, we performed a comprehensive analysis of MHC I expression on distinct TEC subsets. Using our scRNAseq datasets, we now include data that demonstrate TECs as the only populations to show broadly decreased expression after damage (**Fig. 7a** and **Response Fig. 6**). We have also now shown this in detailed flow cytometry data demonstrating all TEC subsets (except for Tuft cells) but not

endothelial, mesothelial, or fibroblasts decrease their expression (Fig. 7b and Response Fig. 6). Interestingly, given the increase in IFN γ it may be expected that there would be an increase in MHCII expression, and indeed when we look at CD45+ subsets we do in fact find an increase in MHCII in Macrophages, cDC1 and some thymocyte subsets (Figure 7b, Extended Data Fig. 7a and Response Fig. 6). In follow-up experiments Rae-1 proved to be challenging to capture by flow cytometry due to inconsistent staining. Therefore we have elected to omit the original Rae1 data but to instead focus on a more detailed subset expression of MHCII.

C.2. In Figure 5e, the authors show that NK cells target RMA-S cells, but the relevance of this to TEC as an NK target is not clear. Can the authors show that thymic NK cells from SI mice kill TEC from SI mice? Here, a comparison of NK targeting of TEC from control and SI mice is important, as it is suggested that changes in MHCII and Rae1 expression in TEC of SI mice (see point 4 below) renders them more susceptible to NK targeting.

C.3. In Fig 6i, perforin KO mice are shown to have improved thymus regeneration. To further support the proposed model, that NK cells target TEC and limit regeneration, inclusion of data on cell death within TEC is needed. Here, compared to controls, there should be less cell death detected within TEC in perforin KO mice after SI.

C.4. Fig 7c, d shows that absence of IL18, or anti NK1.1 treatment, enhances recovery of mTEC but not cTEC. This seems at odds with data in Fig 7a, where MHCII and Rae1 changes following SI are detectable in both mTEC and cTEC (Fig 7a), which would indicate that both mTEC and cTEC are NK targets. If cTEC are not a target of NK cells, and cTEC are the key regulators of CD4-8- and CD4+8+ thymocyte populations, why then is thymocyte development so much affected following SI?

C.5. In Fig 1f-i, improvements in thymus regeneration as indicated by total thymus cellularity are shown when IL18 function is perturbed (eg IL18 KO mice, IL18R KO mice, anti-IL18 injection). Do these improvements in thymus cellularity relate to increases in TEC numbers, and reduced cell death in TEC? If this could be shown, then it would fit very well with the proposed model.

We appreciate this important point and have endeavored to be clearer in what we think is happening and provide more evidence. To that end, and as described above, we now provide a more comprehensive analysis of MHCII expression on TECs and other subsets in the thymus using both transcriptomic and flow cytometric data. This approach shows that there is a specific decrease in MHCII on most TEC subsets, but not on other cells, including fibroblasts, endothelial cells and thymocytes (Fig. 7a-b and Response Fig. 6). In fact, perhaps not surprisingly given the increase in IFN γ , there is actually an increase in MHCII expression on DCs and thymocytes (Fig. S7a and Response Fig. 6). We have also now provided considerably more evidence that TECs are the targets of NK cell killing after damage. Now we show that not only are NK cells functionally more able to kill a target cell after damage (in the original submission and Fig. 5f) but have also performed a similar

assay where we can show that TECs isolated early after damage can be targeted by activated NK cells (**Fig. 7c** and **Response Fig. 7**). In addition, concentrating on the *Il18r1* deficient mouse model (which was the most repeatable of the knockout models we used), we found that there is an increase in TEC viability and numbers early after damage, suggesting increased sparing (**Fig. 7d-e**, and **Response Fig. 7**). Finally, we have now also performed stronger imaging analysis suggesting that the distance between NK cells and both cTECs and mTECs are decreased early after damage (**Fig. 7f** and **Response Fig. 7**). Though in our original submission we posited that this NK cell targeting was specific to mTECs. These data suggest that NK cells target mTECs and cTECs alike but, critically, do not target non-TECs and we have now reframed the manuscript accordingly.

Minor points

1. Fig 1H shows that recombinant IL18 impairs regeneration following SI, while anti-IL18 improves regeneration following SCT. Does anti-IL18 also improve regeneration following SI?

This is a very interesting point and something that we are planning to actively pursue in the lab as a potential therapeutic strategy. However, since we largely focused on the mechanistic aspects of this phenomena in this manuscript, we have not rigorously performed those studies yet. Therefore, although exciting studies that are well worth doing, those studies are beyond the scope of this current manuscript.

2. Fig 1F shows thymus cellularity of WT and various KO mice 7 days after SI. Thymus cellularities at day 0 (ie pre SI) should also be included to ensure a similar starting point.

We have now included data on day 0 (before damage) and at day 3 after damage in WT, *Il18*^{-/-}, and *Il18r1*^{-/-} mice. Interestingly while there was no difference between WT and *IL-18R* deficient mice, there was an increase in thymus cellularity at baseline in the *IL-18* deficient mice (**Extended Data Fig. 1b** and **Response Fig. 8**). However, despite this difference at baseline, we observed no difference in cellularity in any population at the nadir of cellularity at day 3 after TBI (**Extended Data Fig. 1b** and **Response Fig. 8**). Interestingly, in the *IL-18* deficient mice we saw no change in serum cortisol, although there may have been a slight increase at day 3 after damage (**Extended Data Fig. 1c** and **Response Fig. 8**), although it would be expected that increased cortisol would lead to reduced cellularity. However, due to these potential confounders at baseline in *IL-18* deficient mice, in the rest of the manuscript rely exclusively on *Il18r1*^{-/-} mice.

Reviewer #2:

Remarks to the Author:

In this study, the authors report that inflammatory IL-18 produced following thymic tissue damage impairs thymic regeneration. They used a combination of mouse models and cell culture assays to demonstrate the origin of IL-18 secretion and showed that it induced a cytotoxic response by resident NK cells that limits tissue regeneration. Finally, they identified mTECs as potential targets of this cytotoxic response. The manuscript is well written, the methods are appropriate, and the work is significant as there is a need to improve strategies for thymus regeneration. The study also has implications when considering IL-18 as an immunotherapeutic agent for cancer.

We appreciate the general enthusiasm for our manuscript. We have attempted to respond to the comments and concerns and feel that the manuscript is in much better shape because of it.

One of the weaknesses of the study is the lack of data on spatial distribution of the different cell types involved in this pathway (*IL-18* producing cells, cytotoxic NK cells, and TECs targeted by NK cells). For example, are

the stromal cells and dendritic cells that produce IL-18 near Il18r+ NK cells? Where are the granzyme+/perforin+ NK cells located in the thymus (cortex vs medulla)? Demonstrating proximity between cytotoxic NK cells and dying TECs would help support the conclusions.

To address this, we have now improved our imaging analysis clearly showing the acute increased frequency of NKp46+ cells in the thymus early after damage. We could also quantify this data and show that there are increased frequency of NK cells in both the cortex and medulla (Fig. 4e, and Response Fig. 9). Furthermore, nearest neighbor analysis revealed that while at baseline NK cells are closer to cortical than medullary epithelium, after damage there is a significant reduction in the distance between NKp46+ cells and both cortical and medullary epithelium (Fig. 7e-f, and Response Fig. 9).

Other comments:

Do Il18r-negative NK cells also upregulate granzyme and perforin following injury?

Do Il18r-negative NK cells also upregulate granzyme and perforin following injury?

This is an excellent question. To address this we have now and re-analyzed our data to include expression in both IL-18R- and IL-18R+ populations. This clearly shows that while there is upregulation of IFN γ and perforin in both IL-18R+ and IL-18R- NK cells after damage, there is a greater increase in IL-18R+ cells suggesting that these effector molecules are at least partially driven by IL-18 signaling (Fig. 6c and Response Fig. 10). In contrast, we found no such association with Granzyme B suggesting that its regulation is independent of IL-18. Consistent with these findings, we found reduced expression of Perforin in Il18r1 deficient mice, and reduced levels of IFN γ and Perforin in Il18r1^{ANK/ILC1} mice (Fig. 6d-e and Response Fig. 10).

Fig. 6D – why is basal expression of IFN γ and perforin much higher than in Fig. 5B? Which cell population is shown?

Good observation. This can likely be attributed to two things. Firstly, there is always some experimental variation that will lead to different levels. In our hands we have found considerable experiment to experiment variability in the GREAT IFN γ -reporter mice. Secondly, the data in (old) Fig 5b was gated on a broader population of cells (NK1.1+TCRb-) compared to Fig. 6d which was gated on CD49b_+ cells.

It would be helpful to have a better phenotyping of MHC and Rae1 expression in TECs. For example, is expression changed in all subsets of mTECs (Aire+, Ccl21+, Dcl1+, etc.)?

As described above in reference to reviewer 1 question in the same vein, to address this, we performed a comprehensive analysis of MHCII expression on distinct TEC subsets. Using our scRNAseq datasets, we now include data that demonstrate TECs as the only populations to show broadly decreased expression after damage (Revised Fig. 7a and Response Fig. 6, copied below). We have also now shown this in detailed flow cytometry data demonstrating all TEC subsets (except for Tuft cells) but not endothelial, mesothelial, or fibroblasts decrease their expression (Fig. 7b, and Response Fig. 6, copied below). Interestingly, given the increase in IFN γ it may be expected that there would be an increase in MHCII expression, and indeed when we look at CD45+ subsets we do in fact find an increase in MHCII in Macrophages, cDC1 and some thymocyte subsets (Revised Figure 7b, Extended Data Fig. 7a and Response Fig. 6, copied below).

Reviewer #2

(Remarks to the Author)

Thank you to the authors for addressing the concerns raised through additional experiments. While I am generally satisfied by the responses, I have comments on the new data on NK cell killing of TECs (Figure 7C and 7D). Why are the majority of the cells staining positive for 7AAD and Annexin V, even in the absence of NK cells? Is the data from this assay reliable if performed with cells that are mostly dead to start with?

We appreciate the acknowledgment of several additional experiments to address the original comments.

With respect to the concerns regarding the high degree of baseline cell death in our *ex vivo* TEC-killing assay (Fig. 7d), there are several factors that may be contributing to this. First, what we show in that panel are cTECs and mTECs isolated from WT mouse thymus 3 days post SL-TBI which were then cultured alone or with NK cells for several hours. Given the short timeframe after ionizing radiation, we would indeed expect a relatively high degree of cell death in TECs. Our data (below) shows that TEC viability at day 0 is higher than at day 3. However it also shows that there is relatively high cell death associated with the isolation and extraction protocols, which require enzymatic digestion with DNase/Collagenase and vigorous pipetting (1.5 hours), combined with long incubation (4-5 hours) at 37 degrees in media designed for NK cell cytotoxicity assay and not preserving TECs (not to mention TECs being notoriously difficult to keep alive in culture conditions). Isolating TECs *ex vivo* and preserving them *in vitro* is challenging even under circumstances specially designed for their preservation. Extended Data Fig. 7c (and below) suggests that mTEC viability at baseline after extraction is 53% and 38% for mTECs and cTECs, respectively. Overall, while this comment raises an important question regarding baseline viability in control TECs, we believe this experiment overcomes significant technical challenges to demonstrate strong evidence for NK cell targeting of damaged TECs *despite* a high degree of background cell death.

An alternative assay such as performing histology to look for apoptosis in TECs while co-staining with an NK cell marker would help support the conclusions.

While we also considered whether it would be possible to show this phenomenon *in situ* by imaging studies as suggested, the combination of the high cellular density of the thymus, high degree of cell death in the far more numerous thymocytes, together with the fact that TECs have many projections and cell bodies making individual TECs difficult to identify made this enormously challenging to achieve. Instead, we used AI algorithms to closely approximate mTECs and cTECs and their nearest distance relative to Np46⁺ cells (Fig. 7g).

I would also recommend making the flow plots in extended Figure 3 bigger and modifying either the axis of the plots or the table to harmonize the names of the markers used.

We have enlarged our flow plots and have changed the layout to make it easier to follow the gating strategy.

Reviewer #3

(Remarks to the Author)

The manuscript presents interesting findings, effectively combining a range of germline and conditionally gene-deleted mouse models, along with antibody-mediated blocking strategies, to support its conclusions. This study demonstrates that acute thymic injury triggers caspase-1 activation and subsequent IL-18 production, which impairs thymus regeneration. IL-18 is produced by both hematopoietic and non-hematopoietic cells in response to damage. IL-18 receptor-expressing thymic NK cells are activated following injury and play a suppressive role in thymic repair. IL-18 stimulation enhances the cytotoxic activity of thymic NK cells, leading to aberrant targeting of thymic epithelial cells (TECs), thereby further hindering thymus regeneration.

A limitation of the study is the relatively modest and variable phenotype observed in thymic cellularity following damage, which necessitates the use of a large number of mice to support conclusions based on differences that are not particularly striking. That said, all the data point in the same direction, supporting the overall message of the study.

In my view, a major point of concern is the presentation of *ex vivo* data analyzing the cytotoxic capacity of thymic NK cells harvested from mice following SL-TBI. These experiments, involving targets such as RMA-S cells or thymic epithelial cells (TECs), are central to the authors' conclusions. Therefore, the clarity of their presentation and their capacity to convince (Fig 5F, Fig 6I, Fig 7E) should be improved to strengthen the study.

There are additional comments that should be addressed:

We appreciate the overall enthusiasm of this reviewer, especially coming in at the resubmission stage. One of the biggest concerns raised by this reviewer is to do with how we have presented the data in Fig. 5f, Extended Data Fig. 6i and 7e. For each of these panels, we chose a relative value so that we could combine multiple experiments with differing absolute values, largely we believe for technical and not biological reasons. To address this we have provided data of absolute values (cell death for 5f and 6i – now Ext Data Fig. 6i, and cell number for 7e) of representative experiments. Using this representative experiment approach shows the same statistically significant effect, albeit with reduced number of dots on the plots.

- Fig. 1F and 1I: *IL-18*^{-/-} mice display increased thymic cellularity at baseline (Extended Fig. 1B), which complicates the interpretation of data at day 7 post-SL-TBI.

This point is well taken and something we discussed as this project developed. However, we ultimately do not feel that this impacts the findings for a number of reasons. First and foremost, while the baseline cellularity is indeed higher in *IL-18* deficient animals, there is no difference between WT and KO animals at the nadir of cellularity of day 3 post-TBI (Ext Data Fig. 1b). This suggests that recovery after damage is starting from the same low point. Secondly, baseline *Il18r1*^{-/-} thymus cellularity is not higher than WT controls and this was the motivation for choosing *Il18r1*^{-/-} in later experiments

(Figs. 6d, 7e-f). While it is unclear why there is a difference between IL-18 and IL-18R deficient mice at baseline, together, we feel our data convincingly demonstrate a novel role for IL-18 in the process of thymus regeneration.

The use of anti-IL-18 treatment in Fig. 1l provides a valuable alternative approach to support the role of IL-18 in thymus regeneration. However, the experimental setting in Fig. 1l is markedly different and not directly comparable. To strengthen the conclusions, an experiment using anti-IL-18 antibody with analysis performed 7 days post-SL-TBI should be included.

While it is true that the experimental condition in Fig. 1l is slightly different, and this manuscript was not designed to be presented as the development of a therapeutic strategy, it is worth noting that the model used in Fig. 1i is more clinically relevant being a model of HCT conditioning-related damage. In early experiments we attempted to block IL-18 signaling in both sublethal TBI as well as after hematopoietic cell transplant and found more consistent results in the HCT model. It is not yet clear why there is a difference in these models, however, it is possible that this is due to differences in timing and/or dosing. The therapeutic translation of these findings is of active interest for the lab, however, studies attempting to repeat IL-18 blockade in the sublethal setting would likely require more time and effort than warranted for this mechanistic paper and would only be in the service of neatness rather than scientific rigor.

-Fig 2E: The contribution of hematopoietic cells to IL-18 production following SL-TBI is explored in Fig. 2l using mice with IL-18-deficient dendritic cells. However, the absence of significant differences between IL-18^{WT} mice and those lacking IL-18 in dendritic cells suggests that additional hematopoietic cell types may be involved. To further support the conclusion that hematopoietic cells contribute to IL-18 production, including a control group in which IL-18^{-/-} bone marrow cells are transferred into WT recipients would support the overall argument.

When choosing *Cre* specific mouse lines with which to cross *Il18^{fl/fl}* mice, at the time the only transcriptomic data we had available was the published data shown in Extended Data Figure 2a (GSE56928) (Ki, S. et al., 2014), which only implicated *Il18* expression in cDC1 and cDC2. Based on this we chose to delete *Il18* in *Zbtb46*-expressing cells, which covers cDCs and endothelial cells. As indicated by the reviewer, this data suggests that there is a reduction in the amount of IL-18 being secreted, but it is still elevated compared to baseline suggesting alternate sources. Our subsequent single cell dataset showed wider expression in other hematopoietic cells like macrophages but also non-hematopoietic mesothelial cells. While we performed the chimera experiments to test the effects of non-hematopoietic cells, ultimately our take home message from the figure is that there multiple sources and therefore individual approaches like *Zbtb46*-cre or BM chimeras (WT into KO) do not in fact account for all IL-18 production. For this reason, another chimera experiment looking at KO donors into WT hosts would only highlight the same finding as we would not expect to get complete ablation in that model either (due to likely mesothelial production). Since the relative contributions of specific IL-18 sources was not a major focus of the manuscript, we have elected instead to take care

to highlight these other likely hematopoietic active IL-18 contributors, and feel that this explanation and the existing data support should suffice. We hope that this approach will be sufficient.

-Fig 4I and 4J: In contrast to the previous experiments, the protocol used in these two figures includes recombinant IL-18 (rec IL-18) injections administered on day 3 post-TBI. What is the rationale for modifying the protocol to include IL-18 administration after TBI? Is the contribution of NKp46⁺ cells observable only in the context of this non-physiological presence of IL-18? It would be important to show the results of experiments assessing thymic cellularity 7 days post-SL-TBI without IL-18 injection to clarify this point.

This experiment has been the single most puzzling aspect of our study. While it is still not entirely clear why our model required supraphysiologic amounts of IL-18, there is some evidence that there may be other mechanisms at play such as increased NK cell numbers at baseline in Cre⁺ mice (Ext Data Fig. 5g and below), suggesting that even in the absence of IL-18R signaling there may be increased overall NK activity. We maintain that Fig. 4f critically implicates NK1.1⁺ population in suppressing thymus regeneration and that Fig 4g is strongly supportive of a connection between these IL-18 and NK1.1⁺ population mechanisms of action. We feel that addition of data such as the fact that we see no difference in thymic cellularity when we give this same supraphysiological dose of IL-18 to mice with a specific deletion of *I18r1* on T and NKT cells (using *I18r1^{fl/fl} Lck-Cre⁺*) support our conclusion that NK cells, and not T or NKT cells, are responsible for regulating thymus regeneration in an IL-18 dependent fashion. In the interests of presenting all the data to the community, we have elected to include the negative data in both of these mouse models in the absence of exogenous IL-18 (Ext Data Fig. 5f and below). The possibility exists that NK dependent mechanisms but IL-18 independent mechanisms may be at play (IL-18 independent expression of GzmB, for example). Overall, we feel confident with the preponderance of evidence that NK cells limit thymic repair by an at least partially IL-18-dependent mechanism, even with the uncertain biology of *Ncr1-cre* mice and we do appreciate that despite this concern the reviewer agrees that “all the data point in the same direction, supporting the overall message of the study”.

-Fig 4J: It remains unclear whether the recruitment or increased number of NK1.1⁺ cells in the thymus following SL-TBI is dependent on IL-18 signaling. Specifically, do NK1.1⁺ cell numbers increase in the thymus of IL-18^{-/-} or IL-18R1-deficient mice post-TBI? Addressing this point would clarify whether IL-18 is required for the accumulation or expansion of NK cells in the thymus in response to damage. -Lines 287-297 suggest that the increase in NK cell numbers in the thymus is due to cell expansion. However, the possibility that NK cells may be recruited from the periphery has not been addressed. This potential mechanism should be discussed

Our data in the manuscript suggests that IL-18 does not seem to be required for the increased number of NK cells in the thymus after damage. This is shown looking at the relative cell increases in IL-18R⁺ and IL-18R⁻ NK cells at day 3 after damage (Fig. 6a-b). However, to further address this point, we have now included data in IL-18R KO mice that demonstrates that there is no difference in

NK cell number between WT and KO cells at day 3 (**Ext Data Fig. 6b and below**). In addition, our collaborators have recently opublished a study looking at recirculating vs resident cels in the thymus using the parabiosis model (Lemarquis et al. 2025, Immunity). Using this elegant model we could interrogate the “intrathymic” vs “extrathymic” status of NK cells before and after damage. Note, we grant that our definition of “extrathymic” only represents the donor-derived infiltrative NK cells and not the presumably same degree of circulatory NK cells that bear the same CD45.1/CD45.2 congenic marker as the host mouse. Using this approach we could show that following TBI there is both an increase in the cells already in the thymus, as well as an influx coming from circulation; although influx of cells represents a higher proproportion of the increase than expansion (**Ext. Data Fig. 6d**)

With respect to the comment about lines 287-297, I think this may have just been poor wording on our behalf. We were not trying to make an assertion as to whether it is expansion or recruitment and have changed our language to refer to the “increase” in NK cells and highlight this is likely due to both expansion and influx.

-Fig 5F. Presenting target cell killing as relative values is not very convincing. The figure should instead display non-normalized data as the percentage of specific lysis.

Additionally, including a representative FACS plot of the co-culture after 5 hours would provide a clearer picture of killing efficiency. Furthermore, according to the Materials and Methods, splenic NK cells from mice treated with poly(I:C) were used as a positive control. This control should be shown in the figure to allow proper evaluation of the thymic NK cells' capacity to kill target cells.

- Fig. 6I: The same comment applies as for Fig. 5F. Presenting target cell killing as relative values is not very convincing. The figure should instead display non-normalized data as the percentage of specific lysis. Additionally, including a representative FACS plot of the co-culture after 5 hours would provide a clearer picture of the killing efficiency.

As discussed above, we chose to show relative counts so that we could combine several experiemnts with disparate absolute counts. We believe that these differences in absolute values are entirely technical and relate to the culturing conditions before the assay (ie, confluence of RMA/S cells). On the reviewers suggestion, we have now included absolute values for both cell killing assays (Figs. 5f

and 6i, which is now Ext Data Fig. 6i) as well as the cell counts (Fig. 7f), which are shown above (and again below).

- Fig. 6H: The IFN- γ plots show only biological duplicates. To ensure statistical validity, the experiments should be reproduced with additional replicates.

We agree that the replicate number for (what was **Fig. 6h**, now **Ext Data Fig. 6g**) was low, and we have now added an additional experiment. Unfortunately due to the availability of IFN γ reporter (GREAT) mice and recombinant IL-18 that was on back order, we have only been able to add a low number of mice. However, we also contend that this whole experiment with recombinant IL-18 (what was **Fig. 6f-i**, now **Ext Data Fig. 6f-i**), while confirmatory, is also not key to the conclusions of the study – especially given this particular panel recapitulates the well established role of IL-18 to induce IFN γ . For these reasons and because we feel that this result does not demonstrate any novel IL-18 function - including an elegant recent study looking at steady-state responses to inflammatory cytokines like IL-18 (Cui et al 2023 Nature) - we have now elected to move this experiment into the extended data figures.

-Lines 326-328: Given the variability of the model, the conclusion that "these data support IL-18 triggering NK perforin upregulation/release more than granzyme B" should be either mitigated or described with greater caution.

Our data suggests that while IFN γ , Perforin and Granzyme B are all increased by NK cells after damage (**Fig. 5b**), when looking at the number of cells producing these cytokines – as a measure of who is the main contributor - only Perforin was expressed mostly by NK cells. Both IFN γ and Granzyme B were produced mostly by NKT cells (**Fig. 5d**). Furthermore, when we look at cells that are responsive to IL-18 signaling, as measured by expression of IL-18R, we found that both Perforin and IFN γ were increased in IL-18R^{hi} NK cells. In contrast, Granzyme B is actually higher in IL-18R^{lo} NK cells (**Fig. 6d**). For these reasons we believe that while Granzyme B is important for limiting repair in addition to Perforin – as evidenced by knockout data (**Fig. 6j**) – only perforin is regulated by IL-18 signaling in NK cells. We have tried to make this plain in the description by stating explicitly "...absence of either Perforin or Granzyme B can improve regeneration, our data suggests that only Perforin expression is regulated by IL-18 signaling in NK cells (**Fig. 6d**)."

-Fig. 7: Downmodulation of MHC I serves as a signal to trigger NK cell cytotoxicity in the context of activating ligand expression. In addition to evaluating MHC I expression, the expression of NK cell activating ligands, such as Rae1, should also be assessed on TECs. Increase of an NK activating ligand was something we were very keen to demonstrate. While we did show increased Rae1 expression on global cTECs and mTECs in our original submission, as highlighted in our reviewer response, when we performed further studies looking at these individual TEC subsets, we found that Rae1 expression was less robust or reproducible. We have subsequently spent some time trying different, newly available since our initial submission commercial antibodies and have thus now included data showing increased RAE1 across range of TEC subsets (Fig. 7c and below).

Minor comments:

- Line 224: NK cells repeated twice
- Line 257 refers to an increased expression of NKG2D at the protein level following TBI; however,

this data is not shown.

- Line 261 incorrectly references Extended Fig. 5F for the annotation of Itga1 and Itga2 in the scRNA-seq data. These annotations are actually shown in Extended Fig. 6A and should be correctly cited to avoid confusion.

We have gone through the manuscript and mended these minor errors.